# IRE1α determines ferroptosis sensitivity through regulation of glutathione synthesis

Dadi Jiang [1,6] ✉, Youming Guo[1,6], Tianyu Wang[1,2], Liang Wang [1], Yuelong Yan [3], Ling Xia[1], Rakesh Bam [4], Zhifen Yang[4], Hyemin Lee [3], Takao Iwawaki[5], Boyi Gan [2,3] & Albert C. Koong [1,2] ✉

Cellular sensitivity to ferroptosis is primarily regulated by mechanisms mediating lipid hydroperoxide detoxification. We show that inositol-requiring enzyme 1 (IRE1α), an endoplasmic reticulum (ER) resident protein critical for the unfolded protein response (UPR), also determines cellular sensitivity to ferroptosis. Cancer and normal cells depleted of IRE1α gain resistance to ferroptosis, while enhanced IRE1α expression promotes sensitivity to ferroptosis. Mechanistically, IRE1α's endoribonuclease activity cleaves and down-regulates the mRNA of key glutathione biosynthesis regulators glutamate-cysteine ligase catalytic subunit (GCLC) and solute carrier family 7 member 11 (SLC7A11). This activity of IRE1α is independent of its role in regulating the UPR and is evolutionarily conserved. Genetic deficiency and pharmacological inhibition of IRE1α have similar effects in inhibiting ferroptosis and reducing renal ischemia–reperfusion injury in mice. Our findings reveal a previously unidentified role of IRE1α to regulate ferroptosis and suggests inhibition of IRE1α as a promising therapeutic strategy to mitigate ferroptosis-associated pathological conditions.

The cellular defense against ferroptosis occurs primarily through four mechanisms with distinctive subcellular locations: GPX4 utilizes reduced glutathione (GSH) to detoxify lipid hydroperoxides in both the cytosol and mitochondria[1,2]. FSP1 and DHODH convert ubiquinone to ubiquinol that traps lipid peroxyl radicals at the plasma membrane and the inner mitochondrial membrane, respectively[3–5]. In addition, GCH-1 suppresses ferroptosis throughout the cell by producing tetrahydrobiopterin (BH₄), another radical trapping antioxidant[6–8]. The ferroptosis-inducing agent erastin activates the eIF2α-ATF4 signaling axis[9], which is also induced during the UPR through PERK activation[10]. The mammalian UPR consists of multiple signaling branches that carry out distinct albeit redundant cellular functions[10]. However, the mechanisms of how other key UPR components regulate ferroptosis are largely unknown.

## Results

### Loss of IRE1α dictates resistance to ferroptosis induction

Although earlier studies identified that certain ferroptosis inducers (FINs) activate the eIF2α-ATF4 signaling axis by interacting with the UPR, defining the role of the UPR in regulating ferroptosis has been elusive[9]. As the most evolutionarily conserved UPR branch, the IRE1α-X-box binding protein 1 (XBP1) pathway determines cell fate under endoplasmic reticulum (ER) stress by regulating protein homeostasis and chaperone expression, but has never been functionally implicated in ferroptosis[10]. We found that when IRE1α was depleted in MDA-MB-231 triple-negative breast cancer cells, these cells became significantly more resistant to ferroptotic cell death induced by either erastin, an inhibitor of the cystine-glutamate antiporter SLC7A11, or cystine depletion (Fig. 1A and D; Supplementary Fig. 1A, C and D;

[1]Department of Radiation Oncology, The University of Texas MD Anderson Cancer Center, Houston, TX, USA. [2]The University of Texas MD Anderson Cancer Center UTHealth Houston Graduate School of Biomedical Sciences, Houston, TX, USA. [3]Department of Experimental Radiation Oncology, The University of Texas MD Anderson Cancer Center, Houston, TX, USA. [4]Division of Radiation and Cancer Biology, Department of Radiation Oncology, Stanford University School of Medicine, Stanford, CA, USA. [5]Division of Cell Medicine, Department of Life Science, Medical Research Institute, Kanazawa Medical University, Uchinada, Japan. [6]These authors contributed equally: Dadi Jiang, Youming Guo. ✉e-mail: djiang2@mdanderson.org; ACKoong@mdanderson.org

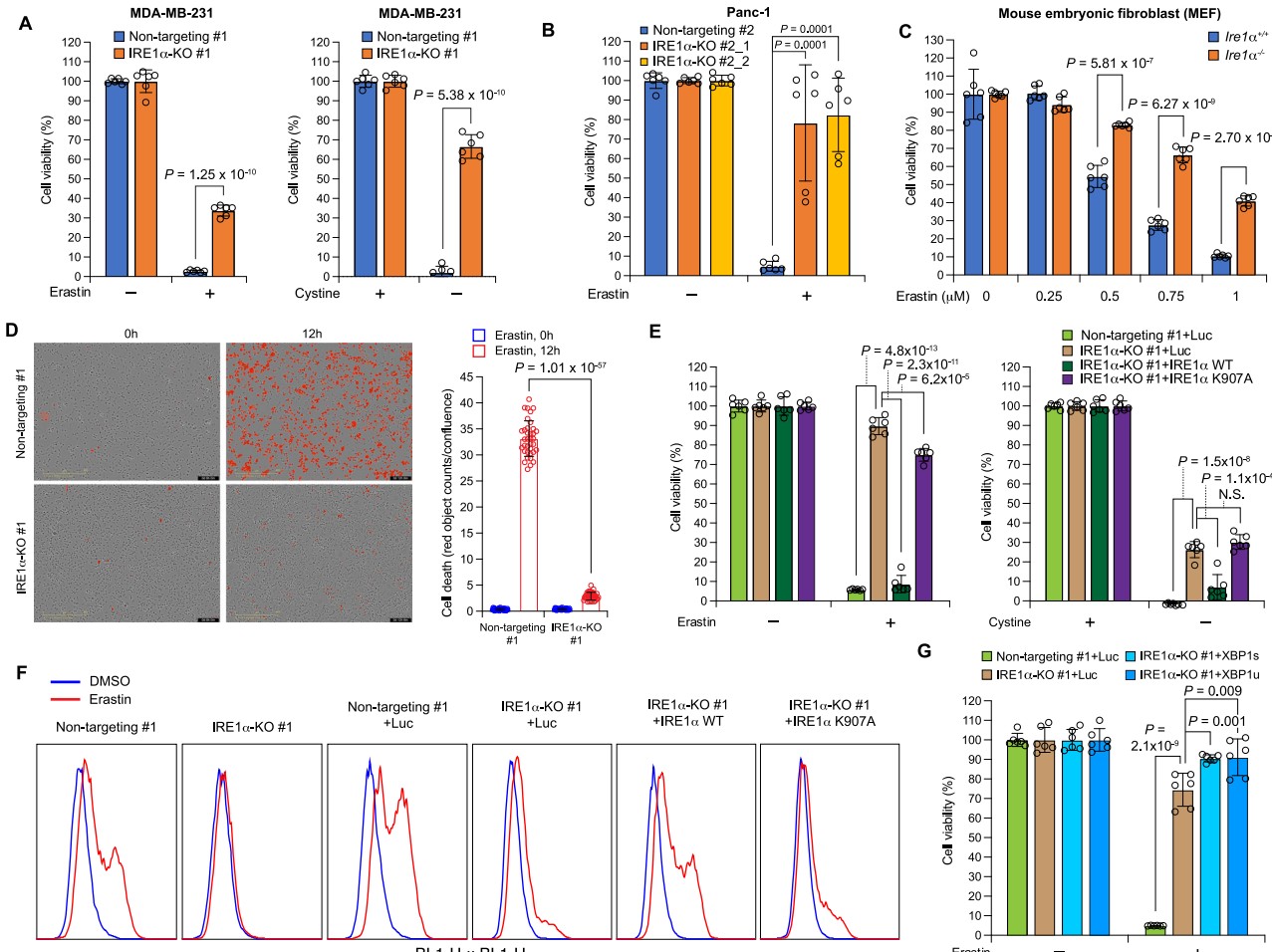

**Fig. 1 | IRE1α determines sensitivity to ferroptosis. A** Viabilities of control (non-targeting) and IRE1α-null MDA-MB-231 human triple-negative breast cancer cells upon treatment with erastin (2.5 µM, left) or cystine depletion (right). 24 h after erastin treatment or cystine withdrawal, cell viability was measured by CCK8 assay. Viability was set as 100% with 0 µM erastin (DMSO only) or 200 µM cystine. The same measurements were made with Panc-1 human pancreatic adenocarcinoma cells in (**B**) (2.5 µM erastin) and mouse embryonic fibroblasts (MEFs) in (**C**) (0.25–1 µM erastin). **D** Non-targeting and IRE1α-null MDA-MB-231 cells were subjected to IncuCyte Cytotox Red staining & imaging upon treatment with 10 µM erastin. Representative phase+red fluorescence channel images from 0 and 12 h are shown on the left and quantitation on the right. **E** Non-targeting and IRE1α-null MDA-MB-231 cells overexpressing firefly luciferase (Luc), wild-type IRE1α (WT) or the K907A mutant IRE1α were subjected to 2.5 µM erastin treatment (left) or cystine depletion (right) and CCK8 assay was performed after 24 h. Viability was set as 100% with 0 µM erastin (DMSO only) or 200 µM cystine. **F** Cells used in (**A**) and (**E**) were subjected to 2.5 µM erastin treatment for 18 h, then C11-BODIPY 581/591 staining and flow cytometry analysis were performed to measure lipid peroxidation. **G** Non-targeting and IRE1α-null MDA-MB-231 cells overexpressing Luc, spliced/activated XBP1 (XBP1s), or unspliceable XBP1 (XBP1u) were subjected to 5 µM erastin treatment and CCK8 assay was performed after 24 h. Viability was set as 100% with 0 µM erastin (DMSO only). Data are presented as mean ± s.d. in all panels except (**F**). $n = 6$ in all panels except (**F**) and (**D**). $n$ indicates independent repeats. In (**D**), quantification is based on $n = 36$ images in each group. Unpaired, two-tailed Student's $t$ tests were performed to calculate the $P$ values for all the statistical analyses. N.S. not significant. Source data are provided as a Source Data file.

Supplementary Fig. 3C; Supplementary Fig. 4E). This observation can also be extended to another cancer cell line (Fig. 1B; Supplementary Fig. 1A) and to non-cancerous cells (Fig. 1C; Supplementary Fig. 1A), suggesting a universal molecular regulatory circuit between IRE1α and the cellular ferroptotic machinery. Another FIN with a distinct mechanism of action in inducing ferroptosis, RSL3[11], had similar differential ferroptosis induction in wild-type and IRE1α-null MDA-MB-231 cells, suggesting that the effect of IRE1α on ferroptosis sensitivity is not limited to erastin or cystine starvation-induced ferroptosis (Supplementary Fig. 1E, F). Expression reconstitution with wild-type IRE1α, but not the endoribonuclease (RNase)-dead K907A mutant, re-sensitized IRE1α-null MDA-MB-231 cells to ferroptosis induction (Fig. 1E; Supplementary Fig. 2; Supplementary Fig. 4A), indicating that the RNase activity of IRE1α is responsible for this regulation. Compared to control cells, IRE1α-null MDA-MB-231 cells displayed attenuated lipid peroxidation upon erastin treatment, whereas reconstitution with wild-type IRE1α but not the K907A mutant restored lipid peroxidation (Fig. 1F),

demonstrating that the change in sensitivity to ferroptosis is due to IRE1α's effect on lipid peroxidation. As the unconventional XBP1 mRNA splicing by IRE1α and the subsequent activation of the spliced XBP1 protein expression are the major mechanism of ER homeostasis regulation by IRE1α[10], we tested whether XBP1 plays a role in IRE1α-mediated ferroptosis sensitization. Surprisingly, overexpression of either spliced XBP1 (XBP1s) or unspliced XBP1 (XBP1u) failed to re-sensitize IRE1α-null MDA-MB-231 cells to ferroptosis induction (Fig. 1G; Supplementary Fig. 1B). Furthermore, transient knockdown of XBP1 expression through doxycycline-inducible shRNA in MDA-MB-231 cells did not change their sensitivity to erastin-induced ferroptosis (Fig. 2B). These findings suggest that the regulatory function of IRE1α in ferroptosis is independent of XBP1.

To ascertain a comprehensive role of the UPR in cystine uptake inhibition-induced ferroptosis, we examined the status of the three canonical UPR pathways: (1) IRE1α-XBP1s, (2) PERK-ATF4-eIF2α and (3) ATF6 in both MDA-MB-231 and Panc-1 cells upon erastin

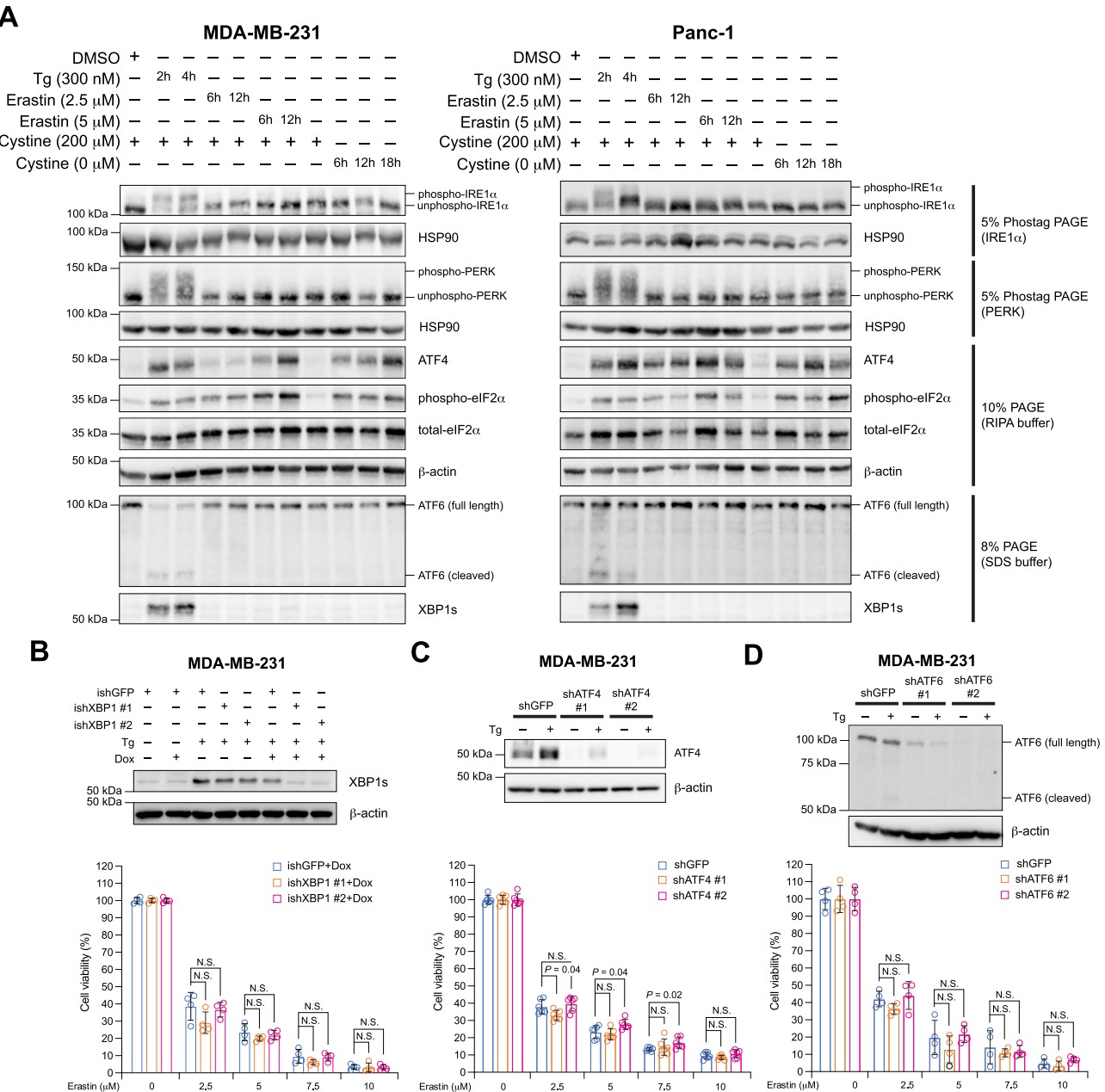

**Fig. 2 | The unfolded protein response (UPR) is not activated during ferroptosis induced by cystine deficiency. A** MDA-MB-231 (left panel) and Panc-1 (right panel) cells were treated with thapsigargin (Tg), erastin, or cystine starvation with the indicated concentration and duration, then the expression and modification status of key UPR pathway proteins were analyzed with Western blotting. To examine the phosphorylation status of IRE1α and PERK, Phostag PAGE was used to resolve the whole-cell protein lysates. To detect ATF6 cleavage, SDS sample buffer was used to ensure efficient membrane protein extraction. The rest of the analyses used RIPA buffer for protein extraction. HSP90 and β-actin were used as loading controls. Total GCN2 from the same blot in Supplementary Fig. 3A was used as the loading control for the 8% PAGE. **B** MDA-MB-231 cells were transduced with doxycycline (Dox)-inducible shRNA targeting GFP (ishGFP) or XBP1 (ishXBP1). ishGFP was used as a negative knockdown control and two independent XBP1 shRNAs were used. Top panel: Western blotting analysis of spliced XBP1 (XBP1s) expression upon Tg

treatment or cystine starvation. In both types of cells, upon ER stress induction by thapsigargin (Tg) treatment, all the three UPR pathways were activated, as determined by phosphorylation of IRE1α, PERK, eIF2α, and cleavage of ATF6 to generate the activated form, as well as increased expression of the downstream

(300 nM, 6 h) and/or Dox (0.5 mg/ml, 24 h) treatment. β-actin was used as a loading control. Bottom panel: shRNA-transduced and Dox-treated MDA-MB-231 cells were treated with increasing concentrations of erastin. 24 h after erastin treatment, cell viability was measured by CCK8 assay. Viability was set as 100% with 0 μM erastin (DMSO-only). Similarly, the expression of ATF4 (**C**) or ATF6 (**D**) was inhibited in MDA-MB-231 cells with two independent non-inducible shRNAs per gene and shGFP as a negative control. The expression level of ATF4 or ATF6 upon Tg treatment was analyzed with Western blotting (top panels) and the effect on erastin-induced ferroptosis was analyzed by CCK8 (bottom panels). Data are presented as mean ± s.d. in the bottom panels of (**B**), (**C**) and (**D**). *n* = 4 in the bottom panel of (**B**) and *n* = 6 in the bottom panels of (**C**) and (**D**). *n* indicates independent repeats. Unpaired, two-tailed Student's *t* tests were performed to calculate the *P* values for all the statistical analyses. N.S. not significant. Source data are provided as a Source Data file.

transcription factors XBP1s and ATF4 (Fig. 2A). In contrast, neither erastin nor cystine starvation activated the UPR signaling branches, except for phosphorylated eIF2α and ATF4, which were robustly induced by either condition, consistent with a previous report[9].

As eIF2α phosphorylation and ATF4 expression were induced by erastin or cystine starvation independent of PERK phosphorylation, we examined whether another eIF2α-ATF4 upstream activator, GCN2[12], was activated in this context. As expected, GCN2 was phosphorylated by either erastin or cystine starvation but not Tg, likely due to its role in sensing amino acid deficiency[12] (Supplementary Fig. 3A). Taken together, we identified that while inhibition of cystine uptake induced ferroptosis, the UPR was not activated under these conditions, suggesting that this type of oxidative damage in the lipid compartment does not affect protein folding homeostasis. Other chemicals known to induce ferroptosis, including RSL3 and buthionine sulfoximine (BSO), did not increase the expression of XBP1s or ATF4 (Supplementary Fig. 4G). Interestingly, BSO alone was insufficient to induce ferroptosis in MDA-MB-231 cells (Supplementary Fig. 4F). In addition to the general lack of UPR activation during ferroptosis, inhibition of key UPR transcription factors XBP1, ATF4, or ATF6 expression did not alter the cellular sensitivity to erastin in MDA-MB-231 cells (Fig. 2B-D).

## IRE1α suppresses glutathione synthesis through RIDD

As one of the major cellular defensive mechanisms against ferroptosis, the peroxidase, GPX4, utilizes GSH as its cofactor to reduce lipid peroxides to lipid alcohols[1,2]. Therefore, cellular availability of GSH through glutathione synthesis is a critical determinant of sensitivity to ferroptosis. To investigate whether IRE1α mediates ferroptosis sensitivity through regulation of GSH availability, we measured cellular GSH levels in MDA-MB-231 cells with differential IRE1α expression. Compared to wild-type cells, IRE1α-null cells have significantly elevated GSH levels, indicating that the resistance to ferroptosis correlates with altered GSH metabolism in IRE1α-deficient cells (Fig. 3A). Expression reconstitution with wild-type IRE1α, but not the IRE1α K907A mutant

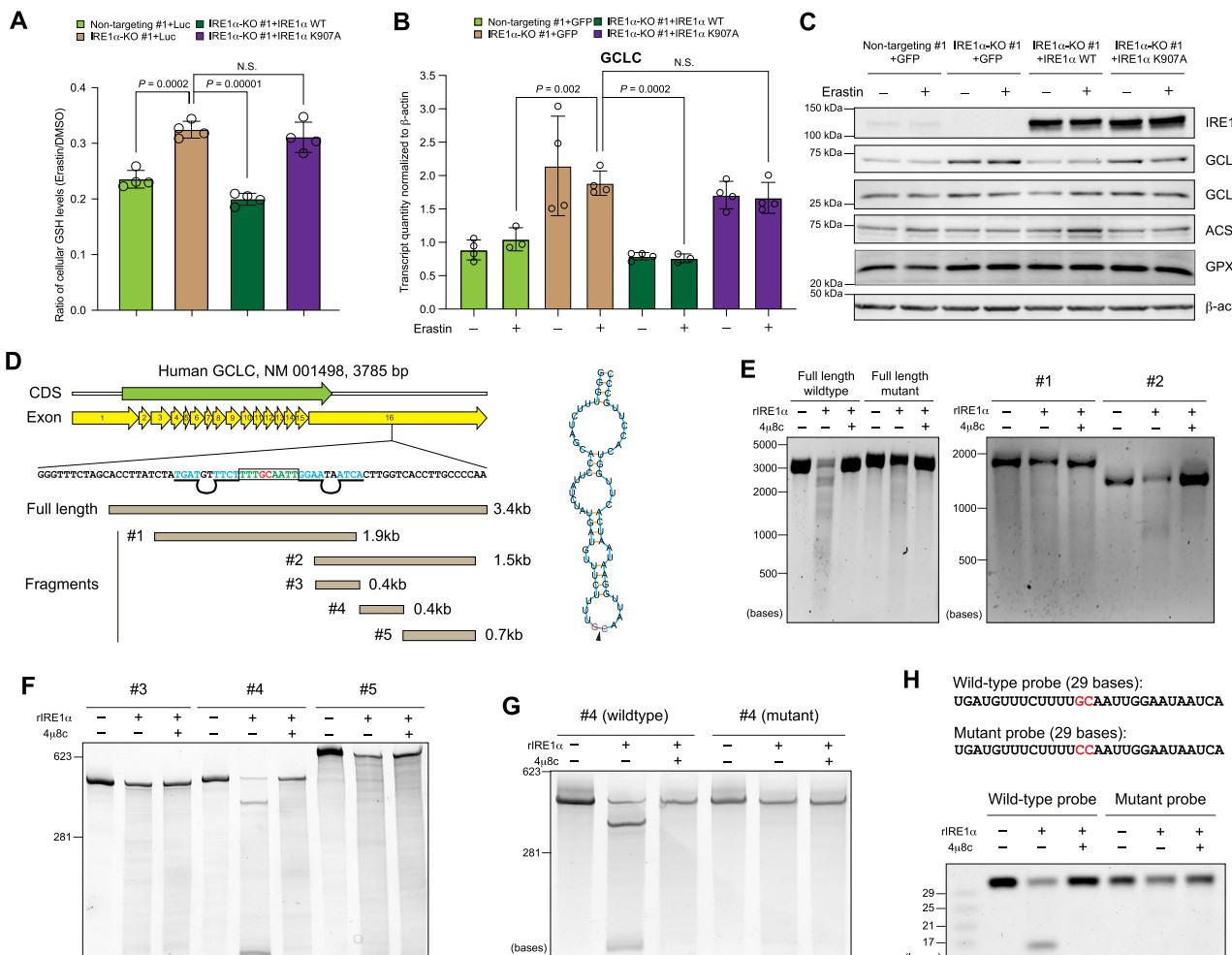

**Fig. 3 | GCLC is a target of the RIDD activity of IRE1α. A** Cellular reduced glutathione (GSH) levels in control (non-targeting) and IRE1α-null MDA-MB-231 cells overexpressing luciferase (Luc), wild-type (WT) or the K907A mutant IRE1α after 2.5 μM erastin treatment (12 h). **B** qRT-PCR analysis of GCLC mRNA expression in MDA-MB-231 cells with different IRE1α expression after DMSO or 10 μM erastin treatment (8 h). β-actin: normalization control. **C** Western blotting analysis of ferroptosis-related proteins in MDA-MB-231 cells with variable IRE1α expression ± erastin (10 μM, 6 hours). GFP: overexpression control. β-actin: loading control. **D** Left: schematic view of human GCLC transcript with coding sequence (CDS, green), exon boundaries (yellow), and fragments for in vitro transcription (light brown). The sequence flanking the identified IRE1α cleavage site is also shown with the GC cleavage site (red), loop sequence (green) and stem sequence (light blue). The pairing bases in the stem part are marked with black underlines and unpaired bases are marked with black loops. Right: the predicted secondary structure of the mRNA sequence flanking the GCLC IRE1α cleavage site (red). The black triangle marks the cleavage between G and C. **E** In vitro RNA cleavage assay using full length (wild-type and cleavage site mutated) and two large fragments of human GCLC mRNA (#1 and #2 as shown in **D**), incubated in the presence or absence of recombinant cytosolic portion of IRE1α (rIRE1α) ± IRE1α inhibitor 4μ8c (30 μM). Nucleotide base markers are shown on the left side. **F.** In vitro RNA cleavage assay using three smaller fragments of human GCLC mRNA (#3, #4 and #5 as shown in **D**). **G** In vitro RNA cleavage assay using wild-type and cleavage site-mutated (GC to CC) fragment #4. **H** In vitro RNA cleavage assay using wild-type and cleavage site-mutated (GC to CC) 29-base RNA probes within fragment #4. In (**A**) and (**B**), data are presented as mean ± s.d., n = 4 independent repeats. Unpaired, two-tailed Student's t tests were performed to calculate the P values for all the statistical analyses. N.S. not significant. Source data are provided as a Source Data file.

reduced GSH levels to that of the wild-type control cells, suggesting that the RNase function of IRE1α is responsible for regulating GSH levels (Fig. 3A). Cellular GSH synthesis is determined by the availability of the precursor amino acid cysteine as well as the rate-limiting enzyme glutamate cysteine ligase (GCL), which consists of a catalytic (GCLC) and a modifier (GCLM) subunit. We found that the GCL inhibitor BSO re-sensitized IRE1α-null cells to erastin-induced ferroptosis, indicating that GCL is involved in IRE1α-regulated GSH synthesis (Supplementary Fig. 4E). We then investigated whether IRE1α controls the expression of GCL enzymes and identified that IRE1α suppresses GCLC expression at both the mRNA and protein levels (Fig. 3B, C). This suppression is dependent on the RNase activity of IRE1α, as expression reconstitution with wild-type IRE1α but not the K907A mutant in IRE1α-null cells inhibited GCLC expression (Fig. 3B, C). In contrast, the expression of other ferroptosis regulators, including GCLM, ACSL4 and GPX4, was not significantly affected by IRE1α (Fig. 3C). Although previous studies reported that GPX4 protein level decreased during erastin (>10 μM) and other FIN treatments[13–17], we did not observe this in either MDA-MB-231 or Panc-1 cells during erastin (≤10 M) or cystine starvation induced ferroptosis (Fig. 3C; Supplementary Fig. 3A), suggesting that this may be a drug concentration and cell type-specific phenomenon.

Given that XBP1 does not play a significant role in ferroptosis regulation by IRE1α (Fig. 1G; Supplementary Fig. 1B; Fig. 2B), we tested whether the decrease in GCLC expression is mediated by the regulated Ire1-dependent decay (RIDD) activity of IRE1α through its RNase domain[18]. We found that the full-length human GCLC transcript was cleaved by recombinant human IRE1α in vitro, which can be inhibited by 4μ8c, a potent IRE1α RNase inhibitor[19] (Fig. 3E). We analyzed specific regions of the GCLC transcript by dividing it into two shorter fragments (#1 and #2) for in vitro transcription (Fig. 3D) and identified that fragment #2 (3' fragment, covering mainly the 3' UTR region) but not fragment #1 (5' fragment) was cleaved by IRE1α (Fig. 3E). We then further divided fragment #2 into three even shorter fragments (#3-5, Fig. 3D) and found that fragment #4 but not the other two fragments was cleaved by IRE1α (Fig. 3F). Within fragment #4, we identified a stem-loop-stem structure with the UG^C consensus sequence (UUG^CAA)[20] in the middle (Fig. 3D). When we mutated the cleavage site from UUGCAA to UUCCAA, fragment #4 as well as the full-length fragment was no longer cleaved by IRE1α (Fig. 3E and G). To further confirm the specificity of the cleavage, we synthesized a 29-base RNA oligo encompassing the cleavage site and found that it can be cleaved into the predicted fragments, which was completely inhibited by 4μ8c (Fig. 3H). When we used a mutant oligonucleotide with the same UUGCAA to UUCCAA mutation, cleavage was no longer observed (Fig. 3H). To demonstrate functionally that IRE1α regulates cellular sensitivity to ferroptosis through GCLC down-regulation, we over-expressed GCLC in IRE1α-null MDA-MB-231 cells reconstituted with wild-type IRE1α and found that GCLC overexpression completely restored the GSH level repressed by reconstituted wild-type IRE1α (Supplementary Fig. 4B, C) and resistance to erastin-induced ferroptosis (Supplementary Fig. 4A). Conversely, GCLC knockdown in IRE1α-null MDA-MB-231 cells re-sensitized these cells to erastin-induced ferroptosis (Supplementary Fig. 4C, D), consistent with the effect of the GCL inhibitor BSO (Supplementary Fig. 4E). Collectively, these findings reveal that IRE1α negatively regulates cellular GSH availability and maintains cellular sensitivity to ferroptosis by dampening GCLC expression through its RIDD activity.

Extracellular cystine is the predominant precursor for intracellular cysteine, the availability of which limits GSH synthesis[21]. SLC7A11 encodes the cystine/glutamate antiporter xCT, the major cystine transporter in the cell. Interestingly, we found that SLC7A11 mRNA is also a target of the IRE1α RIDD activity. Similar to its effect on GCLC, IRE1α suppresses SLC7A11 expression at both the mRNA and protein levels (Supplementary Fig. 6A, B). Consistent with their elevated SLC7A11 expression level, IRE1α-null MDA-MB-231 cells also demonstrated increased cystine uptake, which can be inhibited by IRE1α reconstitution (Supplementary Fig. 6C). Simultaneously, there is a reciprocal pattern of intracellular glutamate levels, which matches the cystine/glutamate antiporter function of SLC7A11 (Supplementary Fig. 3B). We further identified that a 3.3 kb fragment encompassing the coding sequence (CDS) of SLC7A11 was cleaved by recombinant human IRE1α in vitro, which can be inhibited by 4μ8c (Supplementary Fig. 6D, E). Within the SLC7A11 transcript sequence, we identified three UG^C consensus-containing stem-loop-stem structures, one of which is within the 3.3 kb fragment and the other two in the 3'UTR region (Site 1–3, Supplementary Fig. 6D). We synthesized 40-base RNA oligonucleotides encompassing the predicted cleavage sites and found that they can be cleaved into the predicted fragments, which was completely inhibited by 4μ8c (Supplementary Fig. 6F). When the UG^C cleavage sites were mutated into UCC, cleavage was no longer observed in all the three mutant oligonucleotides (Supplementary Fig. 6F). These findings suggest that IRE1α regulates multiple components of the GSH biosynthesis pathway through RIDD.

As IRE1α is the most evolutionarily conserved UPR sensor, we investigated whether it also determines ferroptosis sensitivity in lower organisms. It was recently demonstrated that ferroptosis can be induced in *C. elegans* through dietary ingestion of the polyunsaturated fatty acid (PUFA), dihomogamma-linolenic acid (DGLA; 20:3n-6)[22]. It was also found that iron accumulation and GSH depletion during late stage of *C. elegans* life induce ferroptosis and shortening of lifespan[23]. Consistent with these findings we showed that feeding *C. elegans* with DGLA significantly reduced the lifespan of the wild-type strain (N2) (Fig. 4A; Supplementary Fig. 7A). In contrast, the two *ire-1* mutant strains (RB925, *ire-1* deletion and SJ30, *ire-1* mutation in the kinase domain resulting in defective RNase activity; *ire-1* is the *C. elegans* homolog of human *ERN1* gene that encodes the IRE1α protein) were both resistant to DGLA treatment (Fig. 4B, C; Supplementary Fig. 7B, C). We further confirmed that the transcript of *C. elegans* homolog of *GCLC*, *gcs-1*, was cleaved by IRE1α and that cleavage was inhibited by 4μ8c (Fig. 4D). Although the transcript fragment that covers the whole CDS of *gcs-1* (full-length) was cleaved by IRE1α, only the first half of the full-length fragment (smaller fragment #1) was cleavable (Fig. 4D). Based on this finding we further identified a stem-loop-stem structure with the IRE1α cleavage site close to the 5' end of the *gcs-1* transcript that matches the UG^C consensus sequence (Fig. 4D and E). We synthesized a 40-base RNA oligonucleotide encompassing the cleavage site and found that it can be cleaved into the predicted fragments, which was completely inhibited by 4μ8c (Fig. 4F). When the UG^C cleavage site was mutated into UCC, the cleavage was abrogated (Fig. 4F). Together, these data reveal a previously unidentified, evolutionarily conserved regulatory mechanism between IRE1α and ferroptosis through RIDD activity on GCLC, a key enzyme in the GSH synthesis pathway.

## Pharmacological IRE1α inhibition de-sensitizes cells to ferroptosis

As the RNase activity of IRE1α can be inhibited pharmacologically[24], we tested the effect of specific IRE1α RNase inhibitors including 4μ8c[19] and MKC9989[25], on erastin-induced ferroptosis. These inhibitors significantly blocked erastin-induced ferroptosis in wild-type (Fig. 5A; Supplementary Fig. 3D) but not IRE1α-null MDA-MB-231 cells (Supplementary Fig. 3E), suggesting that they specifically inhibit IRE1α-mediated ferroptosis sensitization. Consistent with the effect of IRE1α on decreasing cellular GSH levels, both IRE1α inhibitors significantly increased cellular GSH levels (Fig. 5B). We also found that 4μ8c treatment increased GCLC mRNA expression in MDA-MB-231 cells compared to vehicle treatment (Fig. 5C). These data are consistent with the in vitro activity of this IRE1α inhibitor in inhibiting GCLC, SLC7A11 and gcs-1 mRNA cleavage as shown previously (Fig. 3E–H; Fig. 4D, F;

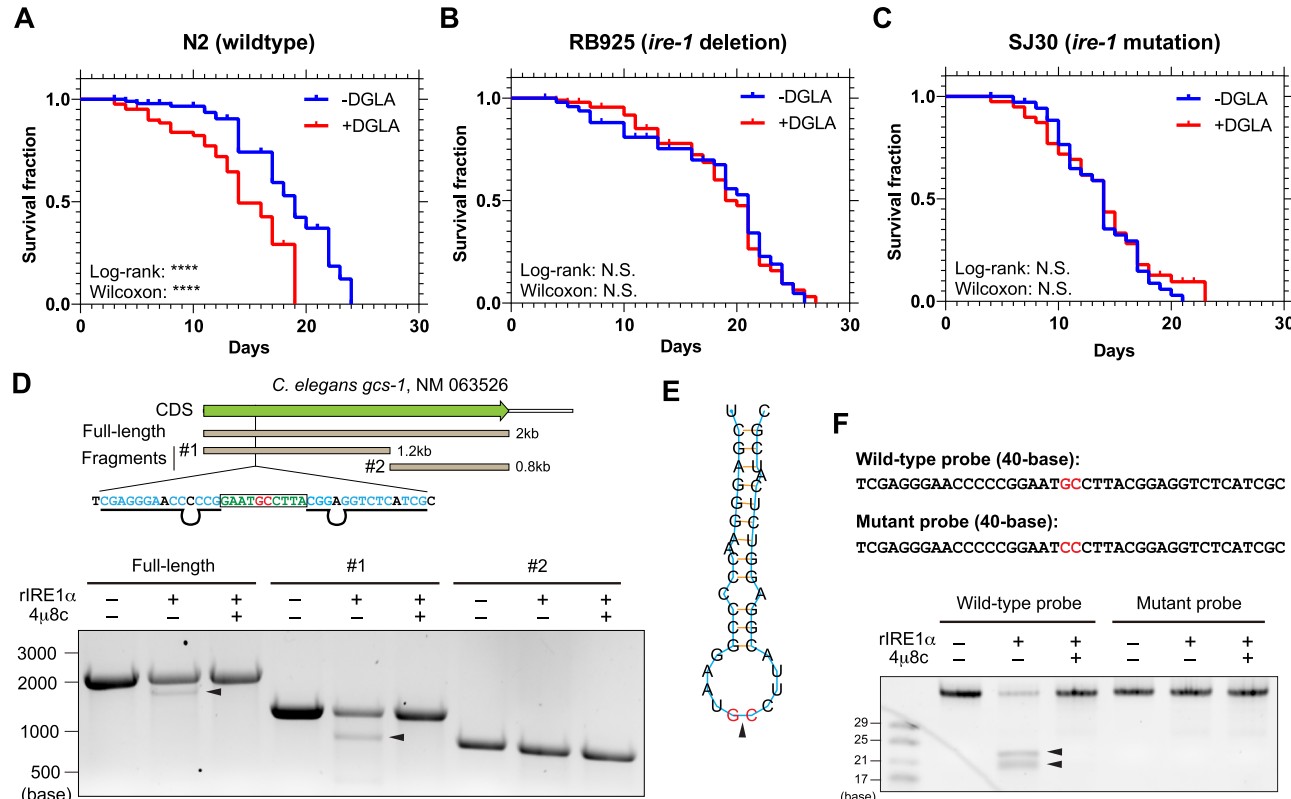

**Fig. 4 | IRE1α regulates ferroptosis sensitivity in *C. elegans*.** Kaplan-Meier survival curve of wild-type (N2, **A**), *ire-1* deleted (RB925, **B**) and *ire-1* mutated (SJ30, **C**) *C. elegans* ± dietary ingestion of the polyunsaturated fatty acid dihomogamma-linolenic acid (DGLA). Both two-sided Log-rank (Mantel-Cox) and Gehan-Breslow-Wilcoxon tests were used to calculate the *P* values. In (**A**), $n = 98$ (N2 - DGLA) and $n = 82$ (N2 + DGLA). In (**B**), $n = 101$ (RB925 - DGLA) and $n = 102$ (RB925 + DGLA). In (**C**), $n = 45$ (SJ30 - DGLA) and $n = 47$ (SJ30 + DGLA). **** $P \le 0.0001$. N.S.: not significant. Median survival: N2: 19 (-DGLA) and 14 (+DGLA); RB925: 21 (-DGLA) and 20 (+ DGLA); SJ30: 14 (-DGLA) and 14 (+DGLA). **D** Top, schematic view of the transcript of gcs-1 (GCLC homolog in *C. elegans*) with coding sequence (CDS) region shown on the top (green) and relative locations of the three fragments for in vitro transcription on the bottom (light brown). The sequence flanking the identified IRE1α cleavage site is shown with the GC cleavage site (red), sequence in the loop part

(green) and sequence in the stem part (light blue). The pairing bases in the stem part are marked with black underlines and unpaired bases are marked with black loops. Bottom: In vitro RNA cleavage assay using full-length or two smaller fragments of *C. elegans* gcs-1 mRNA (#1 and #2), incubated in the presence or absence of recombinant cytosolic portion of IRE1α (rIRE1α). Experiments were performed in the presence or absence of 30 μM IRE1α inhibitor 4μ8c. **E** The predicted secondary structure of the mRNA sequence flanking the gcs-1 mRNA IRE1α cleavage site (shown in red). The black triangle marks the cleavage between G and C. **F** In vitro RNA cleavage assay using wild-type and cleavage site-mutated (GC to CC) 40-base RNA probes within fragment #1 of *C. elegans* gcs-1 mRNA, incubated in the presence or absence of rIRE1α and 4μ8c. The number of RNA bases of the molecular markers are shown on the left side of the gels. The visible cleaved bands are marked by black triangles. Source data are provided as a Source Data file.

Supplementary Fig. 6E, F). To investigate the potential to further sensitize cells to ferroptosis by enhancing IRE1α activity, we overexpressed IRE1α in MDA-MB-231 cells. IRE1α overexpression significantly decreased GCLC mRNA expression (Fig. 5D) and increased erastin-induced ferroptosis (Fig. 5E). Overall, these results indicate that the regulation of ferroptosis sensitivity by IRE1α can be modulated pharmacologically. Although the development of specific IRE1α activators remains primarily at a preclinical stage, a promising cancer therapeutic approach would be to utilize small molecule IRE1α activators to induce ferroptosis selectively in tumors or to restore ferroptosis in tumors that have developed resistance to this mechanism of cell death.

## IRE1α inhibition reduces ischemic kidney damage

Previous studies have established the role of ferroptosis in promoting renal ischemia–reperfusion injury (IRI)[2,26,27]. To further translate our findings into a mammalian in vivo setting, we investigated the effect of genetic IRE1α deficiency in alleviating IRI. We established an inducible Ire1α-null mouse line by crossing the conditional *Ern1*[f/f] mouse strain[28] (*Ern1* is the gene symbol for Ire1α in mice) with the *Rosa26-CreERT2* strain[29] to generate the *Ern1*[f/f];*Rosa26-CreERT2*+ mice. Upon tamoxifen administration, *Ire1α* deletion in major organs (liver and kidney) was

confirmed by genomic PCR (Supplementary Fig. 8B) as well as immunohistochemistry analysis of Ire1α-induced Xbp1s expression following tunicamycin treatment (Tm, a glycosylation inhibitor to induce ER stress and activation of the Ire1α-Xbp1s pathway in mouse tissue[30–32]; Supplementary Fig. 8A). Intriguingly, we also detected elevated expression of Gclc at both mRNA and protein levels in Ire1α-null mouse kidneys compared to control (Ire1α wild-type) kidneys (Supplementary Fig. 8C, D), confirming that the previously identified Ire1α-Gclc regulatory circuit is functional in this mouse model. We then subjected the control and Ire1α-null mice to an established renal IRI protocol[26,27]. Compared to control mice, Ire1α-null mice showed decreased levels of renal damage markers including blood urea nitrogen (BUN) and creatinine, which are typically elevated after renal IRI (Fig. 6A, B). Histologically, when compared to control kidneys, Ire1α-null kidneys had significantly reduced tubular damage quantified by tubular dilation, brush border loss, flattened epithelial cells or sloughing of cells and tubular cast formation after IRI (Fig. 6C). Immunohistochemical staining of the lipid peroxidation marker 4-hydroxy-2-noneal (4-HNE) confirmed that when compared to control kidneys, Ire1α-null kidneys had significantly reduced 4-HNE staining induced by IRI (Fig. 6D).

To assess whether pharmacological inhibition of Ire1α can also alleviate IRI, we treated wild-type C57BL6 mice with vehicle,

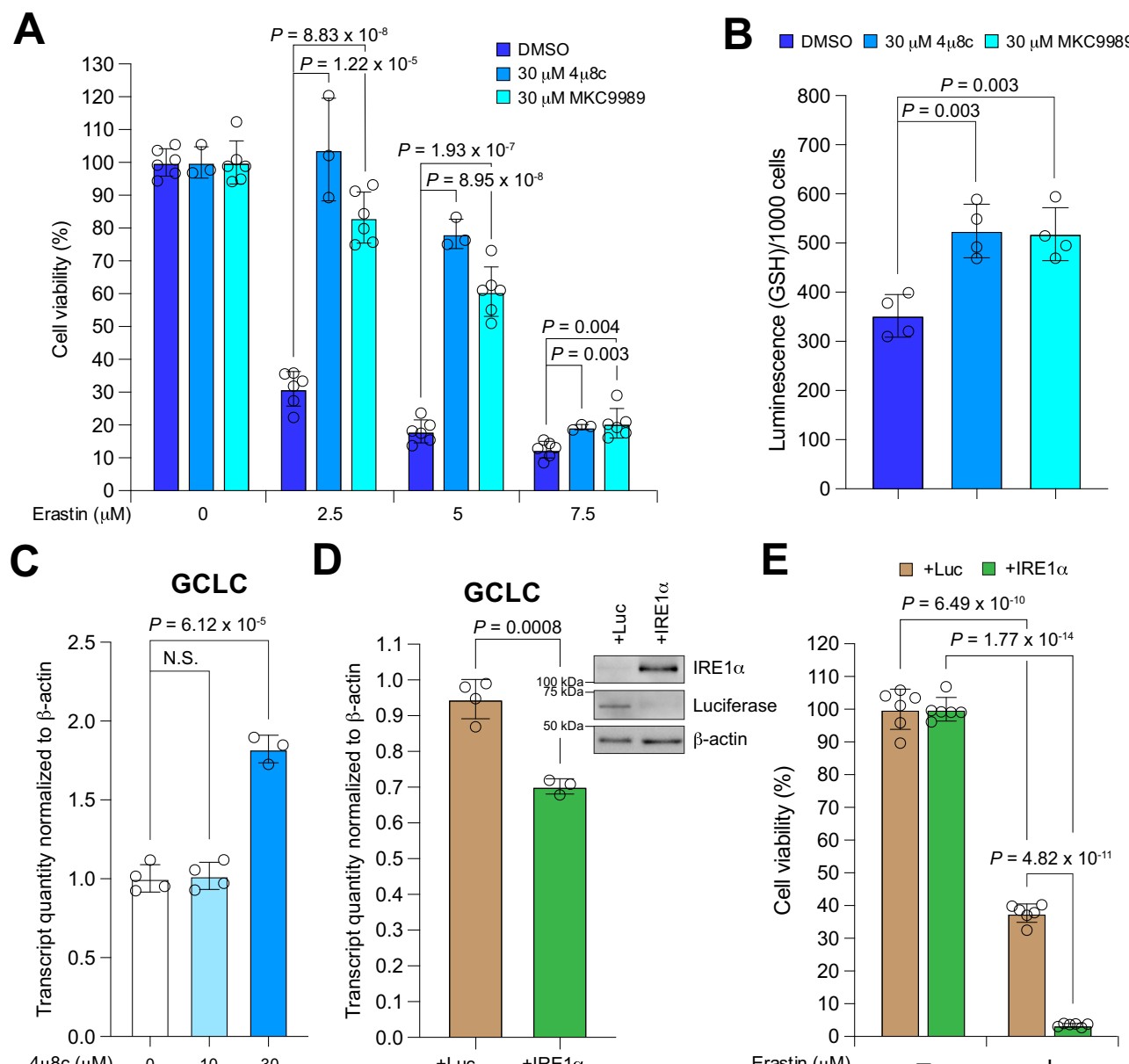

**Fig. 5 | Pharmacological IRE1α inhibition promotes ferroptosis resistance.**
**A** Viabilities of MDA-MB-231 cells co-treated with erastin and two widely used small molecule IRE1α inhibitors (4μ8c and MKC9989). 24 h after starting drug treatment, cell viability was measured by CCK8 assay. Values were normalized to treatment with 0 μM erastin (DMSO) which is set to be 100% viable. **B** Measurement of cellular GSH levels in MDA-MB-231 cells co-treated with DMSO or IRE1α inhibitor with 2.5 μM erastin for 12 h. The luminescence readings were normalized to cell numbers (/1000 cells). **C** Quantitative real-time PCR analysis of GCLC expression in MDA-MB-231 cells treated with different concentrations of 4μ8c for 7 h. β-actin was used as a normalization control. **D** Quantitative real-time PCR analysis of GCLC mRNA level in MDA-MB-231 cells overexpressing luciferase control (Luc) or IRE1α. β-actin was used as normalization control. Insert (top right corner): Western blotting analysis of IRE1α expression in these cells. Luciferase (Luc) was used as an overexpression control and β-actin as a loading control. **E** Viabilities of MDA-MB-231 cells over-expressing Luc or IRE1α with 2.5 μM erastin treatment. 24 h after starting erastin treatment, cell viability was measured by CCK8 assay. Viability is set as 100% with 0 μM erastin (DMSO). Data are presented as mean ± s.d. in all panels with *n* = 6 in (**A**) and (**E**) except *n* = 3 for the 30 μM 4μ8c group in (**A**), and *n* = 4 in (**B**, **C**, and **D**) except *n* = 3 for the 30 μM 4μ8c group in (**C**) and the +IRE1α group in (**D**). *n* indicates independent repeats. Unpaired, two-tailed Student's *t* tests were performed to calculate the *P* values for all the statistical analyses. N.S. not significant. Source data are provided as a Source Data file.

4μ8c, or liproxstatin-1 (a ferroptosis inhibitor used as a positive control for reducing IRI Ref. 2) then subjected the mice to the same renal IRI protocol. As expected, liproxstatin-1 significantly decreased BUN and creatinine levels (Fig. 7A, B), reduced tubular damage (Fig. 7C) and 4-HNE staining after IRI (Fig. 7D). To determine the optimal in vivo dose of 4μ8c for our study, based on previously reported doses[33–37], we treated the mice with 20 or 40 mg/kg 4μ8c and found that although 20 mg/kg 4μ8c efficiently inhibited Tm-induced Xbp1s expression, 40 mg/kg was required to increase Gclc expression level in the kidney (Supplementary Fig. 8F, G). Consistent with this finding, 40 mg/kg 4μ8c significantly reduced BUN and creatinine levels after IRI (Fig. 7A, B). Comparable to liproxstatin-1, 40 mg/kg 4μ8c treatment also significantly reduced tubular damage (Fig. 7C) and 4-HNE staining induced by IRI (Fig. 7D). Taken together, these results demonstrate that pharmacological Ire1α inhibition suppresses lipid peroxidation induced by renal IRI in vivo, and through this mechanism alleviates renal damages caused by IRI.

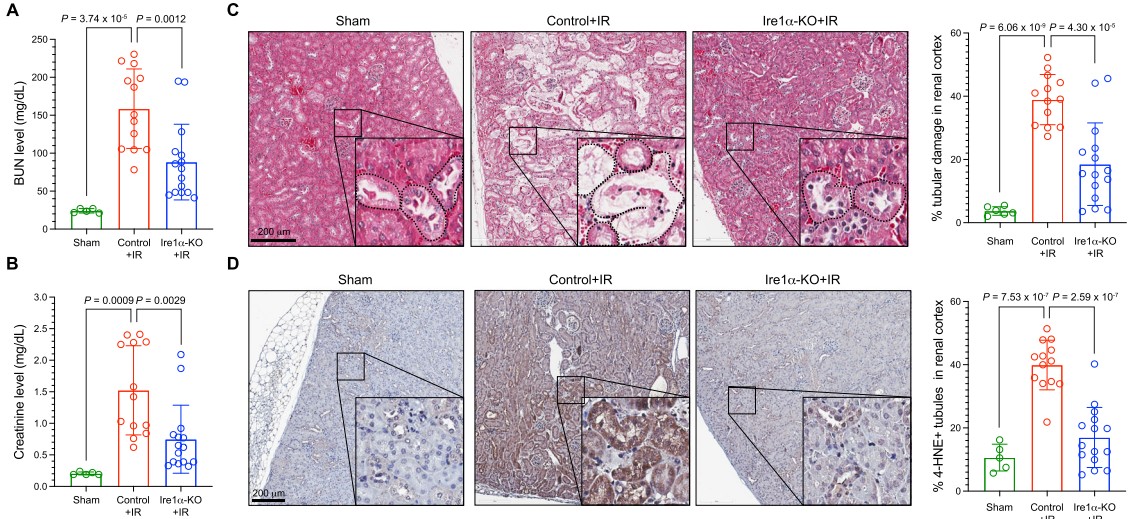

**Fig. 6 | Ire1α deficiency reduces ischemic kidney damage.** Quantification of (**A**) blood urea nitrogen (BUN) and (**B**) creatinine levels in sham-treated and control (*Ern1^{f/f}* + tamoxifen) as well as Ire1α-KO (*Ern1^{f/f};Rosa26-CreERT2^+* + tamoxifen) mice subjected to ischemia–reperfusion (IR). **C** Representative microscopic images of haematoxylin and eosin (H&E) staining of the renal cortex after treatment. Selected representative regions are enlarged and displayed in the corner. Dash lines outline damaged renal tubules (as characterized by tubular dilatation, tubule brush border loss, flattened epithelial cells or sloughing of cells). Scale bars: 200 µm. The histogram on the right shows the quantification of renal tubule damages.

**D** Representative images of lipid peroxidation marker 4-HNE immunohistochemistry staining of renal cortex after IR treatment. Selected representative regions are enlarged and displayed at the corner. Scale bars: 200 µm. The histogram on the right shows the quantification of 4-HNE-positive tubules in the renal cortex. Data are presented as mean ± s.d. in (**A**, **B**) and the right panels of (**C**, **D**), with n = 5 (sham-treated), n = 13 (control + IR) and n = 15 (Ire1α-KO + IR) independent kidneys in the corresponding groups. *P* values were calculated from unpaired two-tailed Student's *t* test. Source data are provided as a Source Data file.

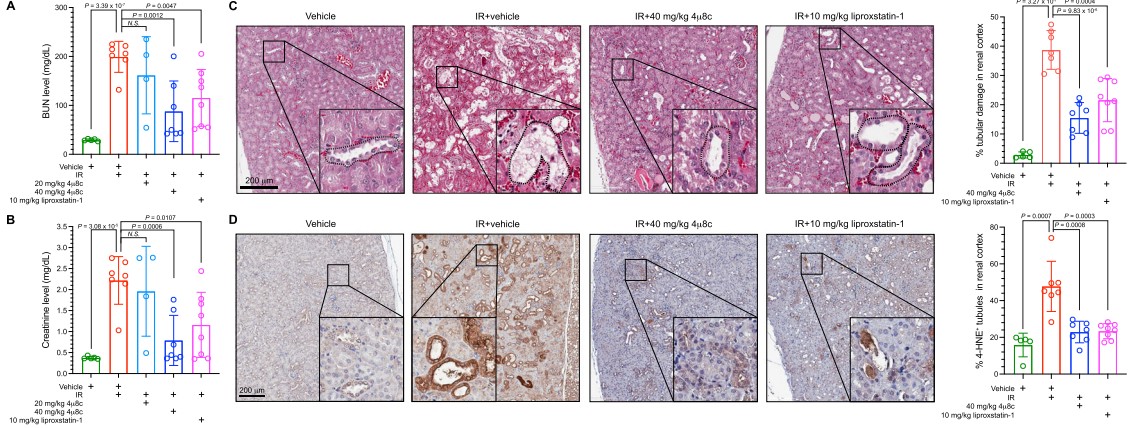

**Fig. 7 | Pharmacological Ire1α inhibition reduces ischemic kidney damage.** Quantification of (**A**) blood urea nitrogen (BUN) and (**B**) creatinine levels in mice treated with either vehicle, 4µ8c (20 mg/kg or 40 mg/kg), or liproxstatin-1 (10 mg/kg) then subjected to ischemia–reperfusion (IR). **C** Representative microscopic images of haematoxylin and eosin (H&E) staining of the renal cortex after treatment. Selected representative regions are enlarged and displayed at the corner. Dash lines outline damaged renal tubules (as characterized by tubular dilatation, tubule brush border loss, flattened epithelial cells or sloughing of cells). Scale bars: 200 µm. The histogram on the right shows the quantification of renal tubule

damages. **D**. Representative images of lipid peroxidation marker 4-HNE immunohistochemistry staining of renal cortex after treatment. Scale bars: 200 µm. The histogram on the right shows the quantification of 4-HNE-positive tubules in the renal cortex. Data are presented as mean ± s.d. in (**A**, **B**) and the right panels of (**C** and **D**), with n = 5 (vehicle only), n = 7 (IR + vehicle), n = 4 (IR + 20 mg/kg 4µ8c), n = 7 (IR + 40 mg/kg 4µ8c) and n = 8 (IR + 10 mg/kg liproxstatin-1) independent kidneys in the corresponding groups. *P* values were calculated from unpaired two-tailed Student's *t* test. Source data are provided as a Source Data file.

To further delineate the molecular basis of Ire1α inhibition in alleviating IRI, we confirmed that IRI did not significantly induce apoptosis and autophagy markers, consisting of cleaved caspase 3 and processing of LC3B, respectively, in mouse kidney (Supplementary Fig. 9E). In addition, in vitro hypoxia-reoxygenation (H/R) treatment did not induce apoptosis in HK-2 immortalized human proximal tubule epithelial cells (Supplementary Fig. 9B). These findings suggest that ferroptotic cell death is the major mechanism of cell death that contributes to IRI, which is consistent with previous reports[26,38,39].

Interestingly, a recent study identified that IRE1α knockdown in HK-2 cells led to attenuated ferroptosis induction following H/R and a less commonly used IRE1α inhibitor (Irestatin 9389) also alleviated IRI in mice[40]. Although these investigators proposed a mechanism based on JNK activation through IRE1α, we did not observe detectable JNK activation in HK-2 cells after H/R, possibly due to shorter reoxygenation time used (up to 72 h by the previous study[40] vs. 2 h by our study, Supplementary Fig. 9C). These discrepancies suggest that depending on the experimental conditions, there could be multiple mechanisms

involved in IRE1α-mediated sensitization to ferroptosis induction in this renal damage model. Moreover, various IRE1α inhibitors may have different off-target effects, depending on the experimental context. Interestingly, we found that both ferrostatin-1, a ferroptosis inhibitor[41] and 4μ8c reduced the endogenous lipid peroxidation level in HK-2 cells and inhibition of IRE1α expression had similar but intermediate effects (Supplementary Fig. 9A, D), possibly due to the intermediate level of IRE1α expression knockdown by shRNA (Supplementary Fig. 9A). In contrast to the in vivo IRI studies that showed non-detectable effect on autophagy in mouse kidney (Supplementary Fig. 9E), we detected increased processed LC3B (LC3B-I to LC3B-II) in HK-2 cells upon H/R and RSL3 treatment (Supplementary Fig. 9B). This may be due to the differences between cell death pathways induced in mouse kidney tissue in vivo and human kidney cells cultured in vitro, as well as in vivo IR protocol and in vitro H/R protocol/RSL3 treatment. However, IRE1α knockdown did not have a substantial effect on the induction of autophagy by H/R or RSL3 (Supplementary Fig. 9B), suggesting that IRE1α is not primarily involved in mediating autophagy under this context.

## Discussion

In summary, our findings uncover a previously unknown link between the ER resident and UPR sensor protein IRE1α and ferroptosis regulation independent of UPR activation and IRE1α's canonical downstream effector XBP1. Previous studies reported that determination of cellular sensitivity to ferroptosis is mainly confined to the plasma membrane (where lipid peroxidation and FSP1-mediated detoxification occur during ferroptosis), the cytosol (where the GSH-GPX4 axis resides) and the mitochondria (where the DHODH-mediated anti-ferroptotic pathway is located). Although the ER is a cell organelle critical for secretory and membrane protein homeostasis as well as a sensor for reconciling various cellular stresses, such as nutrient deficiency, pH disturbance and lack of oxygen, its role in ferroptosis is largely unknown. A previous study showed that erastin activates the eIF2α-ATF4 pathway which intersects with the PERK-eIF2α-ATF4 branch of the UPR, possibly through cellular cystine/cysteine deficiency caused by erastin-mediated system $x_c^-$ (SLC7A11) inhibition[9]. A recent study[42] identified the endoplasmic reticulum (ER) membrane as an important site of lipid peroxidation and several previous studies suggested activation of the UPR in ferroptosis[9,43–45]. However, these studies only utilized the activation of ATF4 and its downstream target genes as markers of UPR activation. In addition to activation by PERK as part of the UPR, eIF2α-ATF4 can also be activated by other stress sensors, including GCN2, PKR and HRI, collectively referred to as the Integrated Stress Response (ISR)[46]. Deprivation of cellular amino acid cystine/cysteine is a direct consequence of SLC7A11 inhibition by erastin, which activates the GCN2-eIF2α-ATF4 signaling axis. Supporting this hypothesis, we observed the activation of GCN2, a key regulator of amino acid deficiency, but not PERK upon erastin treatment or cystine starvation in both MDA-MB-231 and Panc-1 cells (Fig. 2A; Supplementary Fig. 3A). This is also consistent with earlier studies showing that cystine starvation activates GCN2 rather than PERK[47] and the lack of enhanced splicing of the XBP1 mRNA upon erastin treatment[9]. Additionally, we performed widely accepted UPR assays to show that none of the UPR sensors, including IRE1α, PERK, and ATF6, were activated by erastin or cystine starvation in both cell types (Fig. 2A). XBP1 splicing was not induced under these conditions, either (Fig. 2A). These results suggest that in our experimental systems these two ferroptosis-inducing agents do not activate the UPR. Although both the PERK-eIF2α-ATF4 and ATF6 branches of the UPR have been implicated in the regulation of ferroptosis with diverse cellular context and inducing agents[48–60], we found that when the expression of the key transcription factors of the three UPR branches, XBP1, ATF4 and ATF6 was inhibited individually in MDA-MB-231 cells, ferroptosis induction by erastin treatment was not affected (Fig. 2B–D). Therefore, IRE1α's RIDD activity in mediating

cellular sensitivity to ferroptosis is unique to IRE1α and functions as an important regulatory component of ferroptosis.

Our study demonstrates that the UPR sensor protein IRE1α is a critical factor in determining cellular sensitivity to ferroptosis through regulating GSH availability (Fig. 8). The mRNA transcripts of the GCLC subunit of glutamate cysteine ligase, which is the rate-limiting enzyme in GSH biosynthesis, as well as SLC7A11 which encodes the cystine/glutamate antiporter xCT, are previously unidentified targets for the RIDD activity of IRE1α. We further demonstrated that IRE1α deficiency led to resistance to RSL3, a GPX4 inhibitor that induces ferroptosis (Supplementary Fig. 1E, F)[11]. In these experiments, we used a concentration of RSL3 (1 μM) that does not completely inhibit GPX4 (Supplementary Fig. 5A, as concentrations of RSL3 higher than 1 μM further induced cell death), and under these conditions, IRE1α deficiency would increase cystine uptake and GSH level through SLC7A11 and GCLC upregulation, mitigating the effects of GPX4 inhibition by RSL3. Consistent with this mechanism, in the CTRP database (https://portals.broadinstitute.org/ctrp.v2.1/), both SLC7A11 and GCLC are present within the group of genes highly correlated with resistance to RSL3 (Supplementary Fig. 5B). We further demonstrated that in both MDA-MB-231 and Panc-1 cells, SLC7A11 overexpression promoted resistance to RSL3-induced cell death (Supplementary Fig. 5C, D, G), while attenuated SLC7A11 expression further sensitized these cells to RSL3 (Supplementary Fig. 5E, F, G). These results clearly support that the activity and expression level of the upstream regulator (SLC7A11) still regulate the effects of the downstream GPX4 inhibition by RSL3. Therefore, IRE1α, through its negative regulation on both SLC7A11 and GCLC, determines cellular sensitivity to RSL3.

RIDD targets the mRNA of both ER-localized/secretory proteins and cytosolic proteins and certain pre-miRNAs and miRNAs[61–63]. Although previous studies identified that RIDD functions mainly in the clearance of ER-bound transcripts as a mechanism to reduce the load of ER client proteins during ER stress, more recent studies have also revealed an active role of RIDD in determining cell fate[61]. As discussed previously, RIDD represents a highly conserved function of IRE1α in cellular homeostasis[64]. Consistent with this concept, we also demonstrated that IRE1α-mediated GCLC mRNA cleavage is conserved in *C. elegans* and that *ire-1* mutant worms were resistant to DGLA (dietary PUFA). Although in our study, feeding with DGLA (0.3 mM) induced ferroptosis and shortened the lifespan in worms, a previous study found that exposure to lower level of DGLA (10 μM) induced autophagy and increased worm lifespan[65].

In addition to RIDD, IRE1α also induces non-conventional splicing and activation of XBP1, a major transcription factor regulating ER homeostasis[10,66–69]. Our results indicate that IRE1α-regulated ferroptosis sensitivity is independent of XBP1 activation, as forced expression of activated XBP1 in IRE1α-null cells failed to restore sensitization to ferroptosis (Fig. 1G; Supplementary Fig. 1B). Moreover, inhibition of XBP1 expression did not alter the sensitivity to erastin-induced ferroptosis (Fig. 2B). Nevertheless, XBP1 has been shown to promote antioxidant activities[70–72] which are critical components of the cellular defense mechanism against ferroptosis.

As emerging studies have revealed that ferroptosis serves as a tumor suppression mechanism[73–81], a promising therapeutic strategy is to activate ferroptosis pharmacologically during cancer treatment. We found that ectopically expressed IRE1α sensitized tumor cells to FIN-induced ferroptosis (Fig. 5E), which suggests that pharmacological activation of the RIDD activity of IRE1α may sensitize tumor cells to ferroptosis when combined with other ferroptosis-inducing agents, including FINs, radiation therapy and immunotherapy[5,82–86]. Our study revealed a mechanism of ferroptosis regulation that is common to both normal and cancer cells. The differential ferroptosis susceptibility or resistance of cancer versus normal cells will continue to be an area of active investigation.

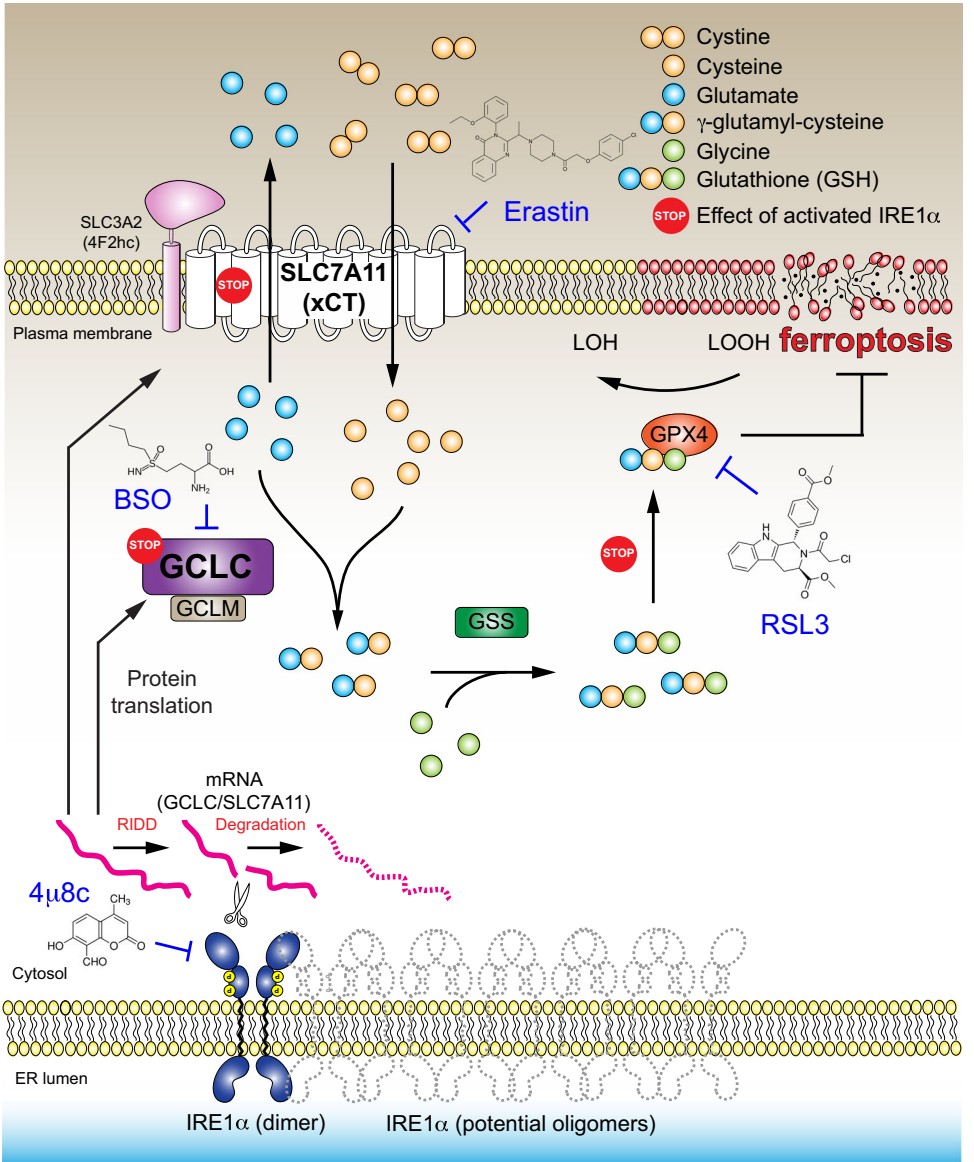

**Fig. 8 | A working model of how IRE1α regulates cellular glutathione synthesis and sensitivity to ferroptosis.** GCLC Glutamate-Cysteine Ligase Catalytic Subunit, GCLM Glutamate-Cysteine Ligase Modifier Subunit, GPX4 Glutathione Peroxidase 4, GSS glutathione synthetase, LOOH lipid hydroxide, LOH alcohol, BSO buthionine sulfoximine. SLC3A2 Solute Carrier Family 3 Member 2. SLC7A11 Solute Carrier Family 7 Member 11.

In conclusion, our study revealed a previously unidentified function of the ER stress sensor protein IRE1α in determining cellular sensitivity to ferroptosis through regulating GSH availability, a finding that may lead to the development of treatment strategies for ferroptosis-associated normal tissue damage and improvement in the efficacy of existing cancer therapies.

## Methods

This research complies with all relevant ethical regulations of the University of Texas MD Anderson Cancer Center. All animal studies were performed in accordance with a protocol approved by the Institutional Animal Care and Use Committee.

### Cell culture studies

The MDA-MB-231 and HK-2 cell lines were purchased from the American Type Culture Collection. The Panc-1 cell line was purchased from Sigma Aldrich. The *Ire1*[+/+] and *Ire1*[−/−] MEFs were obtained from Dr. Fumihiko Urano at Washington University in St. Louis. All cell lines were free of mycoplasma at the time of the assay tested with the MycoAlert Mycoplasma Detection Kit (Lonza). No cell line used in this study has been found in the International Cell Line Authentication Committee (ICLAC) database of commonly misidentified cell lines (version 11). Cell lines except HK-2 were cultured in Dulbecco's modified Eagle's medium (DMEM) with 10% (volume/volume; v/v) heat-inactivated FBS (GE HyClone) and 1% (v/v) penicillin/streptomycin (Gibco) at 37 °C and 5% CO₂. HK-2 cell were cultured in Keratinocyte SFM (serum-free medium, Gibco), supplemented with Bovine Pituitary Extract and EGF (Thermo Fisher Scientific). All cell lines were maintained in 10 cm tissue culture plates (Corning Falcon) and sub-cultured using 0.025% trypsin-EDTA (Gibco). Cells were treated with ferroptosis inducers including erastin (SelleckChem) and RSL3 (Sigma Aldrich), ferroptosis inhibitor ferrostatin-1 (Sigma Aldrich), γ-glutamylcysteine synthetase inhibitor L-Buthionine-(S,R)-sulfoximine, (BSO, Sigma Aldrich) and IRE1α inhibitors 4μ8c (Sigma Aldrich) and MKC9989 (MedChemExpress). For cystine starvation, DMEM lacking glutamine/methionine/cystine (Thermo Fisher Scientific) with glutamine (Thermo Fisher Scientific) and methionine (Sigma Aldrich) supplement was used as a base cystine-deficient culture medium. 200 μM

cystine (Sigma Aldrich) was added to the base medium to form the full complement medium. For hypoxia-reoxygenation (H/R) treatment, HK-2 cells growing in 35 mm tissue culture dishes (Sarstedt Inc.) or 6-well plates (Corning Falcon) were kept in 1% $O_2$ for 12 h in a Baker Ruskinn InvivO$_2$ 500 hypoxia chamber, then in 21% $O_2$ for 2 h.

### Plasmid constructs

Lentiviral human IRE1α expression plasmid was constructed by inserting the protein-coding sequence (CDS) of human *ERN1* gene into the pLEX-MCS vector (Thermo Scientific Open Biosystems) between the SpeI and XhoI restriction sites (pLEX-hIRE1α). The K907A mutant IRE1α expression plasmid was generated through site-directed mutagenesis using the pLEX-hIRE1α plasmid (pLEX-hIRE1α-K907A). Control expression plasmids were constructed by inserting the protein CDS of firefly luciferase or enhanced green fluorescent protein (EGFP) into the pLEX-MCS vector (pLEX-luciferase and pLEX-GFP). Human XBP1s and XBP1u expression plasmids were constructed by inserting the CDS of either spliced XBP1 (1131 bp) or 3X human influenza hemagglutinin (HA)-tagged unspliced XBP1 (786 bp) between the SpeI and XhoI restriction sites in pLEX-MCS (pLEX-hXBP1s and pLEX-3HA-hXBP1u). To prevent further splicing of the pLEX-3HA-XBP1u transcript by IRE1α which reduces XBP1u protein expression, both splicing sites were mutated (1st: AGT<u>CCG</u>^CA to AG<u>AGC</u>GCA; 2nd: CTCT<u>G</u>^CAG to CTCT<u>CC</u>AG) without changing the amino acid sequence. The SLC7A11-myc overexpression lentiviral vector as well as the lentiCRISPR-sgSLC7A11 vectors were constructed as described previously[87]. The human GCLC expression plasmid was constructed using the Gateway Cloning system (Thermo Fisher Scientific) and the pCMV-SPORT6-GCLC vector containing a human GCLC cDNA clone (Horizon Discovery Catalog #: MHS6278-202759380, Clone ID: 6052162, NCBI GeneBank Accession: BC039894), the Gateway™ pDONR™221 Vector (Thermo Fisher Scientific), and the pLenti-CMV-Neo-DEST (Addgene Plasmid #17392) destination vector. The Tet-pLKO-puro plasmid was purchased from Addgene (Plasmid #21915) and the inducible shRNA constructs were generated according to the protocol provided by the depositor. DNA oligonucleotide sequences synthesized and ligated into the Tet-pLKO-puro backbone are summarized in Supplementary Table 1. All restriction enzymes used in the study were purchased from New England Biolabs. The pGEM-T plasmid (Promega) was used to generate DNA templates for in vitro transcription. PCR primers used to amplify DNA fragments to be inserted into pGEM-T are listed in Supplementary Table 2.

### CRISPR−Cas9-mediated gene knockout

Knockout of *ERN1* (gene name of IRE1α) in human cell lines was performed using single guide-RNAs (sgRNAs) based on the pSpCas9(BB)-2A-Puro (PX459) (Addgene Plasmid #48139) vector developed previously[88]. Complementary single-stranded DNA oligonucleotides encoding the sgRNAs were synthesized (Sigma Aldrich Genosys) and annealed, then ligated into *Bbs*I-digested PX459 vector. The sequences of the sgRNAs used in this study are listed in Supplementary Table 1. Transient transfection of the PX459-sgRNA into the target cancer cell lines was performed using the Lipofectamine 3000 Transfection Reagent (Invitrogen) in 6-well plates (Corning Falcon) followed by puromycin selection (Gibco, 1–5 μg/ml) for 3 days. The remaining cells were replated into 15-cm plates (Corning Falcon) to allow single colony formation. After 1–2 weeks, formed single colonies were isolated and expanded. Knockout of *ERN1* was verified by Western blotting and Sanger sequencing.

### Stable cell line generation through viral transduction

Cell lines stably overexpressing target genes were generated as reported previously[89]. Briefly, HEK293T packaging cells were transfected with control (firefly luciferase, GFP or shGFP) or target gene-expressing pLEX and pLenti constructs or pLKO shRNA constructs, together with pCMV-VSVG and pCMV-Δ8.2 lentiviral packaging plasmids. 24 and 48 h later, viral supernatant was collected and filtered through a 0.45 μm filter. To infect the target cell lines, 0.8 μg/ml polybrene (Sigma Aldrich) was added to the viral supernatant before adding to the target cells. 24 h after starting infection, 1-5 μg/ml puromycin (Gibco) or 1 mg/ml G418 (Roche) was added to the cells for antibiotic selection. Cells transduced with inducible shRNA (Tet-pLKO-puro) were treated with 0.5 μg/ml doxycycline (Sigma Aldrich) for 24 h to induce shRNA expression.

### Cell viability assay

Cell viability was measured using Cell Counting Kit-8 (CCK-8, APExBio) as previously described[90,91]. Briefly, cells were plated onto 96-well plates (Corning Falcon) at a density of $1 \times 10^4$ cells in 80 μl medium per well. 24 h later, 20 μl medium with compounds at 5X of the desired concentration was added to the well for 24 h. For cystine starvation, the original medium in each well was discarded and the well was washed with PBS, then 100 μl fresh medium depleted of cystine was added to the well. After 24 h of drug treatment or cystine starvation, 10 μl of CCK-8 reagent was added to each well containing 100 μl medium in total and the plate was incubated at 37 °C, 5% $CO_2$ for 3 h. The absorbance at 450 nm was collected using a BioTek Cytation 5 imaging plate reader running Gen5 3.13 software.

### Real-time cell imaging-based cell death assay

Cell death was quantitated with the IncuCyte Cytotox Red reagent and the IncuCyte S3 Live-Cell Analysis System running software version 2019B Rev2. Briefly, cells were plated onto 6-well plates (Corning Falcon) at a density of $3 \times 10^5$ cells in 2 ml medium per well. 24 h later, the medium in each well was replaced with 2 ml fresh medium with compounds for treatment at the desired concentrations as well as 250 nM Cytotox Red reagent. The plate was incubated in Incucyte S3 with fixed image acquisition intervals for both the phase and red channels using a 10× objective. 36 images per well were acquired at each timepoint. Cell death was quantitated as Red Object Count (per image) normalized to confluence (per image).

### Lipid peroxidation measurement

Cells were plated at a density of $3 \times 10^5$ cells in 2 ml medium per well in six-well plates (Corning Falcon). The next day, the medium in each well was replaced with 2 ml fresh medium with compounds for treatment. 18 h later, the cells in each well were washed with PBS once, then incubated with 1 ml PBS containing 2 μM C11-BODIPY 581/591 (Invitrogen) at 37 °C for 15 min. After a brief PBS rinse, the cells were trypsinized and resuspended in 500 μl PBS and analyzed in an Attune NxT flow cytometer (Thermo Fisher Scientific) with the BL1 detector using the 488 nm laser line. A minimum of 10,000 single cells were analyzed per sample.

### Cellular glutathione and glutamate measurement

Cellular glutathione and glutamate levels were measured using the GSH-Glo™ Glutathione Assay and Glutamate-Glo™ Assay (Promega), respectively. Briefly, 8000 cells per well were plated into 96-well plates (Corning Falcon) and allowed to attach to the plate overnight. Compounds for treatment were added to the cells the second day. After 12 h, one plate was subjected to the GSH-Glo™ Glutathione Assay or Glutamate-Glo™ Assay according to the manufacture's protocol and a duplicate plate for cell counts. Luminescence readings were collected using a BioTek Cytation 5 imaging plate reader. Raw glutathione/glutamate measurements were normalized to the cell counts (/1000 cells) from the duplicate plate using a Bio-Rad TC20 Automated Cell Counter.

## Cystine uptake assay

Cystine uptake was measured as described previously[92]. Briefly, $4 \times 10^4$ cells were plated in 12-well plates (Corning Falcon). On the next day, the medium was replaced with fresh DMEM (which contains 5 μM cystine) containing [$^{14}$C] cystine (0.04 μCi, PerkinElmer) with/without 1 μM erastin, and cells were incubated for the indicated time periods. Then the cystine uptake was terminated by rinsing cells with cold PBS once and lysed in 200 μl 0.1 mM NaOH solution. Radioactivity (DPM) was measured using Tri-Carb® Liquid Scintillation Analyzer (PerkinElmer, Model 4810TR) with quench curve. All experiments were carried out in triplicates.

## In vitro RNA cleavage assay

In vitro RNA cleavage assay was performed as described previously[93]. Briefly, RNA substrates were generated by in vitro transcription using the MEGAscript T7 Transcription or MAXIscript SP6/T7 Transcription Kit (Invitrogen) with DNA templates produced through either linearizing plasmids containing the mRNA CDS following an SP6/T7 promoter or annealing chemically synthesized complementary single-stranded DNA oligonucleotides which contain a T7 promoter sequence (Sigma Aldrich Genosys). List of plasmids and oligonucleotide sequences used is listed in Supplementary Table 2. After in vitro transcription, RNA was purified with acidic phenol/chloroform extraction (Invitrogen) and sodium acetate precipitation followed by ethanol wash. RNAs <1 kb were further purified using urea-PAGE gel to isolate the correct in vitro transcription products according to a previously published protocol[94], followed by additional acidic phenol/chloroform extraction. For IRE1α cleavage assay, purified RNA substrates were first incubated with inhibitor or vehicle (DMSO) for 30 min on ice, then incubated with recombinant GST-tagged human IRE1α (aa 465-977 corresponding to the kinase/ribonuclease domain, 11905-H20B, Sino Biological) in reaction buffer (40 mM HEPES pH 7.0, 10 mM Mg(OAc)$_2$, 50 mM KOAc, 5 mM DTT, 1 μM ADP) for 30 minutes at 30 °C. At the end of the reaction, 1 volume of 2X loading buffer (for urea-PAGE gel: #161-0768, Bio-Rad; for formaldehyde-agarose gel: #351-081-661, Quality Biological) was added to the mixture to stop the reaction then the whole sample was heated for 10 minutes at 70 °C or 5 min at 80 °C and chilled on ice for 2 min. The denatured samples were then separated in 13–14% urea-PAGE gel or 1–1.5% formaldehyde-agarose gel. The gels were stained with GelRed Nucleic Acid Gel Stain (Biotium) and visualized on a ChemiDoc MP Imaging System (Bio-Rad). RNA secondary structure prediction was performed using RNAfold WebServer (http://rna.tbi.univie.ac.at/cgi-bin/RNAWebSuite/RNAfold.cgi) with default parameters.

## Quantitative PCR with reverse transcription

Real-Time Quantitative Reverse Transcription PCR (qRT-PCR) was performed as previously described[89,95,96]. In brief, total RNA was extracted using TRIzol Reagent (Invitrogen) according to manufacturer's protocol. Total RNA concentration was determined using a NanoDrop One$^C$ spectrophotometer (Thermo Fisher Scientific). cDNA was synthesized using the High-Capacity cDNA Reverse Transcription Kit (Applied Biosystems) with random priming. qRT-PCR was performed on a QuantStudio 5 system (Applied Biosystems, software v1.5.1) with PowerUp SYBR Green Master Mix (Applied Biosystems). Results were computed relative to a standard curve made with cDNA pooled from all samples. Alternatively, the $2^{-\Delta\Delta Ct}$ method was also used. Target gene quantities were normalized to β-actin. Primer sequences are listed in Supplementary Table 3.

## Immunoblotting

Western blotting to detect protein expression was performed as previously described[97–99]. In brief, cells growing on tissue culture vessels were harvested with cell scraper on ice then lysed with RIPA Lysis Buffer (Millipore) supplemented with Halt Protease and Phosphatase

Inhibitor Cocktail (Thermo Scientific) at 4 °C for 1 h. For ATF6 immunoblotting the SDS sample buffer recipe and protocol provided by the antibody manufacturer (CosmoBio, BAM-73-500-EX) were used. Protein lysates were then cleared in a refrigerated table-top microcentrifuge (Eppendorf) at 21,100 g for 10 min. Cell fractionation was performed using the Subcellular Protein Fractionation Kit (Thermo Scientific). Protein concentration was determined with Pierce BCA Protein Assay (Thermo Scientific) using a NanoDrop One$^C$ spectrophotometer (Thermo Fisher Scientific). Phos-tag$^{TM}$ Western Blotting was performed according to manufacturer's protocol (Fujifilm Wako Chemicals) with RIPA Lysis Buffer without EDTA for harvesting whole-cell protein lysates. The following gel composition and running conditions were used for detecting (1) phospho-IRE1α: 5% Phos-tag gel with 25 μM Phos-bind acrylamide, 60 V for 30 min followed by 100 V for 3 h; (2) phospho-PERK: 5% Phos-tag gel with 3.5 μM Phos-bind acrylamide, 15 mA for 30 min followed by 5 mA for 9.5 h. 40–60 μg of protein per sample was electrophoresed and transferred onto nitrocellulose membrane (Bio-Rad) for immunoblotting using antibodies against IRE1α (3294, Cell Signaling), XBP1 (ab220783, Abcam), PERK (3192, Cell Signaling), phospho-GCN2 (ab75836, Abcam), GCN2 (3302, Cell Signaling), phospho-eIF2α (9721, Cell Signaling), eIF2α (2103, Cell Signaling), ATF4 (11815, Cell Signaling), ATF6 (CosmoBio, BAM-73-500-EX), GCLC (sc-390811, Santa Cruz Biotechnology), GCLM (sc-22754, Santa Cruz Biotechnology), GPX4 (MAB5457, R&D Systems), SLC7A11 (12691, Cell Signaling), ACSL4 (SAB2701949, Sigma Aldrich), phosphor-JNK (9251, Cell Signaling), JNK (9252, Cell Signaling), LC3B (NB600-1384, Novus), Caspase-3 (9662, Cell Signaling), luciferase (NB600-307, Novus Biologicals), calreticulin (ab2907, Abcam), HSP90 (sc-13119, Santa Cruz Biotechnology), GAPDH (8884, Cell Signaling) and β-actin (sc-1615, Santa Cruz Biotechnology) all with 1:1000 dilution in 3% (w/v) BSA (Sigma Aldrich). HRP-conjugated secondary antibodies (rabbit IgG: 111035144, mouse IgG: 115035003, 1:3000 dilution in 3% BSA, Jackson ImmunoResearch) and the ECL-2 (Thermo Scientific) or ECL (GE Amersham) chemiluminescent reagent were used to visualize the bands using a Bio-Rad ChemiDoc imaging system running Image Lab V4.0 software.

## Histology and immunohistochemistry

Fresh mouse kidneys were fixed in 10% neutral buffered formalin for 24 h, washed once with PBS, then stored in 70% ethanol at 4 °C until further processing. The fixed tissues were dehydrated and embedded in paraffin, then sectioned at 5 μm thickness. Tissue sections were subjected to haematoxylin and eosin (H&E) staining using standard protocol. For immunohistochemistry (IHC) staining, after deparaffinization and rehydration, tissue sections were subjected to endogenous peroxidase blocking with 3% H$_2$O$_2$ in PBS for 12 min at room temperature (RT) and permeabilization with 0.4% Triton X-100 for 5 min at RT. Then antigen retrieval was performed with 1X Target Retrieval Solution, pH 6.0 (Dako) at 125 °C for 4 min in a Retriever 2100 (Aptum Bio), followed with blocking using 10% goat serum in PBS for 60 min at RT. The sections were then incubated with primary antibody against 4-HNE (ab46545, Abcam, 1:250) or XBP1 (ab220783, Abcam, 1:100) at 4 °C overnight, then secondary antibody (For 4-HNE: Peroxidase AffiniPure Goat Anti-Rabbit IgG, 111035144, Jackson ImmunoResearch; For XBP1: ImmPRESS Horse Anti-Rabbit IgG Polymer Kit, Peroxidase, MP-7401, Vector Laboratories) for 30 min at RT. Haematoxylin (HT-109, Sigma Aldrich) was used for counterstaining. ImmPACT DAB Substrate Kit, Peroxidase (SK-4105, Vector Laboratories) was applied for 3–5 min to visualize the staining at the end. Mounted slides were scanned using an Aperio Digital Pathology Slide Scanners (Leica Biosystems) and a 20× objective.

## C. elegans study

*C. elegans* strains N2 [var Bristol] (wild-type), RB925 [ire-1(ok799) II] (*ire-1* deletion), and SJ30 [ire-1(zc14) II; zcIs4 V] (*ire-1* kinase domain

mutant) were obtained from the Caenorhabditis Genetics Center (CGC) at the University of Minnesota, which is funded by NIH Office of Research Infrastructure Programs (P40 OD010440). All strains were cultured and maintained at 20 °C on solid nematode growth medium (NGM) seeded with *E. coli* OP50 as a standard diet[100]. The fatty acid-supplemented media was previously described[22]. The sodium salt of dihomo-gamma-linolenic acid (DGLA, 20:3n-6, NuChek Prep, Inc. Cat. # S-1143) was dissolved in nuclease-free water and filter sterilized. 0.3 mM DGLA and 0.1% TERGITOL NP40 (Sigma Aldrich Cat. # NP40S) were formulated in cooled NGM media then poured into 6 cm petri dish. For non DGLA-treated group, only 0.1% NP40 was added to the cooled NGM media. *E. coli* OP50 food source was seeded onto the plates 3 days prior to plating the worms. For the lifespan assay, age-synchronized worms were obtained by hatching the eggs from gravid worms in 6 cm NGM petri plates and incubating at 20 °C until reaching the L4 stage. L4 stage worms were cultured with fresh medium plates exchanged daily. Worms were observed and recorded daily as alive, dead, or censored if bagged or missing until the death of the last worm.

### Kidney ischemia-reperfusion injury (IRI)

The induction of kidney IRI was performed with the dorsal approach and a bilateral renal pedicle clamping as described previously[27,101] approved by the Institutional Animal Care and Use Committee at the University of Texas MD Anderson Cancer Center (IACUC protocol # 00001769). Mice were housed with a 12 h light-12 h dark cycle. The ambient temperature was 70–74 °F, with 40–55% humidity and the mice had ad libitum access to water and food. All the mice were fed with PicoLab Rodent Diet 5053. *Ern1^{f/f}* mice as described previously[28] were provided by RIKEN and Kanazawa Medical University through a Material Transfer Agreement. *Rosa26-CreERT2* mice were obtained from the Jackson Laboratory (Strain #: 008463). All the mice used were of the C57BL/6 background from both genders. To induce Ire1α deficiency, 8-week old *Ern1^{f/f};Rosa26-CreERT2^+* mice were injected with 75 mg/kg tamoxifen (Sigma Aldrich) dissolved in corn oil (Sigma Aldrich) for 5 consecutive days and subjected to the IRI procedure after 2 weeks. Mice were anaesthetized using isoflurane through a Portable Anesthesia Machine (Patterson Vet Supply) and sustained-release buprenorphine (ZooPharm) as an analgesic was injected subcutaneously. During the surgical procedure, heating pads were used to maintain mouse body temperature and eye lubricant to keep moisture. Via the dorsal approach, the right kidney was resected and the left renal pedicle was clamped for 30 min using a vascular clamp (Fine Scientific Tools). The muscle layer and skin layer were closed using 4-0 VICRYL absorbable sterile surgical suture (Surgical Specialties) and skin stapler, respectively. Sham-operated mice received identical surgical procedures, except the renal pedicle clamping. All mice were sacrificed 24 h after reperfusion. In some experiments, 4μ8c (20 or 40 mg/kg, MedChemExpress) and liproxstatin-1 (10 mg/kg, Cayman Chemical), both given in 16% (v/v) Cremophor EL (Sigma Aldrich) saline solution, were intraperitoneally injected into mice daily for 3 days before IRI and the surgical procedure was performed on the day following the last drug injection.

### Statistics and reproducibility

Results from the cell culture and cell-free experiments were collected from at least three independent replicates. In the *C. elegans* survival study, 45–102 worms (42–49 in the replicate study) per group were used to assess the effect on lifespan change. In the kidney IRI study with 4μ8c and liproxstatin-1 treatment, 4–8 mice were randomly assigned to each treatment group. In the kidney IRI study using control and conditional Ire1α-KO mice, 4–8 mice were generated and used in each group. In the H&E and IHC analysis, 7–10 microscopic fields per genotype or treatment group were selected for quantification. In the IncuCyte real-time cell imaging analysis, 36 images per each well were

acquired at each time point for quantification. In the histograms, data are plotted as means ± standard deviation (s.d.). Statistical significance (*P* values) of pairwise comparisons were calculated using unpaired, two-tailed Student's *t* tests in Microsoft Excel (Version 16.80 for Mac) or GraphPad Prism 9.0. *P* values of survival analysis were calculated using Log-rank (Mantel-Cox) and Gehan-Breslow-Wilcoxon tests in GraphPad Prism 9.0.

### Reporting summary

Further information on research design is available in the Nature Portfolio Reporting Summary linked to this article.

## Data availability

The Cancer Therapeutics Response Portal (CTRP, V2.1) is a web-based publicly accessible database (https://portals.broadinstitute.org/ctrp.v2.1/). The data generated in this study are provided in the Supplementary Information/Source Data file. Source data are provided with this paper.

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

## Acknowledgements

This research was supported by the Cancer Prevention and Research Institute of Texas grant RP190192 (to A.C.K. and D.J.) and RP230072 (to B.G.), and the Olga Keith and Harry Carothers Wiess Distinguished University Chair in Cancer Research Endowment Fund (to A.C.K.), as well as National Institutes of Health grants R01CA181196, R01CA244144, R01CA247992, R01CA269646 (to B.G) and U54CA274220 (to B.G. and A.C.K.). We apologize to the researchers whose relevant work cannot be cited in this study due to space limitations.

## Author contributions

D.J. and A.C.K. conceived the project and D.J. designed the experiments. D.J. and Y.G. performed the majority of the experiments with assistance from T.W., L.W., and L.X. R.B. and Z.Y. generated the initial IRE1α-KO cell

lines used in the study. Y.Y. performed the cystine uptake assay. H.L. guided the mouse kidney IRI experiments. T.I. provided the Ern1f/f mice through a Material Transfer Agreement. D.J., B.G., and A.C.K. supervised the study and established collaborations. D.J. wrote the manuscript with significant input from A.C.K. and B.G. All of the authors read and provided edits to the manuscript.

## Competing interests

B.G. is an inventor with patent applications involving targeting ferroptosis in cancer therapy (patent no. PCT/US2022/018663). A.C.K. is a stockholder in Aravive, Inc. The remaining authors declare no competing interests.
