## [Peer Review File · Nature Communications]

IRE1 α determines ferroptosis sensitivity through regulation of glutathione synthesisReviewers' comments:

Reviewer #1 (Remarks to the Author):

In this study, the authors investigated the potential role of IRE1 α in regulating ferroptosis sensitivity. They claim that IRE1 α is a positive regulator of ferroptosis by inhibiting GSH production. Specifically, they showed that IRE1 α -mediated downregulation of GCLC and SLC7A11 contributes to GSH depletion in cancer cells. Finally, they found that IRE1 α inhibitors could prevent renal ischemia-reperfusion injury in mice.

1. Three ER-resident transmembrane signal transducers, PERK, ATF6, and IRE1, constitute the three canonical UPR signaling branches to manage ER stress and restore cellular homeostasis. Previous independent studies have shown that PERK and ATF6 regulates ferroptosis resistance. In contrast to these previous studies, when and how IRE1 plays a different role in promoting ferroptosis. What is the status of other UPR pathways in WT and IRE1 KO cells in response to ferroptosis stimuli? Is this cell type dependent? What are the checkpoints of these three UPR pathways during ferroptosis?

2. The authors show through some rescue experiments that both GCLC and SLC7A11 are mediators of IRE1-related phenotypes. They also claim that IRE1 plays a broad-spectrum role in promoting ferroptosis. However, this may not be true because different ferroptosis activators target different molecules downstream and upstream of GSH synthesis, and additionally have different effects on intracellular iron overload. For example, the authors found that the GCLC inhibitor buthionine sulfoximine (BSO) resensitized IRE1 α -null cells to erastin-induced ferroptosis. We know that BSO is also an inducer of ferroptosis. If IRE1-KO cells are resistant to RSL3 (targeting downstream of GSH depletion), then it is expected to be resistant to BSO as well.

3. The authors suggest that XBP1 is not required for IRE1 activity during ferroptosis. However, this experiment lacked western blot analysis of XBP1 isoform expression in the presence of erastin, RSL3 or BSO. Importantly, this study lacked a positive control, as XBP1 splicing is critical for cell survival under stressful conditions (e.g., apoptosis).

4. The authors used cancer cell lines to study the in vitro function of IRE1. However, the authors used an in vivo kidney I/R model to explore the protective effect of IRE1 inhibitors. There are some problems with this in vivo design. First, the I/R model is a hybrid cell death model. The authors need to examine whether IRE1 inhibitors or genetic deletion of IRE1 prolongs the survival of GPX4 KO animals in the kidney. Second, the authors need to investigate whether targeting the IRE1 pathway affects the anticancer activity of ferroptosis inducers in vivo. Importantly, knockdown of XBP1 is an important in vivo control to support their hypothesis that IRE1 regulates ferroptosis in an XBP1-independent manner.

5. It appeared that IRE1 KO prevented erastin-induced downregulation of GPX4 protein (Fig. S3). What is the mechanism?

6. Individual experimental conditions for this study varied widely, such as the use of different drug concentrations (e.g., 2.5, 5, and 10 μ M erastin in different figures). In addition, its core finding that the effect of IRE on GCLC and SLC7A11 expression is based on changes in the basal state. It is unclear whether IRE1-mediated downregulation of GCLC and SLC7A11 at baseline affects other ER stress sensitivities.

7. The author cites his own publication (ref10) in multiple places showing that ferroptosis inducers activate the UPR pathway. However, this is not the case and the cited literature only states that doxorubicin can activate the UPR response and does not refer to any ferroptosis studies. The accuracy of other references is also questionable.

8. The dose of IRE1 inhibitor used in this study was much higher than the IC50. For example, the IC50 of 4 μ 8c is 76 nM, while 30 μ M was used in this study. This will lead to off-target effects in the interpretation of the conclusions.

9. Finally, the authors' core finding that IRE1 α determines ferroptosis sensitivity through GSH modulation is not new (PMID: 35724508; PMID: 36091771; PMID: 34558645). Although the models may be different, the authors need to discuss these findings from other groups.

Reviewer #2 (Remarks to the Author):

Jiang et al report that modulating Ire1 activity can determine ferroptosis sensitivity in cells and worms. I find the main observation interesting, but I think additional experiments should be done to meet the bar for publication in Nature Comm.

One aspect that the manuscript does not address, but I think it should be addressed, is how Ire1 is being activated in the described experiments. One possibility (1) is that the FINs cause the accumulation of misfolded proteins in the ER lumen, especially since they cause Cys depletion in cells. Other possibility (2) is that the FINs cause lipid stress that is sensed directly by Ire1. The authors can use Ire1 mutations to distinguish if 1 or 2 in the cell studies. I think 2 would be more interesting than 1, but in that case, the authors should provide additional data to show how the lipid peroxides (or other agents of lipid stress) directly activate Ire1 in the ER membrane in these models. Or perhaps they can find a fraction of Ire1 in the plasma membrane or the inner mitochondrial membrane, the sites for accumulation for lipid peroxides ?

Other points:

- Data points should be added to Fig.1A - C and similar panels thereafter.

- Molecular weight markers are absent from 2G. Also they should indicate the localization of the stem loop motif in fragment #4 and the expected size of the fragments upon cleavage.

- The worm data is important and it should be included as a main figure, but smaller molecular weight markers should be added to SFig6D, so that the size of the fragments can be evaluated. Also, in this case

can the authors identify a stem loop motif as the site for cleavage ? If yes, they can also generate mutants that cannot be cleaved by Ire1.

- I would recommend that the authors increase the number of main figures (from 4 to 5-6) and reduce the number of Sup Figs.

Reviewer #3 (Remarks to the Author):

Ferroptosis is an important cell death mechanism. This study by Jiang et al. tried to determine the regulatory effect of IRE1 α on ferroptosis. The study is overall interesting. But there are several questions that need to be addressed.

1. In this study, the regulatory effect of IRE1 α on ferroptosis was only determined in vitro. The authors should also detect the regulatory effect of IRE1 α on ferroptosis and the related pathway in vivo.

2. It is puzzling why Ern1 was knocked out in mouse embryonic fibroblast (MEF), but not IRE1 α , or necessary introduction to Ern1 lacks in this paper.

3. In Fig. 4, the authors claim all the effects presented on ferroptosis. How can the authors exclude that the effects of IRE1 α on other death forms, such as autophagy and apoptosis. The authors need to present a systematic characterization of the different cell death pathways in their models.

4. In this study, the authors use the kidney ischemia model. But why were the renal cell lines not detected in vitro?

5. The authors need to demonstrate how to select the dosage of IRE1 α inhibitor? A dose-response should be demonstrated both in vitro and in vivo. And the inhibition of IRE1 α should be detected.

6. Only pharmacological IRE1 α inhibition is not sufficient to determine the regulatory effect of IRE1 α on ferroptosis in vivo. The IRE1 α knockout mice should be used.

7. In Page 5, line 99, "ASL4" should be revised to "ACSL4".

8. In this paper, even in the same cell line, the ferroptosis inducer Erastin was used with different concentrations for different experiment, such as "10 μ M Erastin, 6h", "2.5 μ M Erastin, 24h" and "10 μ M Erastin, 12h". What is the author's purpose or reason for this?

Reviewer #4 (Remarks to the Author):

The identification and validation of ferroptosis modulators is an important goal for biomedical research, particularly that with a disease focus. Although some regulators of ferroptosis have been characterised to date, many questions remain, including which processes are conserved across taxa. This work by Jiang et al will be a significant contribution and will be of general and specific interest.

Overall, this manuscript is well considered and well written. However, I have several specific concerns that must be addressed before I could recommend this manuscript be accepted for publication. These issues are listed below:

Page 7 Line 144: 'We confirmed that feeding with DGLA significantly reduces the lifespan of the wild-type strain'. Use of 'confirmed' in this sentence is an issue as the cited work (ref 15 Perez et al 2020) reports germ-line not longevity effects. More careful language is needed when introducing previous work exploring DGLA supplementation, ferroptosis and lifespan determination in *C. elegans*.

How does this study relate to the previous work by O'Rourke et al doi: 10.1101/gad.205294.112 *Genes & Dev.* 2013. 27: 429-440? This study reports supplementation of wild type (ad libitum-fed) *C. elegans* with DGLA activates autophagy, and increases median lifespan. Numerous replicate experiments are presented in this earlier study. The different outcomes of DGLA exposure need to be discussed and this earlier work cited. Potential differences in dosing (0.3 mM vs 10 mM) may be relevant and should be discussed. Examination of the lifespan effects across a range of DGLA doses would likely identify the consistencies and inconsistencies across these studies. Given the relative importance of the longevity effects in this manuscript I think these additional studies are needed and justified.

Another significant concern is the lack of replicate data supporting the conclusions regarding DGLA and longevity. Methods Line 198: '...45 – 102 worms per group were used to assess the effect on lifespan'. It appears the *C. elegans* life span data is from a single experiment, i.e. without independent replication. Replication of lifespan experiments is crucial for scientific rigour. If replicate data has been collected, then all of this information must be included. If only a single data set has been assessed, then additional experiments must be performed and the all results presented and summarised (e.g. in the Supplemental data).

Supplemental Fig 6. A-C Longevity or lifespan data is typically summarised with a comparison of median survival (days). Please include this information in the figure legend. The text describing the non-parametric tests (log-rank and Wilcoxon) should also be moved to the Figure legend.

In addition, previous reports suggesting ferroptosis may be in play during late life in *C. elegans* (Jenkins et al eLife, 2020 doi:10.7554/eLife.56580) should also be discussed.

Page 7 Line 149: ‘...Therefore, ferroptosis regulation by IRE1 α is conserved...’, a more appropriate statement would be ‘...Therefore, ferroptosis regulation by IRE1 α appears conserved...’.

The *gcs-1* transcript cleaved by IRE1 α data shown in Supp Fig 6D is important supportive information but the gel image shown is not clear and not well annotated. The relevant and specific transcripts (cleaved and uncleaved) should be clearly marked (i.e. arrow, etc), as they are not very distinct. In addition, the purpose of the annotation ‘(base)’ is unclear.

Reviewer #5 (Remarks to the Author):

\The manuscript entitled “IRE1 α determines ferroptosis sensitivity through regulation of glutathione synthesis” describes the mechanism of how UPR protein IRE1 α regulates ferroptosis. RNase activity of IRE1 α accelerates Ferroptosis by degrading the GCLC and SLC7A11 mRNA thus reducing the glutathione synthesis. Although they suggested a novel ferroptosis regulatory mechanism, there are some concerns needed to be corrected including inappropriate result translation, experimental models, and methods.

Major points

- 1) Figure 4, Page 8; Line 181-182. Their conclusion that “4 μ 8c achieves better reduction than liproxstatin-1” shouldn’t be justified as they treated a much higher dose of 4 μ 8c than the liproxstatin-1.
- 2) Supplementary Fig 6A, Page 7; lines 146-152. They showed that the ire-1 kinase domain mutant strains are resistant to DGLA treatment. This does not match the given mechanism of the RNase domain of IRE1 α mediated ferroptosis induction.
- 3) Page 20; lines 83-91. Glutathione measurements were normalized by CCK-8, but the CCK measures NADPH and NADH which both are affected by redox status. They should apply other methods for the normalization of GSH levels such as total protein quantity or cell number.
- 4) Figure 3. Although the author shows that the GCLC mRNA is cleaved by IRE1 α in the in-vitro system, the IRE1 α -mediated mRNA cleavage is needed to be validated in cells by using QPCR.

Minor points

- 1) Page 6; Line 119-121, Supplementary Figure 4C, GCLC levels are too high which raises concern of false positive effect. The GCLC levels might be needed to be similar to IRE1 α KO cells.
- 2) Figure 2A. The effect should be evaluated in normal conditions without the erastin treatment.
- 3) The effect of IRE1 α is only evaluated in MDA-MB-231 cell line. They need at least one more cell line to show consistent effect.
- 4) Supplement Fig. 5, It might be better to include the IRE1 α KO + IRE1 α K907A over the expression group.
- 5) Supplementary Fig. 4 A and B, it would be better if GSH levels are measured to validate the GCLC function.
- 6) Page 2; lines 39-40. It is ambiguous as they are mentioning the subcellular locations of GPX4, FSP1, and DHODH, but not the location of GCH-1 and IRE1 α .
- 7) Fig 2A: They need to evaluate whether the IRE1 α KO can affect intracellular glutamate levels which are also important in the induction of ferroptosis via both SLC7A11 and glutamate inhibition (Kang et al., Cell Metabolism, 2021).

Reviewers' comments:

Reviewer #1 (Remarks to the Author):

In this study, the authors investigated the potential role of IRE1 α in regulating ferroptosis sensitivity. They claim that IRE1 α is a positive regulator of ferroptosis by inhibiting GSH production. Specifically, they showed that IRE1 α -mediated downregulation of GCLC and SLC7A11 contributes to GSH depletion in cancer cells. Finally, they found that IRE1 α inhibitors could prevent renal ischemia-reperfusion injury in mice.

1. Three ER-resident transmembrane signal transducers, PERK, ATF6, and IRE1, constitute the three canonical UPR signaling branches to manage ER stress and restore cellular homeostasis. Previous independent studies have shown that PERK and ATF6 regulates ferroptosis resistance. In contrast to these previous studies, when and how IRE1 plays a different role in promoting ferroptosis. What is the status of other UPR pathways in WT and IRE1 KO cells in response to ferroptosis stimuli? Is this cell type dependent? What are the checkpoints of these three UPR pathways during ferroptosis?

We thank the reviewer for bringing up the relationship between the other UPR pathways and ferroptosis as this is also an important consideration due to the coordinated regulation of the UPR signaling branches during stress responses. Our data indicate that ***IRE1 α 's regulation on ferroptosis through modulating GSH availability is a unique and basal activity of IRE1 α which is separate from its canonical UPR functions, as well as the other UPR branches.*** To support this conclusion, in this revision we determined the status of all the three UPR branches in a triple-negative breast cancer cell line (MDA-MB-231) and a pancreatic cancer cell line (Panc-1) in response to ferroptosis inducing treatments such as erastin or cystine depletion. We found that in contrast to thapsigargin (Tg, which induces ER stress through inhibiting calcium flux) that activates all the UPR pathways, erastin or cystine starvation does not activate the UPR sensor proteins IRE1 α (which is activated through phosphorylation) as well as its downstream XBP1 splicing, PERK (which is activated through phosphorylation), or ATF6 (which is activated through proteolytic cleavage) in either of these cell lines (**Fig. 2A**). Erastin or cystine starvation did activate eIF2 α -ATF4, through the amino acid deficiency sensor GCN2, independent of PERK activation (**Supplementary Fig. 3A**). To further ascertain whether the canonical UPR impacts ferroptosis induced by cystine uptake inhibition, we inhibited the expression of the key UPR transcription factors, XBP1, ATF4 or ATF6 in MDA-MB-231 cells (the major *in vitro* cell line model used in our study) with shRNA. We found that none of these perturbations altered the cellular sensitivity to erastin-induced ferroptosis (**Fig. 2B-D**). We are aware of previously reported studies that identified other UPR sensor proteins in regulating ferroptosis sensitivity. However, those studies were based on different and diverse cellular context, cell types and inducing stimuli that may not be applicable to the cellular and *in vivo* systems used in our study. Because of this complexity, the literature

contains conflicting data with regard to the role of PERK in regulating ferroptosis sensitivity, suggesting both sensitization¹⁻⁸ and resistance⁹ mechanisms. Therefore, the context for PERK involvement in mediating ferroptosis is highly dependent upon the cellular context.

2. The authors show through some rescue experiments that both GCLC and SLC7A11 are mediators of IRE1-related phenotypes. They also claim that IRE1 plays a broad-spectrum role in promoting ferroptosis. However, this may not be true because different ferroptosis activators target different molecules downstream and upstream of GSH synthesis, and additionally have different effects on intracellular iron overload. For example, the authors found that the GCLC inhibitor buthionine sulfoximine (BSO) resensitized IRE1 α -null cells to erastin-induced ferroptosis. We know that BSO is also an inducer of ferroptosis. If IRE1-KO cells are resistant to RSL3 (targeting downstream of GSH depletion), then it is expected to be resistant to BSO as well.

We thank the reviewer for giving us this opportunity to strengthen our mechanistic findings by clarifying these points.

The reviewer's main criticism is that we did not show IRE1 α -KO cells are also resistant to BSO, an inhibitor of GCLC. Instead, we showed that BSO resensitized IRE1 α -KO cells to erastin-induced ferroptosis.

Firstly, data showing that "IRE1 α -KO cells are resistant to BSO" and data showing that "BSO resensitized IRE1 α -KO cells to erastin-induced ferroptosis" would not be in conflict with each other, as they are results of different types of experiments. Answering the question of whether "IRE1 α -KO cells are resistant to BSO" is determined by a comparison between wild-type and IRE1 α -KO cells treated with BSO, whereas whether "BSO resensitizes IRE1 α -KO cells to erastin-induced ferroptosis" is determined by a comparison between with and without BSO co-treatment of erastin-treated IRE1 α -KO cells.

Secondly, although BSO may be a ferroptosis inducer in certain cell lines, we found that it does not induce ferroptosis in MDA-MB-231 cells even at 100 μ M as a single agent (**Supplementary Fig. 4F**). This is not a unique observation with this cell line, as other investigators also reported that BSO alone was insufficient to induce ferroptosis in Huh7 and HepG2 cells¹⁰, rhabdomyosarcoma cells¹¹ or biliary tract cancer cells¹². Since wild-type MDA-MB-231 cells are not sensitive to BSO at all, it is impossible to show that IRE1 α -KO MDA-MB-231 cells are resistant to BSO. Although not active as a single agent, our data suggest that BSO enhances erastin-induced ferroptosis (**Supplementary Fig. 4E**). In **Supplementary Fig. 4E**, we used BSO (a GCLC inhibitor) as a pharmacological approach to confirm that IRE1 α -KO cells can retain a higher level of GSH due to the higher level of GCLC expression in these cells, because in wild-type cells IRE1 α down-regulates GCLC through its RIDD activity. Therefore, BSO can re-

sensitize IRE1 α -KO cells to erastin due to its known inhibitory activity against the higher level of GCLC in these cells.

Finally, the reviewer linked BSO with RSL3. Although we showed that loss of IRE1 α also led to RSL3 (GPX4 inhibitor) resistance (**Supplementary Fig. 1E and F**), this effect is still dependent on IRE1 α 's effect on GCLC/SLC7A11 and GSH level (see the new data in **Supplementary Fig. 3D and E** and the new Discussion). RSL3's pharmacological inhibition of GPX4 in our study was incomplete (**Supplementary Fig. 3D**) which would make GPX4 partially responsive to cystine uptake and GCLC levels. To support this conclusion, we found that RSL3 causes a dose-dependent induction of ferroptosis and at doses higher than the 1 μ M used in **Supplementary Fig. 1E and F**, further induction of ferroptosis was observed (**Supplementary Fig. 3D**). Furthermore, if RSL3 completely inhibits GPX4 activity, the expression level of SLC7A11 and GCLC (upstream of GPX4) should not affect RSL3-induced ferroptosis. However, according to the updated Cancer Therapeutics Response Portal (CTRP, V2) database, in multiple types of cancer cells, both SLC7A11 and GCLC are within the top genes whose higher expression strongly correlates with resistance to RSL3 (**Supplementary Fig. 3E**). Thus, high expression of SLC7A11 and GCLC can promote resistance to RSL3-induced ferroptosis, particularly under conditions in which GPX4 is only partially inhibited by RSL3. This provides a strong argument that under RSL3-induced ferroptosis, the status of the upstream regulators such as SLC7A11-GCLC-GSH can still determine the cellular sensitivity to ferroptosis.

3. The authors suggest that XBP1 is not required for IRE1 activity during ferroptosis. However, this experiment lacked western blot analysis of XBP1 isoform expression in the presence of erastin, RSL3 or BSO. Importantly, this study lacked a positive control, as XBP1 splicing is critical for cell survival under stressful conditions (e.g., apoptosis).

We agree with the reviewer that it is important to determine the status of XBP1 during the induction of ferroptosis. We included in the new **Fig. 2** Western blot analysis of XBP1s during erastin treatment and cystine starvation and in **Supplementary Fig. 4G** during RSL3 and BSO treatment in both MDA-MB-231 and Panc-1 cells. Consistent with the lack of increased IRE1 α phosphorylation by erastin treatment or cystine starvation, neither treatment induced XBP1s expression (**Fig. 2**). RSL3 or BSO also did not have any effect on XBP1s expression (**Supplementary Fig. 4G**). As shown in **Fig. 2B**, when XBP1 expression was inhibited by shRNA, the cellular sensitivity of MDA-MB-231 cells to erastin was not affected.

4. The authors used cancer cell lines to study the in vitro function of IRE1. However, the authors used an in vivo kidney I/R model to explore the protective effect of IRE1 inhibitors. There are some problems with this in vivo design. First, the I/R model is a hybrid cell death model. The authors need to examine whether IRE1 inhibitors or genetic deletion of IRE1 prolongs the survival of GPX4 KO animals in the kidney. Second, the authors need to investigate whether targeting the IRE1 pathway affects the anticancer activity of ferroptosis inducers in vivo. Importantly, knockdown of XBP1 is an important in vivo control to support their hypothesis that IRE1 regulates ferroptosis in an

XBP1-independent manner.

The major finding of our study is that the RIDD activity of IRE1 α sensitizes cells to ferroptosis, therefore pharmacological inhibition of IRE1 α may alleviate pathological conditions caused by excessive ferroptosis. For this reason, the kidney I/R model is suitable for testing this hypothesis in normal tissues. Furthermore, this model has been widely used by the ferroptosis research field to study the contribution of ferroptosis to tissue damage *in vivo*¹³⁻¹⁵. In this study while we detected robust induction of lipid peroxidation/ferroptosis marker 4-HNE in post-I/R mouse kidney (**Fig. 6 and 7**), we did not detect noticeable apoptosis and autophagy markers, cleaved caspase 3 and processing of LC3B, respectively (**Supplementary Fig. 8E**) in the injured kidneys. This finding suggests that ferroptosis is the major type of cell death induced by I/R in the kidney, which is consistent with previous reports^{14,16,17}. GPX4 deletion in mice causes acute renal failure and death in mice¹³. To date, no combined inactivation of any other gene(s) has been shown to rescue this lethal phenotype. More importantly, as our elucidated mechanism of IRE1 α in regulating ferroptosis sensitivity is upstream of GPX4, GPX4 KO mice would not help to define the role of IRE1 α in ferroptosis regulation *in vivo*. Similarly, promoting anti-cancer activity through **induction of** ferroptosis is beyond the scope of our study which is to **alleviate** ferroptosis-mediated tissue damage through inhibiting IRE1 α . In the revised manuscript, we generated conditional Ire1 α -KO mice and examined the effect of genetic Ire1 α loss on I/R-induced kidney injury (**Fig. 6 and Supplementary Fig. 7A-D**). The results from this genetically modified mouse model confirmed our initial findings with pharmacological Ire1 α inhibition in the original manuscript.

5. It appeared that IRE1 KO prevented erastin-induced downregulation of GPX4 protein (Fig. S3). What is the mechanism?

To address this point, we added the following new data and discussion in the revised manuscript – “Although previous studies reported that GPX4 protein level decreases during erastin (doses >10 μ M) and other FIN treatment¹⁸⁻²², we did not observe this in either MDA-MB-231 or Panc-1 cells during erastin ($\leq 10 \mu$ M) or cystine starvation induced ferroptosis (**Fig. 3C and Supplementary Fig. 3A**), suggesting that this may be a drug concentration and cell type-specific phenomenon.”

6. Individual experimental conditions for this study varied widely, such as the use of different drug concentrations (e.g., 2.5, 5, and 10 μ M erastin in different figures). In addition, its core finding that the effect of IRE on GCLC and SLC7A11 expression is based on changes in the basal state. It is unclear whether IRE1-mediated downregulation of GCLC and SLC7A11 at baseline affects other ER stress sensitivities.

Due to technical considerations and limitations of different types of assays, we utilized different combinations of dose and time for erastin treatment. For example, for all the viability assays based on CCK-8 in MDA-MB-231 and Panc-1 cells, we consistently reported the results at 2.5 μ M erastin after 24 hours (**Fig. 1A-B and E, Fig. 5E, Supplementary Fig. 1C, Supplementary Fig. 4A and E**). For the viability assays on

IRE1 α -null based cell lines (e.g. **Fig. 1G and Supplementary Fig. 4D**), as they are in general more resistant to erastin compared to wild-type cells, we used a higher concentration of erastin (5 μ M) after 24 hours. However, if we applied the same dose and time combination for lipid peroxidation assay by BODIPY flow cytometry (**Fig. 1F**), IncuCyte live cell imaging (**Fig. 1D**) or cellular glutathione measurement (**Fig. 3A, Fig. 5B and Supplementary Fig. 4B**) that is normalized by cell counting, there would be significant number of dead cells especially in the wild-type control or WT IRE1 α -overexpressing groups after 24 hours, which prevents accurate measurement of the assay. Therefore, we collected the data with higher erastin concentrations (5 μ M for the glutathione assay and 10 μ M for the flow cytometry and IncuCyte live cell imaging assay) but after shorter period of time (12 hours). In certain figures, we also evaluated cell viabilities based on CCK-8 under a range of erastin concentrations (**Fig. 1C, Fig. 2B-D, Fig. 5A, and Supplementary Fig. 3C and F-G**). To comprehensively evaluate the dose- and time-dependent induction of ferroptosis by erastin, we included in the revised manuscript IncuCyte live cell imaging of MDA-MB-231 cells under these conditions (**Supplementary Fig. 3C**) covering the whole dose range (0 – 10 μ M) and time (0 - 24 hours) range used in our study. For non-targeting MDA-MB-231 cells overexpressing luciferase, IRE1 α -null cells reconstituted with either luciferase, wild-type IRE1 α or K907A mutant IRE1 α , we also performed the same assay to reveal the dose- and time-dependent response to erastin treatment (**Supplementary Fig. 2**).

With regard to the question on basal activity of IRE1 α in regulating GCLC and SLC7A11 expression, as we show in **Fig. 2A**, there was no detectable induction of the UPR pathways (IRE1 α , PERK and ATF6) during erastin treatment or cystine starvation. In addition, when the expression of individual UPR transcription factors (XBP1, ATF4 and ATF6) was inhibited by shRNA, the sensitivity of MDA-MB-231 cells to erastin was not affected (**Fig. 2B-D**). Therefore, we concluded that this novel activity of IRE1 α is independent of its canonical UPR function in response to ER stress or the other UPR branches.

7. The author cites his own publication (ref10) in multiple places showing that ferroptosis inducers activate the UPR pathway. However, this is not the case and the cited literature only states that doxorubicin can activate the UPR response and does not refer to any ferroptosis studies. The accuracy of other references is also questionable.

As both the main text and the Methods Section had its own reference list in the original manuscript, the reviewer referred to the 10th reference of the Methods Section in the original manuscript (doxorubicin as an IRE1 α -XBP1 inhibitor) where Ref #10 of the main manuscript (erastin activates eIF2 α -ATF4) was actually cited. The existence of two different reference lists was due to technical issues when transferring our manuscript from another Nature journal to Nature Communications. We apologize for this confusion and it has been resolved in the revised manuscript with all the references combined into a single Reference Section.

8. The dose of IRE1 inhibitor used in this study was much higher than the IC₅₀. For example, the IC₅₀ of 4 μ 8c is 76 nM, while 30 μ M was used in this study. This will lead to off-target effects in the interpretation of the conclusions.

76 nM is the IC₅₀ in cell-free assays (selleckchem.com) where the activity of IRE1 α is assayed using recombinant IRE1 α protein mixed with 4 μ 8c directly. The concentration required for cell-based assays as well as dose required for *in vivo* studies are generally much higher due to factors related to drug uptake, cellular and organismic drug metabolism. In the revised manuscript we added an assay to compare a range of 4 μ 8c concentrations in inhibiting erastin-induced ferroptosis in MDA-MB-231 cells. As shown in **Supplementary Fig. 3F**, 30 μ M or higher 4 μ 8c was required to inhibit erastin-induced ferroptosis in these cells. Therefore, we used 30 μ M 4 μ 8c in our *in vitro* cell-based assays. This concentration level was also widely used to study the effects of IRE1 α inhibition in a wide range of *in vitro* cell-based assays previously ²³.

9. Finally, the authors' core finding that IRE1 α determines ferroptosis sensitivity through GSH modulation is not new (PMID: 35724508; PMID: 36091771; PMID: 34558645). Although the models may be different, the authors need to discuss these findings from other groups.

After careful evaluation of the cited articles, our findings provide novel mechanistic insight for the regulation of ferroptosis and reveal a fundamentally novel function of IRE1 \$\alpha\$. PMID: 35724508 investigates how ATF4 suppresses ferroptosis. These authors showed that IRE1 α is upregulated and has a **protective role** in cell survival upon treatment with sorafenib, which has multiple effects on signal transduction pathways in normal tissues and in tumors. Moreover, whether sorafenib can induce ferroptosis is controversial as there are contradictory data in the literature ²⁴. This is in contrast to our finding that IRE1 α **promotes ferroptosis** by limiting glutathione (GSH) synthesis. PMID: 34558645 identified that inhibition of either XBP1, ATF4 or ATF6 expression attenuated dihydroartemisinin (DHA)-induced ferroptosis, as these UPR transcription factors may be required for DHA-induced γ -glutamylcyclotransferase 1 (CHAC1) expression, which leads to GSH degradation and sensitization to ferroptosis. **However, this study did not directly evaluate the role of IRE1 α .** PMID: 36091771 identified that IRE1 α expression inhibition attenuates ischemia-reperfusion (I/R)-induced ferroptosis in human renal epithelial cells and that an uncommonly used small molecule IRE1 α inhibitor (irestatin 9389) can alleviate I/R-induced kidney injury in mice. Investigators in the UPR field have generally used other more specific inhibitors of IRE1 α when studying IRE1 α function experimentally. Nevertheless, these findings are consistent with our study which further augment the significance of IRE1 α in ferroptosis regulation. However, these investigators suggest that IRE1 α regulates ferroptosis sensitivity through its **activation of JNK**, which is distinct from our study (**Supplementary Fig. 8C**). Our study provides strong mechanistic data implicating the mRNA degradation (RIDD) function of IRE1 α as a critical activity for the regulation of ferroptosis. We have included detailed discussion of these previous findings in the revised manuscript (Line 263).

Reviewer #2 (Remarks to the Author):

Jiang et al report that modulating Ire1 activity can determine ferroptosis sensitivity in cells and worms. I find the main observation interesting, but I think additional experiments should be done to meet the bar for publication in Nature Comm.

One aspect that the manuscript does not address, but I think it should be addressed, is how Ire1 is being activated in the described experiments. One possibility (1) is that the FINs cause the accumulation of misfolded proteins in the ER lumen, especially since they cause Cys depletion in cells. Other possibility (2) is that the FINs cause lipid stress that is sensed directly by Ire1. The authors can use Ire1 mutations to distinguish if 1 or 2 in the cell studies. I think 2 would be more interesting than 1, but in that case, the authors should provide additional data to show how the lipid peroxides (or other agents of lipid stress) directly activate Ire1 in the ER membrane in these models. Or perhaps they can find a fraction of Ire1 in the plasma membrane or the inner mitochondrial membrane, the sites for accumulation for lipid peroxides?

We thank the reviewer for the thoughtful analysis and suggestion. Through our detailed analysis, we have now included additional data in **Fig. 2**, showing that ***neither erastin treatment nor cystine starvation (ferroptosis-inducing stimuli used throughout the manuscript) activates IRE1 α or the other UPR pathways***. These data indicate that general induction of ER stress by various ferroptosis inducers is not the mechanism for IRE1 α regulation of ferroptosis induction. Therefore, the effect of IRE1 α on GCLC and SLC7A11, as well as the associated cystine uptake and GSH biosynthesis represent a basal “surveillance” mechanism of IRE1 α . It will be interesting to further investigate why the cell utilizes such a non-canonical function of IRE1 α to regulate the cellular GSH availability and sensitivity to ferroptosis. Based on our current understanding, we believe that this basal activity of IRE1 α may play various as yet unidentified roles in a wide range of cellular processes. As small molecule IRE1 α inhibitors are being tested clinically, these novel findings can potentially reveal novel therapeutic opportunities for treating related pathological conditions.

Other points:

- Data points should be added to Fig.1A - C and similar panels thereafter.

We have added the original data points to all the histograms throughout the revised manuscript.

- Molecular weight markers are absent from 2G. Also they should indicate the localization of the stem loop motif in fragment #4 and the expected size of the fragments upon cleavage.

Molecular weight has been added to Fig. 2G (now **Fig. 3G**). As shown in **Fig. 3D**, the relative positions of the individual fragments (#1-5) are aligned with their genomic

boundaries. Therefore, the location of the stem-loop-stem structure of the IRE1 α cleavage site can be recognized in **Fig. 3D** (on the right side of fragment #4) through the position mark below the exon organization lane. The predicted sizes of the cleaved fragments from fragment #4 are 394 bases and 126 bases, which match the sizes of the fragments in **Fig. 3G**.

- The worm data is important and it should be included as a main figure, but smaller molecular weight markers should be added to SFig6D, so that the size of the fragments can be evaluated. Also, in this case can the authors identify a stem loop motif as the site for cleavage? If yes, they can also generate mutants that cannot be cleaved by Ire1.

As suggested by the reviewer, in the revised manuscript we include the worm data in the new **Fig. 4** as a main figure. Smaller molecular weight markers are also added to **Fig. 4D and F** next to the gel images. We performed cleavage assays similar to those for the human GCLC in **Fig. 3** by first dividing the *gcs-1* transcript into two half-sized fragments and found that the cleavage occurred only in fragment #1. Next, we further identified the stem-loop-stem cleavage site shown in **Fig. 4D and E**. We also confirmed that when the GC sequence in the cleavage site is mutated to CC as we did for human GCLC, the fragment cleavage was abrogated (**Fig. 4F**).

- I would recommend that the authors increase the number of main figures (from 4 to 5-6) and reduce the number of Sup Figs.

We thank the reviewer for this suggestion and increased the number of main figures to 7 in the revised manuscript.

Reviewer #3 (Remarks to the Author):

Ferroptosis is an important cell death mechanism. This study by Jiang et al. tried to determine the regulatory effect of IRE1 α on ferroptosis. The study is overall interesting. But there are several questions that need to be addressed.

1. In this study, the regulatory effect of IRE1 α on ferroptosis was only determined *in vitro*. The authors should also detect the regulatory effect of IRE1 α on ferroptosis and the related pathway *in vivo*.

We appreciate the reviewer's suggestion on evaluating IRE1 α 's regulation on ferroptosis *in vivo*. In the original submission we applied pharmacological IRE1 α inhibition with 4 μ 8c in the mouse kidney ischemia-reperfusion (I/R) model to test its effect on alleviating I/R-induced ferroptosis and tissue damage. In the revised manuscript we added genetic IRE1 α deletion to further confirm the IRE1 α -specific effect on ferroptosis *in vivo* (**Fig. 6 and Supplementary Fig. 7A-D**). We detected elevated expression of *Gclc* at both mRNA and protein level in *Ire1 α* -null mouse kidneys compared to control (*Ire1 α* wild-type) kidneys (**Supplementary Fig. 7C-D**), confirming the identified IRE1 α -GCLC regulatory circuit *in vivo*.

2. It is puzzling why *Ern1* was knocked out in mouse embryonic fibroblast (MEF), but not IRE1 α , or necessary introduction to *Ern1* lacks in this paper.

Ern1 is the gene name of mouse IRE1 α . We added this description in the revised manuscript for clarification (Line 189, 225, and 395). We also changed the labels of the MEFs in **Fig. 1C** to *Ire1 α ^{+/+}* and *Ire1 α ^{-/-}* to avoid this confusion.

3. In Fig. 4, the authors claim all the effects presented on ferroptosis. How can the authors exclude that the effects of IRE1 α on other death forms, such as autophagy and apoptosis. The authors need to present a systematic characterization of the different cell death pathways in their models.

In the revised manuscript, we evaluated the presence and extent of apoptosis and autophagy markers in IRE1 α -null mouse kidneys and kidneys from mice treated with 4 μ 8c upon ischemia-reperfusion (I/R) injury (IRI). As shown in **Supplementary Fig. 8E**, I/R did not significantly induce either the apoptosis (cleaved caspase 3) or autophagy (processed LC3B) markers, nor did IRE1 α deficiency or pharmacological IRE1 α inhibition display any effect on these two markers. This finding suggests that ferroptotic cell death is the major type of cell death that contributes to IRI, which is consistent with previous reports^{14,16,17}.

4. In this study, the authors use the kidney ischemia model. But why were the renal cell lines not detected in vitro?

To address this criticism, we added additional cell based assays using the immortalized human proximal tubule epithelial cell line HK-2. We found that both ferrostatin-1, a ferroptosis inhibitor 1 and 4 μ 8c, an IRE1 α inhibitor, reduced the endogenous lipid peroxidation level in HK-2 cells and inhibiting IRE1 α expression by shRNA had similar but intermediate effects (**Supplementary Fig. 8A and D**). In contrast to the *in vivo* IRI treatment that showed a negligible effect on autophagy in mouse kidney (**Supplementary Fig. 8E**), we detected more processed LC3B (LC3B-I to LC3B-II) in HK-2 cells upon hypoxia-reoxygenation (H/R) and RSL3 treatment (**Supplementary Fig. 8B**). However, IRE1 α inhibition did not have any effect on the induction of autophagy by H/R or RSL3 (**Supplementary Fig. 8B**), suggesting that IRE1 α is not involved in mediating autophagy under this context. In addition, IRI did not induce apoptosis in mouse kidneys (**Supplementary Fig. 8E**) and neither H/R nor RSL3 treatment induced apoptosis in HK-2 cells (**Supplementary Fig. 8B**).

5. The authors need to demonstrate how to select the dosage of IRE1 α inhibitor? A dose-response should be demonstrated both in vitro and in vivo. And the inhibition of IRE1 α should be detected.

Based on the reviewer's suggestion, we added an assay to compare a range of 4 μ 8c concentrations in inhibiting erastin-induced ferroptosis in MDA-MB-231 cells. As shown in **Supplementary Fig. 3F**, 30 μ M or higher 4 μ 8c was required to inhibit erastin-induced

ferroptosis. Therefore, we used 30 μM 4 μ8c in our *in vitro* assays. This concentration level was also widely used to study the effects of IRE1 α inhibition in a wide range of *in vitro* assays²³. To determine the *in vivo* dose of 4 μ8c required to suppress ferroptosis, we compared 20 mg/kg and 40 mg/kg 4 μ8c in mouse kidney and found that 40 mg/kg 4 μ8c was required to up-regulate Gclc expression level (**Supplementary Fig. 7G**). Consistent with this finding, while 40 mg/kg 4 μ8c significantly alleviated kidney damage measured by BUN and creatinine levels after I/R, 20 mg/kg 4 μ8c didn't (**Fig. 7A-B**).

6. Only pharmacological IRE1 α inhibition is not sufficient to determine the regulatory effect of IRE1 α on ferroptosis *in vivo*. The IRE1 α knockout mice should be used.

We have included additional data to address this point. Please refer to IRE1 α genetic deletion data in mice as described in response to critique #1 above (**Fig. 6 and Supplementary Fig. 7A-D**).

7. In Page 5, line 99, "ASL4" should be revised to "ACSL4".

We thank the reviewer for identifying this typo and it has been corrected in the revised manuscript.

8. In this paper, even in the same cell line, the ferroptosis inducer Erastin was used with different concentrations for different experiment, such as "10 μM Erastin, 6h", "2.5 μM Erastin, 24h" and "10 μM Erastin, 12h". What is the author's purpose or reason for this?

Reviewer #1 expressed a similar concern in Point #6. Here's our response –

Due to technical considerations and limitations of different types of assays, we utilized different combinations of dose and time for erastin treatment. For example, for all the viability assays based on CCK-8 in MDA-MB-231 and Panc-1 cells, we consistently reported the results at 2.5 μM erastin after 24 hours (**Fig. 1A-B and E, Fig. 5E, Supplementary Fig. 1C, Supplementary Fig. 4A and E**). For the viability assays on IRE1 α -null based cell lines (e.g. **Fig. 1G and Supplementary Fig. 4D**), as they are in general more resistant to erastin compared to wild-type cells, we used a higher concentration of erastin (5 μM) after 24 hours. However, if we applied the same dose and time combination for lipid peroxidation assay by BODIPY flow cytometry (**Fig. 1F**), IncuCyte live cell imaging (**Fig. 1D**) or cellular glutathione measurement (**Fig. 3A, Fig. 5B and Supplementary Fig. 4B**) that is normalized by cell counting, there would be significant number of dead cells especially in the wild-type control or WT IRE1 α -overexpressing groups after 24 hours, which prevents accurate measurement of the assay. Therefore, we collected the data with higher erastin concentrations (5 μM for the glutathione assay and 10 μM for the flow cytometry and IncuCyte live cell imaging assay) but after shorter period of time (12 hours). In certain figures, we also evaluated cell viabilities based on CCK-8 under a range of erastin concentrations (**Fig. 1C, Fig. 2B-D, Fig. 5A, and Supplementary Fig. 3C and F-G**). To comprehensively evaluate the dose- and time-dependent induction of ferroptosis by erastin, we included in the revised manuscript IncuCyte live cell imaging of MDA-MB-231 cells under these

conditions (**Supplementary Fig. 3C**) covering the whole dose range (0 – 10 μ M) and time (0 - 24 hours) range used in our study. For non-targeting MDA-MB-231 cells overexpressing luciferase, IRE1 α -null cells reconstituted with either luciferase, wild-type IRE1 α or K907A mutant IRE1 α , we also performed the same assay to reveal the dose- and time-dependent response to erastin treatment (**Supplementary Fig. 2**).

Reviewer #4 (Remarks to the Author):

The identification and validation of ferroptosis modulators is an important goal for biomedical research, particularly that with a disease focus. Although some regulators of ferroptosis have been characterised to date, many questions remain, including which processes are conserved across taxa. This work by Jiang et al will be a significant contribution and will be of general and specific interest.

We thank the reviewer for emphasizing the significance and the potential impact of our study in biomedical research and translation into clinical practice.

Overall, this manuscript is well considered and well written. However, I have several specific concerns that must be addressed before I could recommend this manuscript be accepted for publication. These issues are listed below:

Page 7 Line 144: 'We confirmed that feeding with DGLA significantly reduces the lifespan of the wild-type strain'. Use of 'confirmed' in this sentence is an issue as the cited work (ref 15 Perez et al 2020) reports germ-line not longevity effects. More careful language is needed when introducing previous work exploring DGLA supplementation, ferroptosis and lifespan determination in *C. elegans*.

We appreciate the reviewer's advice to improve the accuracy of our description. We have changed the wording to be "Consistent with these findings we showed that feeding *C. elegans* with DGLA significantly reduced the lifespan of the wild-type strain (N2)" (Line 185).

How does this study relate to the previous work by O'Rourke et al doi: 10.1101/gad.205294.112 Genes & Dev. 2013. 27: 429-440? This study reports supplementation of wild type (ad libitum-fed) *C. elegans* with DGLA activates autophagy, and increases median lifespan. Numerous replicate experiments are presented in this earlier study. The different outcomes of DGLA exposure need to be discussed and this earlier work cited. Potential differences in dosing (0.3 mM vs 10 mM) may be relevant and should be discussed. Examination of the lifespan effects across a range of DGLA doses would likely identify the consistencies and inconsistencies across these studies. Given the relative importance of the longevity effects in this manuscript I think these additional studies are needed and justified.

As suggested by the reviewer, in the revised manuscript we included a detailed discussion on the previous work by O'Rourke et al and cited the article (Line 324). We agree that the dose-dependent effect of DGLA on worm lifespan is an interesting and

important research topic. However, as our study focuses on evaluating the role of IRE1 α (ire-1 in *C. elegans*) in the context of ferroptosis induction, we feel that additional studies regarding the DGLA dose selection and other types of cellular/organismal response in addition to ferroptosis would distract from the focus of this article and be best suited for a future manuscript.

Another significant concern is the lack of replicate data supporting the conclusions regarding DGLA and longevity. Methods Line 198: ‘...45 – 102 worms per group were used to assess the effect on lifespan’. It appears the *C. elegans* life span data is from a single experiment, i.e without independent replication. Replication of lifespan experiments is crucial for scientific rigour. If replicate data has been collected, then all of this information must to be included. If only a single data set has been assessed, then additional experiments must to be performed and the all results presented and summarised (e.g. in the Supplemental data).

Thank you for this comment regarding additional data for scientific rigor. We repeated the DGLA-fed *C. elegans* life span experiments and reached the same conclusion. We have included the replicate data in the new **Supplementary Fig. 6**.

Supplemental Fig 6. A-C Longevity or lifespan data is typically summarised with a comparison of median survival (days). Please include this information in the figure legend. The text describing the non-parametric tests (log-rank and Wilcoxon) should also be moved the Figure legend.

We have included median survival data and description of the statistical tests in the figure legends of the relevant figures (now **Fig. 4 and Supplementary Fig. 6**).

In addition, previous reports suggesting ferroptosis may be in play during late life in *C. elegans* (Jenkins et al eLife, 2020 doi:10.7554/eLife.56580) should also be discussed.

As suggested by the reviewer, we have added discussion on this important study in the revised manuscript (Line 183).

Page 7 Line 149: ‘...Therefore, ferroptosis regulation by IRE1 α is conserved...’, a more appropriate statement would be “...Therefore, ferroptosis regulation by IRE1 α appears conserved...’.

Thank you for this suggestion. We have made the change according to the reviewer’s suggestion. The new description in the revised manuscript is “Together, these data reveal a novel, evolutionarily conserved regulatory mechanism between IRE1 α and ferroptosis through RIDD activity on GCLC, a key enzyme in the GSH synthesis pathway.” (Line 199)

The gcs-1 transcript cleaved by IRE1 α data shown in Supp Fig 6D is important supportive information but the gel image shown is not clear and not well annotated. The relevant and specific transcripts (cleaved and uncleaved) should be clearly marked (i.e.

arrow, etc), as they are not very distinct. In addition, the purpose of the annotation '(base)' is unclear.

We have moved the main *C. elegans* study data to **Fig. 4** as one of the main figures. The old gel image in the previous Supplementary Fig. 6D has been replaced with a new one of better quality (**Fig. 4D**). We also included cleavage of the full-length *gcs-1* transcript to better support the conclusion. In addition, we added the cleavage of 40-base synthesized RNA oligos encompassing the identified cleavage site (wild-type and mutant) to further confirm the specificity of the cleavage (**Fig. 4F**). In each image, the cleaved products are marked with black triangles. The annotation of "base" is the unit of the molecular weight markers shown on the left side of the gel images. As the substrate of IRE1 α cleavage is single-stranded RNA, the unit of the molecular weight marker is "base" instead of "base pair/bp".

Reviewer #5 (Remarks to the Author):

\The manuscript entitled "IRE1 α determines ferroptosis sensitivity through regulation of glutathione synthesis" describes the mechanism of how UPR protein IRE1 α regulates ferroptosis. RNase activity of IRE1 α accelerates Ferroptosis by degrading the GCLC and SLC7A11 mRNA thus reducing the glutathione synthesis. Although they suggested a novel ferroptosis regulatory mechanism, there are some concerns needed to be corrected including inappropriate result translation, experimental models, and methods.

Major points

1) Figure 4, Page 8; Line 181-182. Their conclusion that "4 μ 8c achieves better reduction than liproxstatin-1" shouldn't be justified as they treated a much higher dose of 4 μ 8c than the liproxstatin-1.

4 μ 8c and liproxstatin-1 are two different chemical compounds with different physicochemical properties and *in vivo* pharmacological kinetics (PKs). Therefore, it is difficult to interpret a direct dose comparison of the *in vivo* dosing for these compounds. A detailed PK/PD study for these compounds is beyond the scope of this study and we utilized doses that are generally accepted by the other investigators in the field. In the revised manuscript, we changed the description pointed out by the reviewer to "Comparable to liproxstatin-1, 40 mg/kg 4 μ 8c treatment also significantly reduced tubular damage" to avoid the confusion.

2) Supplementary Fig 6A, Page 7; lines 146-152. They showed that the *ire-1* kinase domain mutant strains are resistant to DGLA treatment. This does not match the given mechanism of the RNase domain of IRE1 α mediated ferroptosis induction.

IRE1 α 's RNase domain activity is governed by its kinase domain. Activation of the RNase activity starts with the luminal domain sensing unfolded and misfolded proteins in the ER lumen, which triggers dimerization and autophosphorylation of the kinase domain. Activation of the kinase domain eventually activates the RNase domain and its activity. Therefore, if the kinase domain is mutated the RNase activity is lost. The

detailed molecular mechanisms of IRE1 α activation have been extensively studied²⁵⁻²⁹. We have added additional text in the manuscript to clarify this point (Line 188).

3) Page 20; lines 83-91. Glutathione measurements were normalized by CCK-8, but the CCK measures NADPH and NADH which both are affected by redox status. They should apply other methods for the normalization of GSH levels such as total protein quantity or cell number.

As suggested by the reviewer, in the revised manuscript, all the glutathione measurements were normalized to cell counts (**Fig. 3A, Fig. 5B, and Supplementary Fig. 4B**).

4) Figure 3. Although the author shows that the GCLC mRNA is cleaved by IRE1 α in the in-vitro system, the IRE1 α -mediated mRNA cleavage is needed to be validated in cells by using QPCR.

We performed qPCR analysis of GCLC mRNA expression under different IRE1 α status and the data were summarized in the original Fig. 2B and the revised **Fig. 3B**. We strongly agree with the reviewer that it is necessary to verify GCLC mRNA levels in cells but this reviewer may have missed the data included in the original submission.

Minor points

1) Page 6; Line 119-121, Supplementary Figure 4C, GCLC levels are too high which raises concern of false positive effect. The GCLC levels might be needed to be similar to IRE1 α KO cells.

In the original Supplementary Fig. 4C (new **Supplementary Fig. 4A**), the purpose of overexpressing GCLC in IRE1 α -null cells already overexpressing wild-type IRE1 α was to evaluate whether compensating for the reduced level of GCLC caused by wild-type IRE1 α can reverse the increased sensitivity to ferroptosis due to decreased GSH level. Although the level of GCLC reconstitution suggested by the reviewer represents the optimal experimental condition for this assay, practically it is difficult to control the overexpression level of GCLC through viral transduction with an exogenous promoter. Furthermore, as elevated GCLC level and the associated higher GSH level may promote cell survival and proliferation, the resulting stable cell line may favor high level of GCLC expression. However, we feel that even though the overexpressed GCLC level exceeds what we see in IRE1 α -null cells, mechanistically it revealed that forced GCLC expression can rescue the decreased cell viability in IRE1 α -null cells reconstituted with wild-type IRE1 α , therefore confirmed that GCLC is the key downstream effector in IRE1 α -mediated sensitization to ferroptosis. Our results were confirmed through multiple replicates of these experiments.

2) Figure 2A. The effect should be evaluated in normal conditions without the erastin treatment.

In the original Fig. 2A we measured GSH level under erastin treatment because this is the condition with which we observed difference in viability between wild-type and IRE1 α -null cells, as well as between IRE1 α -null cells reconstituted with luciferase, wild-type IRE1 α or the K907A mutant IRE1 α . In the new **Fig. 3A**, we included the GSH measurements without erastin treatment (DMSO only) in each group and normalized GSH levels under erastin treatment to the values under DMSO treatment. The conclusion of this assay remained the same.

3) The effect of IRE1 α is only evaluated in MDA-MB-231 cell line. They need at least one more cell line to show consistent effect.

We feel that the reviewer may have overlooked the data demonstrating the effect of IRE1 α on Panc-1 cells and MEFs in the original manuscript (See **Fig 1**). In the revised manuscript we also tested the effect in HK-2 cells (**Supplementary Fig. 8**).

4) Supplement Fig. 5, It might be better to include the IRE1 α KO + IRE1 α K907A over the expression group.

We added the IRE1 α -KO + IRE1 α K907A group into the new **Supplementary Fig. 5B**.

5) Supplementary Fig. 4 A and B, it would be better if GSH levels are measured to validate the GCLC function.

We added GSH measurements to the new **Supplementary Fig. 4B**.

6) Page 2; lines 39-40. It is ambiguous as they are mentioning the subcellular locations of GPX4, FSP1, and DHODH, but not the location of GCH-1 and IRE1 α .

GCH-1 is present in more than a single subcellular location, including the cytosol and nucleus, and possibly the mitochondria³⁰. Therefore, we didn't assign a specific localization to GCH-1 and in the revised manuscript we describe the localization of GCH-1 as "broadly localized" (Line 26). In the original manuscript we did state that IRE1 α is "an endoplasmic reticulum (ER) resident protein". We added description in the revised manuscript to clarify these points (Line 26-27).

7) Fig 2A: They need to evaluate whether the IRE1 α KO can affect intracellular glutamate levels which are also important in the induction of ferroptosis via both SLC7A11 and glutamate inhibition (Kang et al., Cell Metabolism, 2021).

Following the reviewer's suggestion, we measured intracellular glutamate levels in MDA-MB-231 cells with different IRE1 α status (non-targeting+luciferase, IRE1 α -null+luciferase/wild-type IRE1 α /K907A IRE1 α). The results are included in **Supplementary Fig. 3B** and the measured glutamate levels are normalized to cell numbers as in the updated glutathione measurements.

References:

- 1 Liang, Y. *et al.* Toosendanin induced hepatotoxicity via triggering PERK-eIF2alpha-ATF4 mediated ferroptosis. *Toxicol Lett* **377**, 51-61 (2023). <https://doi.org:10.1016/j.toxlet.2023.02.006>
- 2 Wei, R. *et al.* Tagitinin C induces ferroptosis through PERK-Nrf2-HO-1 signaling pathway in colorectal cancer cells. *Int J Biol Sci* **17**, 2703-2717 (2021). <https://doi.org:10.7150/ijbs.59404>
- 3 Zhang, X. *et al.* Resveratrol protected acrolein-induced ferroptosis and insulin secretion dysfunction via ER-stress-related PERK pathway in MIN6 cells. *Toxicology* **465**, 153048 (2022). <https://doi.org:10.1016/j.tox.2021.153048>
- 4 Li, M. D. *et al.* Arsenic induces ferroptosis and acute lung injury through mtROS-mediated mitochondria-associated endoplasmic reticulum membrane dysfunction. *Ecotoxicol Environ Saf* **238**, 113595 (2022). <https://doi.org:10.1016/j.ecoenv.2022.113595>
- 5 He, Z. *et al.* Cadmium induces liver dysfunction and ferroptosis through the endoplasmic stress-ferritinophagy axis. *Ecotoxicol Environ Saf* **245**, 114123 (2022). <https://doi.org:10.1016/j.ecoenv.2022.114123>
- 6 Zhao, C. *et al.* Endoplasmic reticulum stress-mediated autophagy activation is involved in cadmium-induced ferroptosis of renal tubular epithelial cells. *Free Radic Biol Med* **175**, 236-248 (2021). <https://doi.org:10.1016/j.freeradbiomed.2021.09.008>
- 7 Shao, Y. *et al.* The inhibition of ORMDL3 prevents Alzheimer's disease through ferroptosis by PERK/ATF4/HSPA5 pathway. *IET Nanobiotechnol* **17**, 182-196 (2023). <https://doi.org:10.1049/nbt2.12113>
- 8 Pu, X. *et al.* PERK/ATF3-Reduced ER Stress on high potassium environment in the suppression of tumor ferroptosis. *J Cancer* **14**, 1336-1349 (2023). <https://doi.org:10.7150/jca.83556>
- 9 Chen, Y. *et al.* Dihydroartemisinin-induced unfolded protein response feedback attenuates ferroptosis via PERK/ATF4/HSPA5 pathway in glioma cells. *J Exp Clin Cancer Res* **38**, 402 (2019). <https://doi.org:10.1186/s13046-019-1413-7>
- 10 Lippmann, J., Petri, K., Fulda, S. & Liese, J. Redox Modulation and Induction of Ferroptosis as a New Therapeutic Strategy in Hepatocellular Carcinoma. *Transl Oncol* **13**, 100785 (2020). <https://doi.org:10.1016/j.tranon.2020.100785>
- 11 Habermann, K. J., Grunewald, L., van Wijk, S. & Fulda, S. Targeting redox homeostasis in rhabdomyosarcoma cells: GSH-depleting agents enhance auranofin-induced cell death. *Cell Death Dis* **8**, e3067 (2017). <https://doi.org:10.1038/cddis.2017.412>
- 12 Li, Q. *et al.* The effects of buthionine sulfoximine on the proliferation and apoptosis of biliary tract cancer cells induced by cisplatin and gemcitabine. *Oncol Lett* **11**, 474-480 (2016). <https://doi.org:10.3892/ol.2015.3879>
- 13 Friedmann Angeli, J. P. *et al.* Inactivation of the ferroptosis regulator Gpx4 triggers acute renal failure in mice. *Nat Cell Biol* **16**, 1180-1191 (2014). <https://doi.org:10.1038/ncb3064>

- 14 Linkermann, A. *et al.* Synchronized renal tubular cell death involves ferroptosis. *Proc Natl Acad Sci U S A* **111**, 16836-16841 (2014).
<https://doi.org:10.1073/pnas.1415518111>
- 15 Lee, H. *et al.* Energy-stress-mediated AMPK activation inhibits ferroptosis. *Nat Cell Biol* **22**, 225-234 (2020). <https://doi.org:10.1038/s41556-020-0461-8>
- 16 Tonnus, W. & Linkermann, A. The in vivo evidence for regulated necrosis. *Immunol Rev* **277**, 128-149 (2017). <https://doi.org:10.1111/imr.12551>
- 17 Tonnus, W. *et al.* Dysfunction of the key ferroptosis-surveilling systems hypersensitizes mice to tubular necrosis during acute kidney injury. *Nat Commun* **12**, 4402 (2021). <https://doi.org:10.1038/s41467-021-24712-6>
- 18 Shimada, K. *et al.* Global survey of cell death mechanisms reveals metabolic regulation of ferroptosis. *Nat Chem Biol* **12**, 497-503 (2016).
<https://doi.org:10.1038/nchembio.2079>
- 19 Wang, D. *et al.* Antiferroptotic activity of non-oxidative dopamine. *Biochem Biophys Res Commun* **480**, 602-607 (2016).
<https://doi.org:10.1016/j.bbrc.2016.10.099>
- 20 Zhu, S. *et al.* HSPA5 Regulates Ferroptotic Cell Death in Cancer Cells. *Cancer Res* **77**, 2064-2077 (2017). <https://doi.org:10.1158/0008-5472.CAN-16-1979>
- 21 Wu, Z. *et al.* Chaperone-mediated autophagy is involved in the execution of ferroptosis. *Proc Natl Acad Sci U S A* **116**, 2996-3005 (2019).
<https://doi.org:10.1073/pnas.1819728116>
- 22 Tang, D., Chen, X., Kang, R. & Kroemer, G. Ferroptosis: molecular mechanisms and health implications. *Cell Res* **31**, 107-125 (2021).
<https://doi.org:10.1038/s41422-020-00441-1>
- 23 Jiang, D., Niwa, M. & Koong, A. C. Targeting the IRE1alpha-XBP1 branch of the unfolded protein response in human diseases. *Semin Cancer Biol* **33**, 48-56 (2015). <https://doi.org:10.1016/j.semcancer.2015.04.010>
- 24 Zheng, J. *et al.* Sorafenib fails to trigger ferroptosis across a wide range of cancer cell lines. *Cell Death Dis* **12**, 698 (2021). <https://doi.org:10.1038/s41419-021-03998-w>
- 25 Kimata, Y. *et al.* Two regulatory steps of ER-stress sensor Ire1 involving its cluster formation and interaction with unfolded proteins. *J Cell Biol* **179**, 75-86 (2007). <https://doi.org:10.1083/jcb.200704166>
- 26 Lee, K. P. *et al.* Structure of the dual enzyme Ire1 reveals the basis for catalysis and regulation in nonconventional RNA splicing. *Cell* **132**, 89-100 (2008).
<https://doi.org:10.1016/j.cell.2007.10.057>
- 27 Korennykh, A. V. *et al.* The unfolded protein response signals through high-order assembly of Ire1. *Nature* **457**, 687-693 (2009).
<https://doi.org:10.1038/nature07661>
- 28 Bertolotti, A., Zhang, Y., Hendershot, L. M., Harding, H. P. & Ron, D. Dynamic interaction of BiP and ER stress transducers in the unfolded-protein response. *Nat Cell Biol* **2**, 326-332 (2000). <https://doi.org:10.1038/35014014>
- 29 Zhou, J. *et al.* The crystal structure of human IRE1 luminal domain reveals a conserved dimerization interface required for activation of the unfolded protein response. *Proc Natl Acad Sci U S A* **103**, 14343-14348 (2006).
<https://doi.org:10.1073/pnas.0606480103>

- 30 Du, J. *et al.* Identification of proteins interacting with GTP cyclohydrolase I. *Biochem Biophys Res Commun* **385**, 143-147 (2009).
<https://doi.org:10.1016/j.bbrc.2009.05.026>

REVIEWER COMMENTS

Reviewer #1 (Remarks to the Author):

I appreciate that the authors have made some efforts to address my previous comments. However, it is essential to emphasize that certain critical concerns have not been satisfactorily addressed. I will highlight these concerns and provide a more robust statement:

1. The authors assert that the knockdown of XBP1, ATF4, or ATF6 in MDA-MB-231 cells using shRNA suggests that the UPR pathway is not involved in ferroptosis regulation. This finding contradicts current knowledge regarding the role of ER stress and UPR in ferroptosis. Since this observation was limited to MDA-MB-231 cells, it may be cell type-dependent. In other words, the authors claim that IRE1 α 's regulation of ferroptosis through modulating GSH availability is distinct from its canonical UPR functions and other UPR branches. However, the authors have not provided compelling evidence to explain how IRE1 α selectively plays different roles in these two processes.
2. The authors have not adequately addressed why BSO resensitizes IRE1 α -null cells to erastin-induced ferroptosis. Their main claim is that IRE1 α plays a broad-spectrum role in promoting ferroptosis. However, the authors found that IRE1 α -null cells exhibit resistance not only to erastin (an inhibitor of GSH upstream signal) but also to RSL3 (an inhibitor of GSH downstream signal). If this is accurate, the notion that BSO can rescue ferroptosis resistance in IRE1 α KO cells seems implausible.
3. The study heavily relies on the use of erastin. The authors assert that under RSL3-induced ferroptosis, the status of upstream regulators such as SLC7A11-GCLC-GSH can still determine cellular sensitivity to ferroptosis. This conclusion is not supported by existing literature, as RSL3 primarily targets GPX4 and has no significant effects on GSH levels.
4. While the authors have clarified their use of different erastin concentrations in various experiments, these differences may still introduce potential confounding variables that could compromise the validity of their conclusions.
5. The doses of compounds used in this study are generally higher than the IC50 values, which may lead to off-target effects. While we acknowledge that most IC50 values are determined in cell-free systems, the 50-100 fold higher doses employed here could result in unintended effects.

5. Furthermore, the animal study using the kidney I/R model fails to adequately address concerns. This model does not represent a pure ferroptosis-induced damage model, and it is widely accepted that other cell death modalities may be involved in this process. Additionally, the animal study lacks tumor models and does not incorporate other cell death inhibitors or genetic models, limiting its relevance and applicability.

Reviewer #2 (Remarks to the Author):

The authors addressed well most of the points I raised. My first major point was address in part, only. Nevertheless, I recommend for publication of this revised version.

Reviewer #3 (Remarks to the Author):

The authors have revised their manuscript according to my previous comments and addressed most of my concerns. They have added new experiments to show the effect of IRE1 α on ferroptosis in a mouse model of renal ischemia-reperfusion injury, and to test the effect of IRE1 α on ferroptosis in a human renal cell line. They have also added more details on how they selected the dosage of IRE1 α inhibitor and why they used different concentrations of erastin in different experiments. They have corrected the typo in the gene symbol of ACSL4 and explained the difference between Ern1 and IRE1 α . They have also performed additional experiments to characterize the different cell death pathways in their models and showed that neither apoptosis nor autophagy was significantly induced by ferroptosis stimuli or affected by IRE1 α status.

I appreciate the authors' efforts in improving their manuscript and I think their study is interesting and valuable. However, I still have some questions and suggestions that I hope the authors can address before publication:

How did you distinguish the direct and indirect effects of IRE1 α on ferroptosis? Did you consider the possibility that IRE1 α may affect ferroptosis through other downstream factors or pathways?

How did you evaluate the specificity and safety of IRE1 α inhibitors? Did you test the effects of IRE1 α inhibitors on other cellular functions and organs?

How did you explain the different effects of IRE1 α on ferroptosis in normal cells and cancer cells? Did you explore the differential expression and regulation of IRE1 α in different cell types and tissues?

How did you assess the importance and advantage of IRE1 α in ferroptosis regulation compared to other known ferroptosis regulators?

I look forward to reading the authors' response and the final version of their manuscript.

Reviewer #4 (Remarks to the Author):

The Authors have appropriately responded to my initial review.

Following a second read of the manuscript I recommend that 'IRE1 α ' be defined in the abstract (as is done for GPX4, FSP1, etc) when first introduced. Similarly, XBP1 needs to be defined.

I would also recommend that complete (i.e uncropped) images of all immunoblots be included in the supplemental data. I don't have any concerns with the data presented in this submission. However, there are frequent issues with immunoblots data, so full transparency of data (raw image) should be a mandated requirement for publication.

Once these minor issues are addressed I recommend this submission be accepted for publication.

Reviewer #5 (Remarks to the Author):

The authors have discovered non-canonical function of IRE1 α in the down regulation of GCLC and SLC7A11 independent to UPR function. This will significantly contribute to a better understanding of metabolic regulation to prevent ferroptosis. In the revised manuscript, the authors well addressed most of reviewer's questions thereby improving the overall quality of the results. The molecular mechanism of IRE1 α mediated GCLC and SLC7A11 mRNA degradation is more solid. They also have provided more robust in-vivo experimental results with GEMM model strongly supporting therapeutic potential of IRE1 α inhibition against ferroptosis mediated pathogenic condition. Overall, this manuscript meets enough quality to be published in this journal.

I have one minor suggestion for the authors: Please correct the order of supplementary figures, as they are not matched well with the order of description in the manuscript.

Responses to Reviewer #1:

I appreciate that the authors have made some efforts to address my previous comments. However, it is essential to emphasize that certain critical concerns have not been satisfactorily addressed. I will highlight these concerns and provide a more robust statement:

*1. The authors assert that the knockdown of XBP1, ATF4, or ATF6 in MDA-MB-231 cells using shRNA **suggests that the UPR pathway is not involved in ferroptosis regulation**. This finding contradicts current knowledge regarding the role of ER stress and UPR in ferroptosis. Since **this observation was limited to MDA-MB-231 cells**, it may be cell type-dependent. In other words, the authors claim that IRE1 α 's regulation of ferroptosis through modulating GSH availability is distinct from its canonical UPR functions and other UPR branches. However, the authors have not provided compelling evidence to explain how IRE1 α **selectively plays different roles** in these two processes.*

As shown in **Fig. 2**, our data indicates that the canonical UPR pathways (ER stress induced activation of XBP1, ATF4, and ATF6) are not involved in ferroptosis in MDA-MB-231 and Panc-1 cell lines (**two distinct cell types: triple negative breast cancer and pancreatic cancer**) induced by erastin or cystine starvation (**two distinct treatments to target the GSH pathway**). In our study, we showed that, neither erastin treatment nor cystine starvation induces the UPR as determined by the lack of increased IRE1 α phosphorylation, PERK phosphorylation and ATF6 proteolytic cleavage, all of which are classical markers of UPR activation and are induced by the classical ER stress/UPR activator thapsigargin (Tg) in the same cells. In contrast, we and others have shown that ATF4 is induced by erastin or cystine starvation. However, this is through GCN2 signaling which has been previously defined as part of the Integrated Stress Response (ISR) pathway, but not through PERK signaling as part of the canonical UPR (**Fig. 2 and Supplementary Fig. 3A**). IRE1 α 's involvement in regulating ferroptosis occurs primarily through its endonuclease activity (RIDD) which degrades the mRNA of key proteins involved in glutathione synthesis. This mechanism of IRE1 α activity is separate from its role in regulating the canonical UPR pathway and to our knowledge, has not been previously reported.

Therefore, as demonstrated in multiple cell types, **ferroptosis induced by erastin or cystine starvation and the UPR induced by unfolded/misfolded proteins in the endoplasmic reticulum (ER stress) are distinct cellular processes**. In the absence of UPR activation and ER stress, our data focuses on identifying the role of the **basal activities** of the 3 major UPR branches. We show here that in contrast to the **basal activity** of IRE1 α , the **basal activities** of PERK and ATF6 do not influence cellular sensitivity to erastin/cystine starvation-induced ferroptosis (**Fig. 2B and D**). Therefore, our findings are not in conflict with the effects of PERK and ATF6 in **activated UPR** and represent new findings that will be of significant interest to other investigators in the field.

2. The authors have not adequately addressed why BSO resensitizes IRE1 α -null cells to erastin-induced ferroptosis. Their main claim is that IRE1 α plays a broad-spectrum role in promoting ferroptosis. However, the authors found that IRE1 α -null cells exhibit resistance not only to erastin (an inhibitor of GSH upstream signal) but also to RSL3 (an inhibitor of GSH downstream signal). If this is accurate, the notion that BSO can rescue ferroptosis resistance in IRE1 α KO cells seems implausible.

After careful evaluation of the reviewer's comments on our finding that BSO can rescue ferroptosis resistance in IRE1 α KO cells, we reiterate our explanation based on both previous and new data included in the revised manuscript - **The resistance of IRE1 α KO cells to RSL3 (an inhibitor of GSH downstream signaling) is also caused by elevated GSH levels in the IRE1 α KO cells which can overcome GPX4 inhibition by RSL3.**

In our previous response to the reviewer, we discussed that **IRE1 α does not play a broad-spectrum role in ferroptosis regulation.** Instead, our study focuses on the effect of IRE1 α on the SLC7A11-GCLC-GSH pathway as a defense mechanism against ferroptosis. Ferroptosis and the associated metabolic pathways are regulated through various interacting steps that are not necessarily linear. We provide additional data (**the new Supplementary Fig. 5**) demonstrating that the resistance of IRE1 α -KO cells to RSL3 is primarily due to IRE1 α 's suppressive effect on SLC7A11/GCLC and GSH synthesis. Since the dose of RSL3 used in our study does not completely inhibit GPX4 activity (**Supplementary Fig. 5A**), cells treated with RSL3 may still remain sensitive to GSH depletion (see additional elaboration in our response to Point #3). We also generated new data to demonstrate that modulating cellular SLC7A11 levels (a signal upstream of GPX4 inhibition by RSL3) alters cellular sensitivity to RSL3 (**Supplementary Fig. 5C-G**). **Therefore, the data with RSL3 are an extension of our core findings for IRE1 α 's regulation of SLC7A11/GCLC, instead of a different mechanism through which IRE1 α regulates ferroptosis.**

Therefore, since BSO inhibits GCLC and GSH synthesis and reduces the elevated GSH level observed in IRE1 α KO cells, it can rescue the resistance of these cells to ferroptosis.

3. The study heavily relies on the use of erastin. The authors assert that under RSL3-induced ferroptosis, the status of upstream regulators such as SLC7A11-GCLC-GSH can still determine cellular sensitivity to ferroptosis. This conclusion is not supported by existing literature, as RSL3 primarily targets GPX4 and has no significant effects on GSH levels.

Because the major finding of our study is that IRE1 α regulates cellular sensitivity to ferroptosis by inhibiting the SLC7A11/GCLC/GSH pathway, we mainly used pharmacological inhibitor of SLC7A11, erastin, to induce ferroptosis. **However, we also included cystine starvation as a different approach to target the same pathway in our core experiments (Fig. 1A and E; Fig. 2A; Supplementary Fig. 3A).** Both

approaches induced the expected downstream effects in our assays, demonstrating that the induced ferroptosis was due to inhibition of cystine uptake.

As stated in our response to Point #2, in the revised manuscript we provided additional data to demonstrate that (1) The dose of RSL3 (1 μ M) used in our study does not completely inhibit GPX4 activity (**Supplementary Fig. 5A**), therefore the residual GPX4 activity will remain sensitive to the cellular GSH level regulated by the SLC7A11-GCLC-GSH pathway, and (2) in the CTRP database consisting of 860 cancer cell lines (CCLs), higher SLC7A11 (as well as GCLC) expression level strongly correlates with resistance to RSL3 (**Supplementary Fig. 5B**). Furthermore, we performed additional experiments to demonstrate that SLC7A11 expression level indeed determines cellular sensitivity to RSL3. We showed that in both MDA-MB-231 and Panc-1 cells (multiple cell lines) SLC7A11 overexpression led to significant resistance to RSL3 (**Supplementary Fig. 5C, D and G**). Conversely, when SLC7A11 expression was inhibited, both types of cells became more sensitive to RSL3 (**Supplementary Fig. 5E, F and G**). **These additional data support the conclusion that activity of the SLC7A11-GCLC-GSH pathway modulates sensitivity to RSL3 in our experimental settings.** We agree that RSL3 has no effect on GSH levels. **Our main assertion is that GSH level is determined by the IRE1 α -SLC7A11-GCLC pathway.**

In addition to our own study, multiple recent studies from other groups also support that the status of upstream regulators such as SLC7A11-GCLC-GSH can determine cellular sensitivity to RSL3-induced ferroptosis. Examples are shown below.

Nat Commun. 2021 Mar 11;12(1):1589¹: SLC7A11 deficiency promotes, whereas SLC7A11 overexpression inhibits, RSL3-induced lipid peroxidation and ferroptosis (Fig. 2a–f and Supplementary Fig. 3b–e).

Cell Rep. 2019 Feb 5;26(6):1544-1556.e8. ²: A Genome-wide haploid genetic screen identifies ABCC1/MRP1-mediated GSH efflux sensitizes cells to RSL3-induced ferroptosis in H1299 and U2OS cells (Fig. 4A).

J Hepatol. 2023 Aug;79(2):362-377. ³: ATF4-deficient mouse hepatocytes are more sensitive to RSL3-induced ferroptosis compared to wild-type cells, due to reduced SLC7A11 expression. Reconstitution of SLC7A11 expression in ATF4-deficient hepatocytes provided protection against RSL3-induced ferroptosis (Fig. 3F).

Redox Biol. 2023 Sep;65:102833. ⁴: HT-29 cells became significantly more resistant to RSL3-induced ferroptotic cell death following ectopic SLC7A11 overexpression (Fig. 3P).

4. While the authors have clarified their use of different erastin concentrations in various experiments, these differences may still introduce potential confounding variables that could compromise the validity of their conclusions.

To further improve the consistency of erastin concentration used in the *in vitro* assays, we repeated the flow cytometry and GSH assays with 2.5 μ M erastin (Fig. 1F; Fig. 3A; Fig. 5B; Supplementary Fig. 4B) to be consistent with the rest of the manuscript. Furthermore, we also evaluated ferroptosis induction with a wide range of erastin

concentrations and treatment duration to include all the conditions used in our study (**Supplementary Fig. 1D; Supplementary Fig. 2; Supplementary Fig. 3C and D**).

5. The doses of compounds used in this study are generally higher than the IC₅₀ values, which may lead to off-target effects. While we acknowledge that most IC₅₀ values are determined in cell-free systems, the 50-100 fold higher doses employed here could result in unintended effects.

We acknowledge that the doses of compounds used in this study may have unintended off-target effects but this criticism may be used for **any** study using **any** range of drug concentration. With the long history and widespread acceptability for using 4 μ 8c in the UPR research field, one should be able to draw meaningful conclusions based upon the data shown. **For our studies, we have used the same concentrations and identical experimental conditions as other UPR researchers.**

In the original study that identified 4 μ 8c (CB5305630)⁵, although the cell-free IC₅₀ was defined as 61.58 nM (Fig. 1A-D, left panel below), in the cell-based assays (RT-PCR) **32 μ M** of 4 μ 8c was required to achieve complete XBP1s inhibition (Fig. 5A-B, right panel below). **These data clearly demonstrate the difference between cell-free and cell-based activities of small molecule compounds.** Investigators in this field standardly use the doses of 4 μ 8c that we used in our study and it is generally accepted that these concentrations achieve potent and selective inhibition of XBP1s.

In the UPR research field, **30 – 100 μ M** 4 μ 8c is standardly used in **cell-based assays** to inhibit IRE1 α . Some examples can be found in numerous recent high-impact publications as shown below.

- **Cell Rep.** 2023 Sep 26;42(9):113130. **IMR90 cells. 50 μ M** 4 μ 8c⁶.
- **J Exp Med.** 2023 Nov 6; 220(11): e20230106. **16HBE cells. 35 μ M** 4 μ 8c⁷.
- **Nat Commun.** 2023; 14: 5414. **Primary human trophoblasts. 50 μ M** 4 μ 8c⁸.
- **EMBO J.** 2022 Aug; 41(16): e110501. **Mouse neurons. 40 μ M** 4 μ 8c⁹.

- *eLife*. 2022; 11: e75072. Myometrial cells. **80 μM 4 μ8c** ¹⁰.
- *Nat Cell Biol*. 2022 Sep; 24(9): 1422–1432. A375 human melanoma cells. **50 μM 4 μ8c** ¹¹.
- *J Clin Invest*. 2022 Sep 1; 132(17): e153519. Mouse colonoids. **50 μM 4 μ8c** ¹².
- *Nat Cell Biol*. 2023; 25(5): 726–739. Primary macrophages. **100 μM 4 μ8c** ¹³.

5. Furthermore, the animal study using the kidney I/R model fails to adequately address concerns. This model does not represent a pure ferroptosis-induced damage model, and it is widely accepted that other cell death modalities may be involved in this process. Additionally, the animal study lacks tumor models and does not incorporate other cell death inhibitors or genetic models, limiting its relevance and applicability.

Our study reported a previously unknown role for IRE1 α in regulating ferroptosis. The immediate implication of this work is in the regulation of normal tissue damage. There are currently no ferroptosis inducers in clinical use for cancer therapy whereas IRE1 α inhibitors in clinical trials (e.g., ORIN1001 in NCT03950570) would have obvious relevance to protect normal tissues from ferroptosis-induced damage. Therefore, the kidney I/R model is highly relevant and suitable for our study. ***This model has been widely used by the ferroptosis research field to study the contribution of ferroptosis to tissue damage in vivo*** ¹⁴⁻¹⁶. In the post-I/R kidney tissues in our study, while we detected robust induction of lipid peroxidation/ferroptosis marker 4-HNE (Fig. 6 and 7), we did not detect noticeable apoptosis and autophagy markers, cleaved caspase 3 and processing of LC3B, respectively (Supplementary Fig. 9E) in the injured kidneys. These results suggest that ***ferroptosis is the major type of cell death induced by I/R in the kidney***, which is consistent with previous reports ^{15,17,18}.

As ferroptosis is a major contributor to cell death associated with ischemia-reperfusion injuries, ferroptosis inhibition is a potential therapeutic approach for the treatment of conditions associated with ischemic damage, including brain stroke, ischemic heart disease and injuries to the liver and kidney, which collectively represent a major impact on global health (X. Jiang et al., *Nat. Reviews Mol. Cell Biology*, 2021 ¹⁹). Small molecule IRE1 α inhibitors have been under development for more than a decade and are currently being tested in a Phase I/II clinical trial (NCT03950570). The **clinical significance and impact** of our study are substantial as advancements in understanding the clinical applications of IRE1 α inhibitors are critical to developing effective therapeutic approaches based upon modulating ferroptosis in ischemia-reperfusion injuries.

The reviewer also pointed out that our study lacks tumor models therefore limiting its relevance and applicability. As ferroptosis is recognized as a tumor suppressor mechanism and also induced by cancer therapeutic treatment (e.g. radiation therapy), it would be logical to test activating IRE1 α 's RIDD activity to enhance ferroptosis induction in tumor cells. ***However, the development of IRE1 α activators or ferroptosis inducers for cancer therapy is well beyond the scope of this study.*** To date, no chemical compounds have been identified to specifically activate the RIDD activity of IRE1 α nor are there been any ferroptosis inducers promising enough for clinical testing.

We strongly believe that any additional animal studies, including tumor models, are not required to substantiate the significance of our findings. **We have discussed this point with the Editor who has agreed that additional animal studies are not needed for further consideration of this manuscript.** Given the additional data that are provided here and in the prior revision, we appreciate this reviewer's understanding of our position.

Responses to Reviewer #2:

The authors addressed well most of the points I raised. My first major point was address in part, only. Nevertheless, I recommend for publication of this revised version.

We appreciate Reviewer #2's positive feedback on our additional work to improve the current study.

Responses to Reviewer #3:

The authors have revised their manuscript according to my previous comments and addressed most of my concerns. They have added new experiments to show the effect of IRE1 α on ferroptosis in a mouse model of renal ischemia-reperfusion injury, and to test the effect of IRE1 α on ferroptosis in a human renal cell line. They have also added more details on how they selected the dosage of IRE1 α inhibitor and why they used different concentrations of erastin in different experiments. They have corrected the typo in the gene symbol of ACSL4 and explained the difference between Ern1 and IRE1 α . They have also performed additional experiments to characterize the different cell death pathways in their models and showed that neither apoptosis nor autophagy was significantly induced by ferroptosis stimuli or affected by IRE1 α status.

I appreciate the authors' efforts in improving their manuscript and I think their study is interesting and valuable. However, I still have some questions and suggestions that I hope the authors can address before publication:

1. How did you distinguish the direct and indirect effects of IRE1 α on ferroptosis? Did you consider the possibility that IRE1 α may affect ferroptosis through other downstream factors or pathways?

We thank the reviewer for bringing up this important subject, as in this study we fully considered these possibilities and addressed the key questions experimentally. IRE1 α is well-known to regulate its downstream functions through activating the transcription factor XBP1 by non-canonical splicing of XBP1 mRNA. As shown in **Fig. 1G**, we demonstrated that the acquired resistance to ferroptosis through IRE1 α loss is NOT due to the loss of the activation of XBP1 by IRE1 α , as forced overexpression of the activated form of XBP1 (XBP1s) failed to re-sensitize the IRE1 α -null cells to ferroptosis. As shown in **Fig. 2B**, knockdown of XBP1 expression did not impact cellular sensitivity

to erastin-induced ferroptosis. IRE1 α also activates JNK through TRAF2. However, as shown in **Supplementary Fig. 9C**, in HK-2 cells, ferroptosis induction through either hypoxia-reperfusion or RSL3 treatment did not activate JNK. Based on these observations, we reasoned that the most likely mechanism is that IRE1 α directly regulates cellular sensitivity to ferroptosis through its RIDD (regulated Ire1-dependent decay) activity. Our mechanistic studies demonstrated that through RIDD, IRE1 α indeed down-regulates multiple key components of the cellular ferroptosis defense machinery, including GCLC and SLC7A11.

2. How did you evaluate the specificity and safety of IRE1 α inhibitors? Did you test the effects of IRE1 α inhibitors on other cellular functions and organs?

We thank the reviewer for pointing out this consideration important for the therapeutic applications of IRE1 α inhibitors. The IRE1 α inhibitors (4 μ 8c and MKC9989) used in our study are salicylaldehyde derivatives that specifically inhibit the RNase activity of IRE1 α ^{5,20,21}. The specificity of 4 μ 8c used in our *in vivo* studies was demonstrated by the fact that the RNase activity of the IRE1 α -related endoribonuclease, RNase L, was unaffected by 4 μ 8c⁵. This initial finding was further confirmed by additional studies²². The *in vivo* safety as well as general effects on normal organs of this class of compounds were well characterized in a recent study by *Zhao et al.*²³ In this study, a 4th-generation salicylaldehyde class inhibitor MKC8866 was tested *in vivo*. While MKC8866 suppressed MYC-overexpressing tumor growth *in vivo*, no tissue damage was detected in normal organs including liver, pancreas, kidney, lung, heart, and small intestine. MKC8866 didn't induce apoptosis in liver or pancreas, either. Furthermore, food intake and glucose levels were unchanged during treatment. These findings suggested that this class of IRE1 α inhibitors have minimal *in vivo* toxicity in normal tissues and organs.

3. How did you explain the different effects of IRE1 α on ferroptosis in normal cells and cancer cells? Did you explore the differential expression and regulation of IRE1 α in different cell types and tissues?

In our study, we found that the role of IRE1 α in regulating ferroptosis is shared by both cancer cells (MDA-MB-231 triple-negative breast cancer and Panc-1 pancreatic cancer cells) and normal cells (mouse embryonic fibroblasts and HK-2 normal human kidney epithelial cells) as shown in **Fig. 1A-C** and **Supplementary Fig. 9D**. We also showed that this function of IRE1 α is evolutionarily conserved in *C. elegans* (**Fig. 4** and **Supplementary Fig. 7**). More importantly, this function of IRE1 α requires only its basal activity, as during ferroptosis induction by either erastin treatment, cystine starvation or RSL3 treatment no unfolded protein response (UPR) activation was detected (**Fig. 2A** and **Supplementary Fig. 4G**). Therefore, we believe that this novel activity of IRE1 α in regulating ferroptosis sensitivity is a basal function that is separate from IRE1 α 's role in stress responses. However, as pathological processes like ischemic reperfusion injury may depend on this activity of IRE1 α , pharmacological inhibition of IRE1 α is a promising therapeutic strategy to alleviate the associated damages.

4. How did you assess the importance and advantage of IRE1 α in ferroptosis regulation compared to other known ferroptosis regulators?

Since the discovery of ferroptosis, there have been a plethora of cellular ferroptosis regulators identified. The best characterized regulators include those responsible for PUFA synthesis (e.g. ACSL4, LPCAT3 and ALOXs), iron metabolism, and anti-ferroptotic defense mechanisms (e.g. SLC7A11, GCLC, GPX4, FSP1, DHODH, and GCH-1). These regulators primarily regulate the transcription or post-translational modifications of core ferroptosis proteins. In contrast to these canonical regulatory mechanisms, we are one of the first to report that IRE1 α mainly serves as **a surveillance mechanism to ensure the inducibility of ferroptosis through the cleavage (via RIDD) and downregulation of mRNA encoding key defense regulators including SLC7A11 and GCLC**. As this regulatory function of IRE1 α acts through modulating the cellular level of glutathione, it is likely to link to IRE1 α 's broader roles in regulating cellular redox balance and anti-infection²⁴. We believe our novel findings will promote the scientific research community to further explore IRE1 α 's role in related cellular functions and extend our understanding of ferroptosis mechanisms.

I look forward to reading the authors' response and the final version of their manuscript.

We appreciate the reviewer's valuable suggestions that further clarify and strengthen our conclusions. However, due to page limit we are not able to include these additional discussions in the manuscript. We will include the detailed discussion in a future commentary or review article referring to the current study. Furthermore, as responses to reviewer comments will also be published online by Nature Communications, the audience will have access to our discussion.

Responses to Reviewer #4:

The Authors have appropriately responded to my initial review.

Following a second read of the manuscript I recommend that 'IRE1 α ' be defined in the abstract (as is done for GPX4, FSP1, etc) when first introduced. Similarly, XBP1 needs to be defined.

We thank the reviewer for this valuable suggestion. In the revised manuscript we defined the full name of "IRE1 α " in the abstract and "XBP1" when it shows up the first time.

I would also recommend that complete (i.e uncropped) images of all immunoblots be included in the supplemental data. I don't have any concerns with the data presented in this submission. However, there are frequent issues with immunoblots data, so full transparency of data (raw image) should be a mandated requirement for publication.

Complete and uncropped images of immunoblots used in this manuscript are included in the Source Data file submitted together with the manuscript files.

Once these minor issues are addressed I recommend this submission be accepted for publication.

Responses to Reviewer #5:

The authors have discovered non-canonical function of IRE1 α in the down regulation of GCLC and SLC7A11 independent to UPR function. This will significantly contribute to a better understanding of metabolic regulation to prevent ferroptosis. In the revised manuscript, the authors well addressed most of reviewer's questions thereby improving the overall quality of the results. The molecular mechanism of IRE1 α mediated GCLC and SLC7A11 mRNA degradation is more solid. They also have provided more robust in-vivo experimental results with GEMM model strongly supporting therapeutic potential of IRE1 α inhibition against ferroptosis mediated pathogenic condition. Overall, this manuscript meets enough quality to be published in this journal.

I have one minor suggestion for the authors: Please correct the order of supplementary figures, as they are not matched well with the order of description in the manuscript.

We thank the reviewer for reminding us of the order when citing some of the Supplementary Figures. In the revised manuscript we have performed rigorous proofreading to make sure that the cited figure numbers match the corresponding figures. As there are a large number of supplementary figures (10 in total) in the current submission to support both the Results and Discussion sections, sometimes it is technically difficult to maintain a complete linear order when citing these figure panels in the manuscript text. We appreciate the reviewer's understanding of our position.

References:

- 1 Zhang, Y. *et al.* mTORC1 couples cyst(e)ine availability with GPX4 protein synthesis and ferroptosis regulation. *Nat Commun* **12**, 1589 (2021). <https://doi.org:10.1038/s41467-021-21841-w>
- 2 Cao, J. Y. *et al.* A Genome-wide Haploid Genetic Screen Identifies Regulators of Glutathione Abundance and Ferroptosis Sensitivity. *Cell Rep* **26**, 1544-1556 e1548 (2019). <https://doi.org:10.1016/j.celrep.2019.01.043>
- 3 He, F. *et al.* ATF4 suppresses hepatocarcinogenesis by inducing SLC7A11 (xCT) to block stress-related ferroptosis. *J Hepatol* **79**, 362-377 (2023). <https://doi.org:10.1016/j.jhep.2023.03.016>
- 4 Saini, K. K. *et al.* Loss of PERK function promotes ferroptosis by downregulating SLC7A11 (System Xc(-)) in colorectal cancer. *Redox Biol* **65**, 102833 (2023). <https://doi.org:10.1016/j.redox.2023.102833>

- 5 Cross, B. C. *et al.* The molecular basis for selective inhibition of unconventional mRNA splicing by an IRE1-binding small molecule. *Proc Natl Acad Sci U S A* **109**, E869-878 (2012). <https://doi.org:10.1073/pnas.1115623109>
- 6 Takasugi, M. *et al.* CD44 correlates with longevity and enhances basal ATF6 activity and ER stress resistance. *Cell Rep* **42**, 113130 (2023). <https://doi.org:10.1016/j.celrep.2023.113130>
- 7 Moniruzzaman, M. *et al.* Interleukin-22 suppresses major histocompatibility complex II in mucosal epithelial cells. *J Exp Med* **220** (2023). <https://doi.org:10.1084/jem.20230106>
- 8 Jash, S. *et al.* Cis P-tau is a central circulating and placental etiologic driver and therapeutic target of preeclampsia. *Nat Commun* **14**, 5414 (2023). <https://doi.org:10.1038/s41467-023-41144-6>
- 9 Wolzak, K. *et al.* Neuron-specific translational control shift ensures proteostatic resilience during ER stress. *EMBO J* **41**, e110501 (2022). <https://doi.org:10.15252/embj.2021110501>
- 10 Chen, L. *et al.* Interleukin-33 regulates the endoplasmic reticulum stress of human myometrium via an influx of calcium during initiation of labor. *Elife* **11** (2022). <https://doi.org:10.7554/eLife.75072>
- 11 Verma, S. *et al.* NRF2 mediates melanoma addiction to GCDH by modulating apoptotic signalling. *Nat Cell Biol* **24**, 1422-1432 (2022). <https://doi.org:10.1038/s41556-022-00985-x>
- 12 Grey, M. J. *et al.* The epithelial-specific ER stress sensor ERN2/IRE1beta enables host-microbiota crosstalk to affect colon goblet cell development. *J Clin Invest* **132** (2022). <https://doi.org:10.1172/JCI153519>
- 13 Ji, Y. *et al.* SEL1L-HRD1 endoplasmic reticulum-associated degradation controls STING-mediated innate immunity by limiting the size of the activable STING pool. *Nat Cell Biol* **25**, 726-739 (2023). <https://doi.org:10.1038/s41556-023-01138-4>
- 14 Friedmann Angeli, J. P. *et al.* Inactivation of the ferroptosis regulator Gpx4 triggers acute renal failure in mice. *Nat Cell Biol* **16**, 1180-1191 (2014). <https://doi.org:10.1038/ncb3064>
- 15 Linkermann, A. *et al.* Synchronized renal tubular cell death involves ferroptosis. *Proc Natl Acad Sci U S A* **111**, 16836-16841 (2014). <https://doi.org:10.1073/pnas.1415518111>
- 16 Lee, H. *et al.* Energy-stress-mediated AMPK activation inhibits ferroptosis. *Nat Cell Biol* **22**, 225-234 (2020). <https://doi.org:10.1038/s41556-020-0461-8>
- 17 Tonnus, W. & Linkermann, A. The in vivo evidence for regulated necrosis. *Immunol Rev* **277**, 128-149 (2017). <https://doi.org:10.1111/imr.12551>
- 18 Tonnus, W. *et al.* Dysfunction of the key ferroptosis-surveilling systems hypersensitizes mice to tubular necrosis during acute kidney injury. *Nat Commun* **12**, 4402 (2021). <https://doi.org:10.1038/s41467-021-24712-6>
- 19 Jiang, X., Stockwell, B. R. & Conrad, M. Ferroptosis: mechanisms, biology and role in disease. *Nat Rev Mol Cell Biol* **22**, 266-282 (2021). <https://doi.org:10.1038/s41580-020-00324-8>

- 20 Volkmann, K. *et al.* Potent and selective inhibitors of the inositol-requiring enzyme 1 endoribonuclease. *J Biol Chem* **286**, 12743-12755 (2011). <https://doi.org:10.1074/jbc.M110.199737>
- 21 Sanches, M. *et al.* Structure and mechanism of action of the hydroxy-aryl-aldehyde class of IRE1 endoribonuclease inhibitors. *Nat Commun* **5**, 4202 (2014). <https://doi.org:10.1038/ncomms5202>
- 22 Cairrao, F. *et al.* Pumilio protects Xbp1 mRNA from regulated Ire1-dependent decay. *Nat Commun* **13**, 1587 (2022). <https://doi.org:10.1038/s41467-022-29105-X>
- 23 Zhao, N. *et al.* Pharmacological targeting of MYC-regulated IRE1/XBP1 pathway suppresses MYC-driven breast cancer. *J Clin Invest* **128**, 1283-1299 (2018). <https://doi.org:10.1172/JCI95873>
- 24 Guimaraes, E. S. *et al.* The endoplasmic reticulum stress sensor IRE1alpha modulates macrophage metabolic function during *Brucella abortus* infection. *Front Immunol* **13**, 1063221 (2022). <https://doi.org:10.3389/fimmu.2022.1063221>

REVIEWERS' COMMENTS

Reviewer #1 (Remarks to the Author):

I genuinely appreciate the author's additional efforts to address the comments. However, there are still some lingering concerns that I would like to highlight:

1. The authors suggest that the unfolded protein response (UPR) is not activated during ferroptosis. However, this finding appears to conflict with previous studies. The unfolded protein response is a vital cellular mechanism responsible for maintaining ER protein homeostasis, and it's well-established that ER stress plays a significant role in initiating ferroptosis (PMID: 36747055).
2. In this study, the introduction of PANC1 cells as a new cell model is noteworthy. However, it's worth noting that several important knockout experiments (e.g., ATF4, XBP1, and ATF6 KO) were not repeated in this cell line. Additionally, PANC1 exhibits very low IRE1 α expression (Fig. S1), which might suggest increased resistance to ferroptosis. Nevertheless, numerous studies have indicated PANC1's susceptibility to ferroptosis (PMID: 32241947), which raises questions about the author's hypothesis. Therefore, the decision to include the PANC1 cell line might warrant further consideration, especially given that this study primarily utilized only two cancer cell lines.
3. The experiment employed erastin at a concentration of 2.5 μ M. However, cell death assays indicated no significant cell death at this dose within a 12-hour timeframe (Fig. S3C). Consequently, this concentration may not be ideal for assessing other protein markers after 12 hours of erastin treatment (Fig. S3G).
4. The endogenous expression of SLC7A11 in PANC1 and MDA-MB-231 cells is relatively low in knockin experiments but exhibits elevated baseline expression in knockout experiments (Fig. S5G).
5. While all key in vitro data were obtained from cancer cell lines, the study sought to validate the hypothesis using kidney models. However, it's important to note that this approach might be considered less logically robust, as tumor and non-tumor models often exhibit distinct mechanisms related to the IRE1 pathway.
6. The expression of LC3-2 was induced by RSL3 in vitro in HK2 cells, but it was not induced by I/R in kidney tissue. These findings suggest that the kidney in vivo model may not be the most suitable choice for confirming the author's hypothesis.

Responses to Reviewer #1:

I appreciate that the authors have made some efforts to address my previous comments. However, it is essential to emphasize that certain critical concerns have not been satisfactorily addressed. I will highlight these concerns and provide a more robust statement:

*1. The authors assert that the knockdown of XBP1, ATF4, or ATF6 in MDA-MB-231 cells using shRNA **suggests that the UPR pathway is not involved in ferroptosis regulation**. This finding contradicts current knowledge regarding the role of ER stress and UPR in ferroptosis. Since **this observation was limited to MDA-MB-231 cells**, it may be cell type-dependent. In other words, the authors claim that IRE1 α 's regulation of ferroptosis through modulating GSH availability is distinct from its canonical UPR functions and other UPR branches. However, the authors have not provided compelling evidence to explain how IRE1 α **selectively plays different roles** in these two processes.*

As shown in **Fig. 2**, our data indicates that the canonical UPR pathways (ER stress induced activation of XBP1, ATF4, and ATF6) are not involved in ferroptosis in MDA-MB-231 and Panc-1 cell lines (**two distinct cell types: triple negative breast cancer and pancreatic cancer**) induced by erastin or cystine starvation (**two distinct treatments to target the GSH pathway**). In our study, we showed that, neither erastin treatment nor cystine starvation induces the UPR as determined by the lack of increased IRE1 α phosphorylation, PERK phosphorylation and ATF6 proteolytic cleavage, all of which are classical markers of UPR activation and are induced by the classical ER stress/UPR activator thapsigargin (Tg) in the same cells. In contrast, we and others have shown that ATF4 is induced by erastin or cystine starvation. However, this is through GCN2 signaling which has been previously defined as part of the Integrated Stress Response (ISR) pathway, but not through PERK signaling as part of the canonical UPR (**Fig. 2 and Supplementary Fig. 3A**). IRE1 α 's involvement in regulating ferroptosis occurs primarily through its endonuclease activity (RIDD) which degrades the mRNA of key proteins involved in glutathione synthesis. This mechanism of IRE1 α activity is separate from its role in regulating the canonical UPR pathway and to our knowledge, has not been previously reported.

Therefore, as demonstrated in multiple cell types, **ferroptosis induced by erastin or cystine starvation and the UPR induced by unfolded/misfolded proteins in the endoplasmic reticulum (ER stress) are distinct cellular processes**. In the absence of UPR activation and ER stress, our data focuses on identifying the role of the **basal activities** of the 3 major UPR branches. We show here that in contrast to the **basal activity** of IRE1 α , the **basal activities** of PERK and ATF6 do not influence cellular sensitivity to erastin/cystine starvation-induced ferroptosis (**Fig. 2B and D**). Therefore, our findings are not in conflict with the effects of PERK and ATF6 in **activated UPR** and represent new findings that will be of significant interest to other investigators in the field.

2. The authors have not adequately addressed why BSO resensitizes IRE1 α -null cells to erastin-induced ferroptosis. Their main claim is that IRE1 α plays a broad-spectrum role in promoting ferroptosis. However, the authors found that IRE1 α -null cells exhibit resistance not only to erastin (an inhibitor of GSH upstream signal) but also to RSL3 (an inhibitor of GSH downstream signal). If this is accurate, the notion that BSO can rescue ferroptosis resistance in IRE1 α KO cells seems implausible.

After careful evaluation of the reviewer's comments on our finding that BSO can rescue ferroptosis resistance in IRE1 α KO cells, we reiterate our explanation based on both previous and new data included in the revised manuscript - **The resistance of IRE1 α KO cells to RSL3 (an inhibitor of GSH downstream signaling) is also caused by elevated GSH levels in the IRE1 α KO cells which can overcome GPX4 inhibition by RSL3.**

In our previous response to the reviewer, we discussed that **IRE1 α does not play a broad-spectrum role in ferroptosis regulation.** Instead, our study focuses on the effect of IRE1 α on the SLC7A11-GCLC-GSH pathway as a defense mechanism against ferroptosis. Ferroptosis and the associated metabolic pathways are regulated through various interacting steps that are not necessarily linear. We provide additional data (**the new Supplementary Fig. 5**) demonstrating that the resistance of IRE1 α -KO cells to RSL3 is primarily due to IRE1 α 's suppressive effect on SLC7A11/GCLC and GSH synthesis. Since the dose of RSL3 used in our study does not completely inhibit GPX4 activity (**Supplementary Fig. 5A**), cells treated with RSL3 may still remain sensitive to GSH depletion (see additional elaboration in our response to Point #3). We also generated new data to demonstrate that modulating cellular SLC7A11 levels (a signal upstream of GPX4 inhibition by RSL3) alters cellular sensitivity to RSL3 (**Supplementary Fig. 5C-G**). **Therefore, the data with RSL3 are an extension of our core findings for IRE1 α 's regulation of SLC7A11/GCLC, instead of a different mechanism through which IRE1 α regulates ferroptosis.**

Therefore, since BSO inhibits GCLC and GSH synthesis and reduces the elevated GSH level observed in IRE1 α KO cells, it can rescue the resistance of these cells to ferroptosis.

3. The study heavily relies on the use of erastin. The authors assert that under RSL3-induced ferroptosis, the status of upstream regulators such as SLC7A11-GCLC-GSH can still determine cellular sensitivity to ferroptosis. This conclusion is not supported by existing literature, as RSL3 primarily targets GPX4 and has no significant effects on GSH levels.

Because the major finding of our study is that IRE1 α regulates cellular sensitivity to ferroptosis by inhibiting the SLC7A11/GCLC/GSH pathway, we mainly used pharmacological inhibitor of SLC7A11, erastin, to induce ferroptosis. **However, we also included cystine starvation as a different approach to target the same pathway in our core experiments (Fig. 1A and E; Fig. 2A; Supplementary Fig. 3A).** Both

approaches induced the expected downstream effects in our assays, demonstrating that the induced ferroptosis was due to inhibition of cystine uptake.

As stated in our response to Point #2, in the revised manuscript we provided additional data to demonstrate that (1) The dose of RSL3 (1 μ M) used in our study does not completely inhibit GPX4 activity (**Supplementary Fig. 5A**), therefore the residual GPX4 activity will remain sensitive to the cellular GSH level regulated by the SLC7A11-GCLC-GSH pathway, and (2) in the CTRP database consisting of 860 cancer cell lines (CCLs), higher SLC7A11 (as well as GCLC) expression level strongly correlates with resistance to RSL3 (**Supplementary Fig. 5B**). Furthermore, we performed additional experiments to demonstrate that SLC7A11 expression level indeed determines cellular sensitivity to RSL3. We showed that in both MDA-MB-231 and Panc-1 cells (multiple cell lines) SLC7A11 overexpression led to significant resistance to RSL3 (**Supplementary Fig. 5C, D and G**). Conversely, when SLC7A11 expression was inhibited, both types of cells became more sensitive to RSL3 (**Supplementary Fig. 5E, F and G**). **These additional data support the conclusion that activity of the SLC7A11-GCLC-GSH pathway modulates sensitivity to RSL3 in our experimental settings.** We agree that RSL3 has no effect on GSH levels. **Our main assertion is that GSH level is determined by the IRE1 α -SLC7A11-GCLC pathway.**

In addition to our own study, multiple recent studies from other groups also support that the status of upstream regulators such as SLC7A11-GCLC-GSH can determine cellular sensitivity to RSL3-induced ferroptosis. Examples are shown below.

Nat Commun. 2021 Mar 11;12(1):1589¹: SLC7A11 deficiency promotes, whereas SLC7A11 overexpression inhibits, RSL3-induced lipid peroxidation and ferroptosis (Fig. 2a–f and Supplementary Fig. 3b–e).

Cell Rep. 2019 Feb 5;26(6):1544-1556.e8. ²: A Genome-wide haploid genetic screen identifies ABCC1/MRP1-mediated GSH efflux sensitizes cells to RSL3-induced ferroptosis in H1299 and U2OS cells (Fig. 4A).

J Hepatol. 2023 Aug;79(2):362-377. ³: ATF4-deficient mouse hepatocytes are more sensitive to RSL3-induced ferroptosis compared to wild-type cells, due to reduced SLC7A11 expression. Reconstitution of SLC7A11 expression in ATF4-deficient hepatocytes provided protection against RSL3-induced ferroptosis (Fig. 3F).

Redox Biol. 2023 Sep;65:102833. ⁴: HT-29 cells became significantly more resistant to RSL3-induced ferroptotic cell death following ectopic SLC7A11 overexpression (Fig. 3P).

4. While the authors have clarified their use of different erastin concentrations in various experiments, these differences may still introduce potential confounding variables that could compromise the validity of their conclusions.

To further improve the consistency of erastin concentration used in the *in vitro* assays, we repeated the flow cytometry and GSH assays with 2.5 µM erastin (Fig. 1F; Fig. 3A; Fig. 5B; Supplementary Fig. 4B) to be consistent with the rest of the manuscript. Furthermore, we also evaluated ferroptosis induction with a wide range of erastin

concentrations and treatment duration to include all the conditions used in our study (**Supplementary Fig. 1D; Supplementary Fig. 2; Supplementary Fig. 3C and D**).

5. The doses of compounds used in this study are generally higher than the IC₅₀ values, which may lead to off-target effects. While we acknowledge that most IC₅₀ values are determined in cell-free systems, the 50-100 fold higher doses employed here could result in unintended effects.

We acknowledge that the doses of compounds used in this study may have unintended off-target effects but this criticism may be used for **any** study using **any** range of drug concentration. With the long history and widespread acceptability for using 4 μ 8c in the UPR research field, one should be able to draw meaningful conclusions based upon the data shown. **For our studies, we have used the same concentrations and identical experimental conditions as other UPR researchers.**

In the original study that identified 4 μ 8c (CB5305630)⁵, although the cell-free IC₅₀ was defined as 61.58 nM (Fig. 1A-D, left panel below), in the cell-based assays (RT-PCR) **32 μ M** of 4 μ 8c was required to achieve complete XBP1s inhibition (Fig. 5A-B, right panel below). **These data clearly demonstrate the difference between cell-free and cell-based activities of small molecule compounds.** Investigators in this field standardly use the doses of 4 μ 8c that we used in our study and it is generally accepted that these concentrations achieve potent and selective inhibition of XBP1s.

In the UPR research field, **30 – 100 μ M** 4 μ 8c is standardly used in **cell-based assays** to inhibit IRE1 α . Some examples can be found in numerous recent high-impact publications as shown below.

- **Cell Rep.** 2023 Sep 26;42(9):113130. **IMR90 cells. 50 μ M** 4 μ 8c⁶.
- **J Exp Med.** 2023 Nov 6; 220(11): e20230106. **16HBE cells. 35 μ M** 4 μ 8c⁷.
- **Nat Commun.** 2023; 14: 5414. **Primary human trophoblasts. 50 μ M** 4 μ 8c⁸.
- **EMBO J.** 2022 Aug; 41(16): e110501. **Mouse neurons. 40 μ M** 4 μ 8c⁹.

- *eLife*. 2022; 11: e75072. Myometrial cells. **80 μM 4 μ8c** ¹⁰.
- *Nat Cell Biol*. 2022 Sep; 24(9): 1422–1432. A375 human melanoma cells. **50 μM 4 μ8c** ¹¹.
- *J Clin Invest*. 2022 Sep 1; 132(17): e153519. Mouse colonoids. **50 μM 4 μ8c** ¹².
- *Nat Cell Biol*. 2023; 25(5): 726–739. Primary macrophages. **100 μM 4 μ8c** ¹³.

5. Furthermore, the animal study using the kidney I/R model fails to adequately address concerns. This model does not represent a pure ferroptosis-induced damage model, and it is widely accepted that other cell death modalities may be involved in this process. Additionally, the animal study lacks tumor models and does not incorporate other cell death inhibitors or genetic models, limiting its relevance and applicability.

Our study reported a previously unknown role for IRE1 α in regulating ferroptosis. The immediate implication of this work is in the regulation of normal tissue damage. There are currently no ferroptosis inducers in clinical use for cancer therapy whereas IRE1 α inhibitors in clinical trials (e.g., ORIN1001 in NCT03950570) would have obvious relevance to protect normal tissues from ferroptosis-induced damage. Therefore, the kidney I/R model is highly relevant and suitable for our study. ***This model has been widely used by the ferroptosis research field to study the contribution of ferroptosis to tissue damage in vivo*** ¹⁴⁻¹⁶. In the post-I/R kidney tissues in our study, while we detected robust induction of lipid peroxidation/ferroptosis marker 4-HNE (Fig. 6 and 7), we did not detect noticeable apoptosis and autophagy markers, cleaved caspase 3 and processing of LC3B, respectively (Supplementary Fig. 9E) in the injured kidneys. These results suggest that ***ferroptosis is the major type of cell death induced by I/R in the kidney***, which is consistent with previous reports ^{15,17,18}.

As ferroptosis is a major contributor to cell death associated with ischemia-reperfusion injuries, ferroptosis inhibition is a potential therapeutic approach for the treatment of conditions associated with ischemic damage, including brain stroke, ischemic heart disease and injuries to the liver and kidney, which collectively represent a major impact on global health (X. Jiang et al., *Nat. Reviews Mol. Cell Biology*, 2021 ¹⁹). Small molecule IRE1 α inhibitors have been under development for more than a decade and are currently being tested in a Phase I/II clinical trial (NCT03950570). The **clinical significance and impact** of our study are substantial as advancements in understanding the clinical applications of IRE1 α inhibitors are critical to developing effective therapeutic approaches based upon modulating ferroptosis in ischemia-reperfusion injuries.

The reviewer also pointed out that our study lacks tumor models therefore limiting its relevance and applicability. As ferroptosis is recognized as a tumor suppressor mechanism and also induced by cancer therapeutic treatment (e.g. radiation therapy), it would be logical to test activating IRE1 α 's RIDD activity to enhance ferroptosis induction in tumor cells. ***However, the development of IRE1 α activators or ferroptosis inducers for cancer therapy is well beyond the scope of this study.*** To date, no chemical compounds have been identified to specifically activate the RIDD activity of IRE1 α nor are there been any ferroptosis inducers promising enough for clinical testing.

We strongly believe that any additional animal studies, including tumor models, are not required to substantiate the significance of our findings. **We have discussed this point with the Editor who has agreed that additional animal studies are not needed for further consideration of this manuscript.** Given the additional data that are provided here and in the prior revision, we appreciate this reviewer's understanding of our position.

Responses to Reviewer #2:

The authors addressed well most of the points I raised. My first major point was address in part, only. Nevertheless, I recommend for publication of this revised version.

We appreciate Reviewer #2's positive feedback on our additional work to improve the current study.

Responses to Reviewer #3:

The authors have revised their manuscript according to my previous comments and addressed most of my concerns. They have added new experiments to show the effect of IRE1 α on ferroptosis in a mouse model of renal ischemia-reperfusion injury, and to test the effect of IRE1 α on ferroptosis in a human renal cell line. They have also added more details on how they selected the dosage of IRE1 α inhibitor and why they used different concentrations of erastin in different experiments. They have corrected the typo in the gene symbol of ACSL4 and explained the difference between Ern1 and IRE1 α . They have also performed additional experiments to characterize the different cell death pathways in their models and showed that neither apoptosis nor autophagy was significantly induced by ferroptosis stimuli or affected by IRE1 α status.

I appreciate the authors' efforts in improving their manuscript and I think their study is interesting and valuable. However, I still have some questions and suggestions that I hope the authors can address before publication:

1. How did you distinguish the direct and indirect effects of IRE1 α on ferroptosis? Did you consider the possibility that IRE1 α may affect ferroptosis through other downstream factors or pathways?

We thank the reviewer for bringing up this important subject, as in this study we fully considered these possibilities and addressed the key questions experimentally. IRE1 α is well-known to regulate its downstream functions through activating the transcription factor XBP1 by non-canonical splicing of XBP1 mRNA. As shown in **Fig. 1G**, we demonstrated that the acquired resistance to ferroptosis through IRE1 α loss is NOT due to the loss of the activation of XBP1 by IRE1 α , as forced overexpression of the activated form of XBP1 (XBP1s) failed to re-sensitize the IRE1 α -null cells to ferroptosis. As shown in **Fig. 2B**, knockdown of XBP1 expression did not impact cellular sensitivity

to erastin-induced ferroptosis. IRE1 α also activates JNK through TRAF2. However, as shown in **Supplementary Fig. 9C**, in HK-2 cells, ferroptosis induction through either hypoxia-reperfusion or RSL3 treatment did not activate JNK. Based on these observations, we reasoned that the most likely mechanism is that IRE1 α directly regulates cellular sensitivity to ferroptosis through its RIDD (regulated Ire1-dependent decay) activity. Our mechanistic studies demonstrated that through RIDD, IRE1 α indeed down-regulates multiple key components of the cellular ferroptosis defense machinery, including GCLC and SLC7A11.

2. How did you evaluate the specificity and safety of IRE1 α inhibitors? Did you test the effects of IRE1 α inhibitors on other cellular functions and organs?

We thank the reviewer for pointing out this consideration important for the therapeutic applications of IRE1 α inhibitors. The IRE1 α inhibitors (4 μ 8c and MKC9989) used in our study are salicylaldehyde derivatives that specifically inhibit the RNase activity of IRE1 α ^{5,20,21}. The specificity of 4 μ 8c used in our *in vivo* studies was demonstrated by the fact that the RNase activity of the IRE1 α -related endoribonuclease, RNase L, was unaffected by 4 μ 8c⁵. This initial finding was further confirmed by additional studies²². The *in vivo* safety as well as general effects on normal organs of this class of compounds were well characterized in a recent study by *Zhao et al.*²³ In this study, a 4th-generation salicylaldehyde class inhibitor MKC8866 was tested *in vivo*. While MKC8866 suppressed MYC-overexpressing tumor growth *in vivo*, no tissue damage was detected in normal organs including liver, pancreas, kidney, lung, heart, and small intestine. MKC8866 didn't induce apoptosis in liver or pancreas, either. Furthermore, food intake and glucose levels were unchanged during treatment. These findings suggested that this class of IRE1 α inhibitors have minimal *in vivo* toxicity in normal tissues and organs.

3. How did you explain the different effects of IRE1 α on ferroptosis in normal cells and cancer cells? Did you explore the differential expression and regulation of IRE1 α in different cell types and tissues?

In our study, we found that the role of IRE1 α in regulating ferroptosis is shared by both cancer cells (MDA-MB-231 triple-negative breast cancer and Panc-1 pancreatic cancer cells) and normal cells (mouse embryonic fibroblasts and HK-2 normal human kidney epithelial cells) as shown in **Fig. 1A-C** and **Supplementary Fig. 9D**. We also showed that this function of IRE1 α is evolutionarily conserved in *C. elegans* (**Fig. 4** and **Supplementary Fig. 7**). More importantly, this function of IRE1 α requires only its basal activity, as during ferroptosis induction by either erastin treatment, cystine starvation or RSL3 treatment no unfolded protein response (UPR) activation was detected (**Fig. 2A** and **Supplementary Fig. 4G**). Therefore, we believe that this novel activity of IRE1 α in regulating ferroptosis sensitivity is a basal function that is separate from IRE1 α 's role in stress responses. However, as pathological processes like ischemic reperfusion injury may depend on this activity of IRE1 α , pharmacological inhibition of IRE1 α is a promising therapeutic strategy to alleviate the associated damages.

4. How did you assess the importance and advantage of IRE1 α in ferroptosis regulation compared to other known ferroptosis regulators?

Since the discovery of ferroptosis, there have been a plethora of cellular ferroptosis regulators identified. The best characterized regulators include those responsible for PUFA synthesis (e.g. ACSL4, LPCAT3 and ALOXs), iron metabolism, and anti-ferroptotic defense mechanisms (e.g. SLC7A11, GCLC, GPX4, FSP1, DHODH, and GCH-1). These regulators primarily regulate the transcription or post-translational modifications of core ferroptosis proteins. In contrast to these canonical regulatory mechanisms, we are one of the first to report that IRE1 α mainly serves as **a surveillance mechanism to ensure the inducibility of ferroptosis through the cleavage (via RIDD) and downregulation of mRNA encoding key defense regulators including SLC7A11 and GCLC**. As this regulatory function of IRE1 α acts through modulating the cellular level of glutathione, it is likely to link to IRE1 α 's broader roles in regulating cellular redox balance and anti-infection²⁴. We believe our novel findings will promote the scientific research community to further explore IRE1 α 's role in related cellular functions and extend our understanding of ferroptosis mechanisms.

I look forward to reading the authors' response and the final version of their manuscript.

We appreciate the reviewer's valuable suggestions that further clarify and strengthen our conclusions. However, due to page limit we are not able to include these additional discussions in the manuscript. We will include the detailed discussion in a future commentary or review article referring to the current study. Furthermore, as responses to reviewer comments will also be published online by Nature Communications, the audience will have access to our discussion.

Responses to Reviewer #4:

The Authors have appropriately responded to my initial review.

Following a second read of the manuscript I recommend that 'IRE1 α ' be defined in the abstract (as is done for GPX4, FSP1, etc) when first introduced. Similarly, XBP1 needs to be defined.

We thank the reviewer for this valuable suggestion. In the revised manuscript we defined the full name of "IRE1 α " in the abstract and "XBP1" when it shows up the first time.

I would also recommend that complete (i.e uncropped) images of all immunoblots be included in the supplemental data. I don't have any concerns with the data presented in this submission. However, there are frequent issues with immunoblots data, so full transparency of data (raw image) should be a mandated requirement for publication.

Complete and uncropped images of immunoblots used in this manuscript are included in the Source Data file submitted together with the manuscript files.

Once these minor issues are addressed I recommend this submission be accepted for publication.

Responses to Reviewer #5:

The authors have discovered non-canonical function of IRE1 α in the down regulation of GCLC and SLC7A11 independent to UPR function. This will significantly contribute to a better understanding of metabolic regulation to prevent ferroptosis. In the revised manuscript, the authors well addressed most of reviewer's questions thereby improving the overall quality of the results. The molecular mechanism of IRE1 α mediated GCLC and SLC7A11 mRNA degradation is more solid. They also have provided more robust in-vivo experimental results with GEMM model strongly supporting therapeutic potential of IRE1 α inhibition against ferroptosis mediated pathogenic condition. Overall, this manuscript meets enough quality to be published in this journal.

I have one minor suggestion for the authors: Please correct the order of supplementary figures, as they are not matched well with the order of description in the manuscript.

We thank the reviewer for reminding us of the order when citing some of the Supplementary Figures. In the revised manuscript we have performed rigorous proofreading to make sure that the cited figure numbers match the corresponding figures. As there are a large number of supplementary figures (10 in total) in the current submission to support both the Results and Discussion sections, sometimes it is technically difficult to maintain a complete linear order when citing these figure panels in the manuscript text. We appreciate the reviewer's understanding of our position.

References:

- 1 Zhang, Y. *et al.* mTORC1 couples cyst(e)ine availability with GPX4 protein synthesis and ferroptosis regulation. *Nat Commun* **12**, 1589 (2021). <https://doi.org:10.1038/s41467-021-21841-w>
- 2 Cao, J. Y. *et al.* A Genome-wide Haploid Genetic Screen Identifies Regulators of Glutathione Abundance and Ferroptosis Sensitivity. *Cell Rep* **26**, 1544-1556 e1548 (2019). <https://doi.org:10.1016/j.celrep.2019.01.043>
- 3 He, F. *et al.* ATF4 suppresses hepatocarcinogenesis by inducing SLC7A11 (xCT) to block stress-related ferroptosis. *J Hepatol* **79**, 362-377 (2023). <https://doi.org:10.1016/j.jhep.2023.03.016>
- 4 Saini, K. K. *et al.* Loss of PERK function promotes ferroptosis by downregulating SLC7A11 (System Xc(-)) in colorectal cancer. *Redox Biol* **65**, 102833 (2023). <https://doi.org:10.1016/j.redox.2023.102833>

- 5 Cross, B. C. *et al.* The molecular basis for selective inhibition of unconventional mRNA splicing by an IRE1-binding small molecule. *Proc Natl Acad Sci U S A* **109**, E869-878 (2012). <https://doi.org:10.1073/pnas.1115623109>
- 6 Takasugi, M. *et al.* CD44 correlates with longevity and enhances basal ATF6 activity and ER stress resistance. *Cell Rep* **42**, 113130 (2023). <https://doi.org:10.1016/j.celrep.2023.113130>
- 7 Moniruzzaman, M. *et al.* Interleukin-22 suppresses major histocompatibility complex II in mucosal epithelial cells. *J Exp Med* **220** (2023). <https://doi.org:10.1084/jem.20230106>
- 8 Jash, S. *et al.* Cis P-tau is a central circulating and placental etiologic driver and therapeutic target of preeclampsia. *Nat Commun* **14**, 5414 (2023). <https://doi.org:10.1038/s41467-023-41144-6>
- 9 Wolzak, K. *et al.* Neuron-specific translational control shift ensures proteostatic resilience during ER stress. *EMBO J* **41**, e110501 (2022). <https://doi.org:10.15252/embj.2021110501>
- 10 Chen, L. *et al.* Interleukin-33 regulates the endoplasmic reticulum stress of human myometrium via an influx of calcium during initiation of labor. *Elife* **11** (2022). <https://doi.org:10.7554/eLife.75072>
- 11 Verma, S. *et al.* NRF2 mediates melanoma addiction to GCDH by modulating apoptotic signalling. *Nat Cell Biol* **24**, 1422-1432 (2022). <https://doi.org:10.1038/s41556-022-00985-x>
- 12 Grey, M. J. *et al.* The epithelial-specific ER stress sensor ERN2/IRE1beta enables host-microbiota crosstalk to affect colon goblet cell development. *J Clin Invest* **132** (2022). <https://doi.org:10.1172/JCI153519>
- 13 Ji, Y. *et al.* SEL1L-HRD1 endoplasmic reticulum-associated degradation controls STING-mediated innate immunity by limiting the size of the activable STING pool. *Nat Cell Biol* **25**, 726-739 (2023). <https://doi.org:10.1038/s41556-023-01138-4>
- 14 Friedmann Angeli, J. P. *et al.* Inactivation of the ferroptosis regulator Gpx4 triggers acute renal failure in mice. *Nat Cell Biol* **16**, 1180-1191 (2014). <https://doi.org:10.1038/ncb3064>
- 15 Linkermann, A. *et al.* Synchronized renal tubular cell death involves ferroptosis. *Proc Natl Acad Sci U S A* **111**, 16836-16841 (2014). <https://doi.org:10.1073/pnas.1415518111>
- 16 Lee, H. *et al.* Energy-stress-mediated AMPK activation inhibits ferroptosis. *Nat Cell Biol* **22**, 225-234 (2020). <https://doi.org:10.1038/s41556-020-0461-8>
- 17 Tonnus, W. & Linkermann, A. The in vivo evidence for regulated necrosis. *Immunol Rev* **277**, 128-149 (2017). <https://doi.org:10.1111/imr.12551>
- 18 Tonnus, W. *et al.* Dysfunction of the key ferroptosis-surveilling systems hypersensitizes mice to tubular necrosis during acute kidney injury. *Nat Commun* **12**, 4402 (2021). <https://doi.org:10.1038/s41467-021-24712-6>
- 19 Jiang, X., Stockwell, B. R. & Conrad, M. Ferroptosis: mechanisms, biology and role in disease. *Nat Rev Mol Cell Biol* **22**, 266-282 (2021). <https://doi.org:10.1038/s41580-020-00324-8>

- 20 Volkmann, K. *et al.* Potent and selective inhibitors of the inositol-requiring enzyme 1 endoribonuclease. *J Biol Chem* **286**, 12743-12755 (2011). <https://doi.org:10.1074/jbc.M110.199737>
- 21 Sanches, M. *et al.* Structure and mechanism of action of the hydroxy-aryl-aldehyde class of IRE1 endoribonuclease inhibitors. *Nat Commun* **5**, 4202 (2014). <https://doi.org:10.1038/ncomms5202>
- 22 Cairrao, F. *et al.* Pumilio protects Xbp1 mRNA from regulated Ire1-dependent decay. *Nat Commun* **13**, 1587 (2022). <https://doi.org:10.1038/s41467-022-29105-X>
- 23 Zhao, N. *et al.* Pharmacological targeting of MYC-regulated IRE1/XBP1 pathway suppresses MYC-driven breast cancer. *J Clin Invest* **128**, 1283-1299 (2018). <https://doi.org:10.1172/JCI95873>
- 24 Guimaraes, E. S. *et al.* The endoplasmic reticulum stress sensor IRE1alpha modulates macrophage metabolic function during *Brucella abortus* infection. *Front Immunol* **13**, 1063221 (2022). <https://doi.org:10.3389/fimmu.2022.1063221>